# The N-terminal region of DNMT3A engages the nucleosome surface to aid chromatin recruitment

Hannah Wapenaar[1], Gillian Clifford[1], Willow Rolls [1], Moira Pasquier[2], Hayden Burdett[1], Yujie Zhang [1], Gauri Deák[1], Juan Zou [1], Christos Spanos [1], Mark R D Taylor[1], Jacquie Mills [1,7], James A Watson [1], Dhananjay Kumar[1], Richard Clark [3], Alakta Das[1], Devisree Valsakumar [1,4], Janice Bramham[5], Philipp Voigt [1,4], Duncan Sproul [2,6] & Marcus D Wilson [1,5✉]

## Abstract

**DNA methyltransferase 3A (DNMT3A) plays a critical role in establishing and maintaining DNA methylation patterns in vertebrates. Here we structurally and biochemically explore the interaction of DNMT3A1 with diverse modified nucleosomes indicative of different chromatin environments. A cryo-EM structure of the full-length DNMT3A1-DNMT3L complex with a H2AK119ub nucleosome reveals that the DNMT3A1 ubiquitin-dependent recruitment (UDR) motif interacts specifically with H2AK119ub and makes extensive contacts with the core nucleosome histone surface. This interaction facilitates robust DNMT3A1 binding to nucleosomes, and previously unexplained DNMT3A disease-associated mutations disrupt this interface. Furthermore, the UDR-nucleosome interaction synergises with other DNMT3A chromatin reading elements in the absence of histone ubiquitylation. H2AK119ub does not stimulate DNMT3A DNA methylation activity, as observed for the previously described H3K36me2 mark, which may explain low levels of DNA methylation on H2AK119ub marked facultative heterochromatin. This study highlights the importance of multivalent binding of DNMT3A to histone modifications and the nucleosome surface and increases our understanding of how DNMT3A1 chromatin recruitment occurs.**

**Keywords** Chromatin; Cryo-EM; DNA Methyltransferase; Histone; Epigenetics
**Subject Categories** Chromatin, Transcription & Genomics; Structural Biology

## Introduction

Upwards of 80% of CpG sites in the genome are methylated, but this is not equally distributed. Patterns of cytosine methylation are tissue-specific and heritable, which affects gene expression and subsequent cell fate determination. The correct maintenance and positioning of DNA methylation is critical to normal cellular function. Accordingly, aberrant DNA methylation patterns are found in numerous disorders including cancers, developmental and neurological diseases (Janssen and Lorincz, 2022; Plass et al, 2013).

Vertebrates have three methyltransferase proteins that deposit DNA methylation: DNMT1, DNMT3A and DNMT3B. DNMT3A and DNMT3B deposit de novo CpG methylation during differentiation, but show continued expression and activity into adulthood (Feng et al, 2005; Okano et al, 1999; Wu et al, 2010). Despite high conservation between the two proteins, DNMT3A and DNMT3B exhibit different chromatin binding preferences, different catalytic activity, and different interaction partners (Baubec et al, 2015; Gopalakrishnan et al, 2009; Morselli et al, 2015; Taglini et al, 2024). There are two main isoforms of DNMT3A (Fig. 1A) (Chen et al, 2003; Weisenberger et al, 2002): DNMT3A1 is the full-length form of the protein, while DNMT3A2 utilises an alternative promoter and as such lacks the divergent N-terminal region of the full-length (Manzo et al, 2017). These isoforms show different expression patterns, with DNMT3A2 expressed during early development and in the germline and DNMT3A1 being the predominant postnatal isoform (Gu et al, 2022). However, the functional biochemical differences between isoforms have not been fully unravelled.

It is unclear how different regions of the genome recruit specific DNA methyltransferases. For instance, CpG islands are not normally methylated despite high CpG concentration (Brinkman et al, 2012). The fundamental repeating unit of chromatin, the nucleosome, is integral to this process. The nucleosome acts as a landing platform that integrates many signalling processes to help define genomic loci by extensive histone post-translational modification (Bannister and Kouzarides, 2011). The core

[1]Wellcome Centre for Cell Biology, University of Edinburgh, Michael Swann Building, Kings Buildings, Mayfield Road, Edinburgh EH9 3JR, UK. [2]MRC Human Genetics Unit, Institute of Genetics and Cancer, University of Edinburgh, Edinburgh, UK. [3]Edinburgh Clinical Research Facility, University of Edinburgh, Edinburgh, UK. [4]Epigenetics Programme, Babraham Institute, Cambridge CB22 3AT, UK. [5]Institute of Quantitative Biology, Biochemistry and Biotechnology, University of Edinburgh, Michael Swann Building, Edinburgh EH9 3JR, UK. [6]CRUK Edinburgh Centre, Institute of Genetics and Cancer, University of Edinburgh, Edinburgh, UK. [7]Present address: Cancer Research UK Scotland Institute, University of Glasgow, Bearsden, Glasgow G61 1BD, UK. ✉E-mail: marcus.wilson@ed.ac.uk

**A**

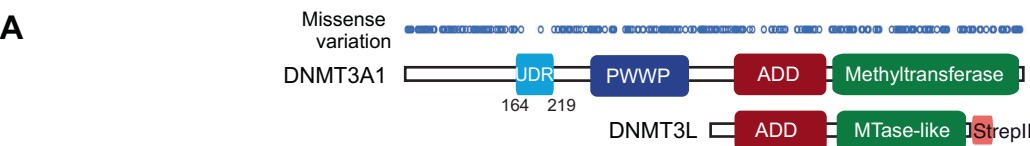

**B**

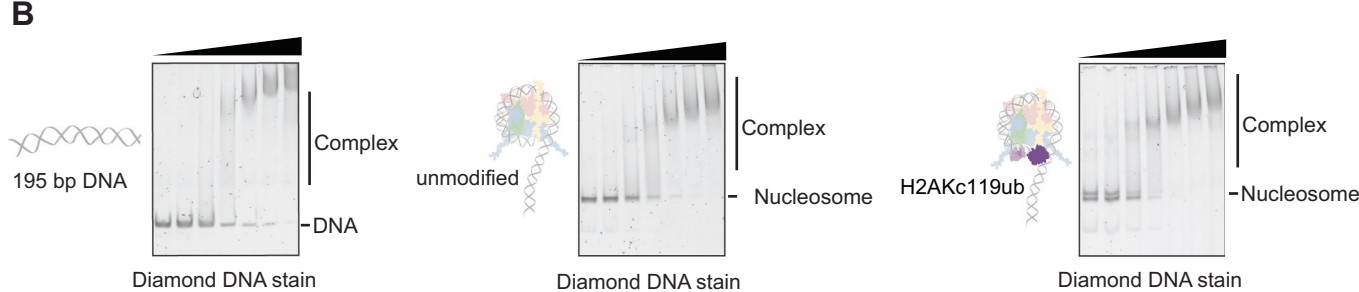

**C**

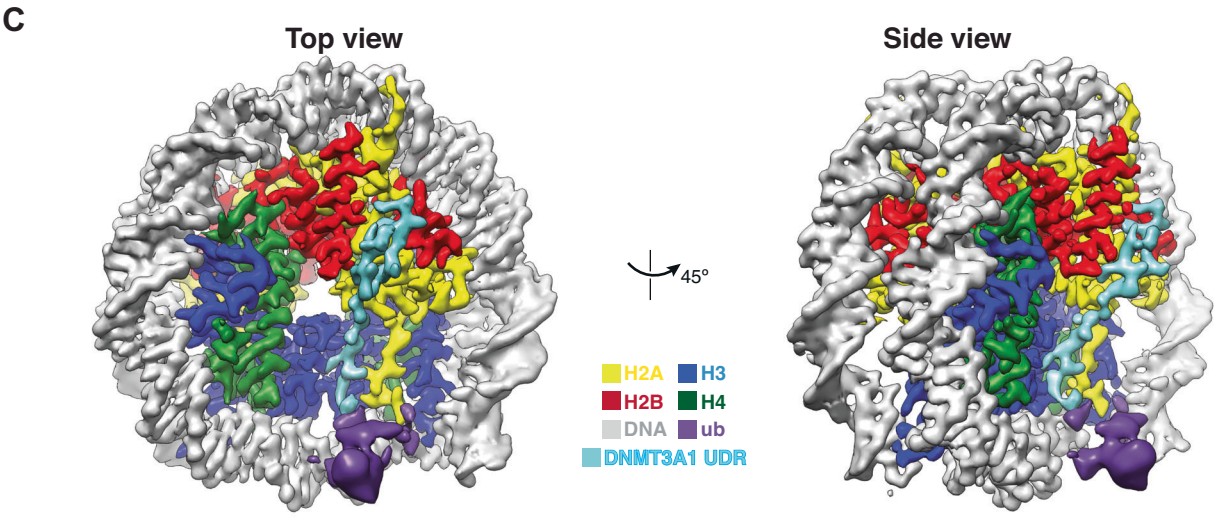

Top view

Side view

45°

H2A    H3
H2B    H4
DNA    ub
DNMT3A1 UDR

**D**    **E**

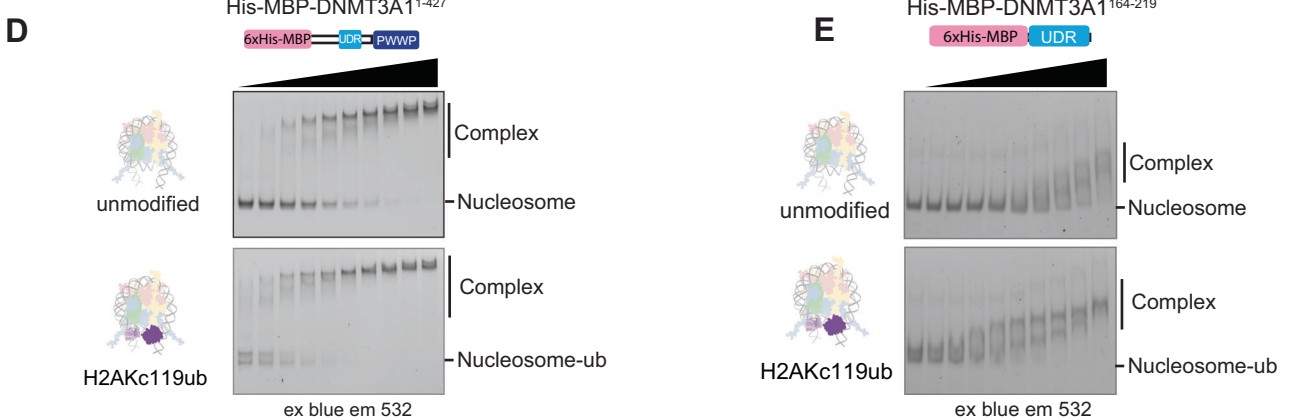

His-MBP-DNMT3A1¹⁻⁴²⁷

His-MBP-DNMT3A1¹⁶⁴⁻²¹⁹

◄ **Figure 1. Structure of DNMT3A1-DNMT3L bound to a H2AK119ub-nucleosome reveals DNMT3A1 interactions on the nucleosome surface.**

(A) Schematic overview of constructs used for cryo-electron microscopy. DNMT3A1-DNMT3L-StrepII was incubated with H2AK119ub nucleosomes wrapped with 195 bp DNA and SAM cofactor. 1D plot of known missense mutations (blue ovals) taken from the gnomAD database, shows region with large sparsity of missense mutations in population (0.26 = frequency of variants in domain/frequency of variants in full-length protein). (B) EMSA comparing binding of full-length DNMT3A1-DNMT3L-StrepII to 195 bp-wrapped unmodified and H2AKc119ub nucleosomes and free 195 bp DNA. Limiting amounts (8 nM) of nucleosomes or DNA were incubated with increasing concentrations (0–2666 nM, 2x dilution series) of full-length DNMT3A1-DNMT3L-StrepII. Complexes were resolved by native-PAGE, stained with diamond stain and imaged using blue light. Representative gels shown of one of two independent experiments, concentrations 41–2666 nM shown for clarity. (C) Surface rendering of the local resolution-filtered focused 3.1 Å resolution DNMT3A-3L: H2AK119ub Nucleosome complex map viewed along the DNA axis and the 45° rotation. Density was segmented and coloured according to local histone (coloured by convention), DNA (grey), Ubiquitin (purple) and UDR (cyan) features. Ubiquitin could not be readily placed in the attributed density and is displayed at 0.13 threshold, compared to the rest of the map at 0.35. (D) EMSA comparing binding of DNMT3A1$^{1-427}$ to unmodified and H2AKc119ub nucleosomes wrapped with 5′ FAM labelled 175 bp Widom601 DNA (FAM = fluorescein). Nucleosomes (2.3 nM) were incubated with His-MBP-DNMT3A1$^{1-427}$ (0–4000 nM, 1.5x dilution series) and resolved by native-PAGE. Imaging for Fluorescent DNA signal was performed using blue light excitation and 532 nm emission filters. Representative gels show concentrations 104–4000 nM for clarity. Experiment repeated four times. (E) EMSA comparing binding of the UDR of DNMT3A1 (DNMT3A1$^{164-219}$) to unmodified and H2AKc119ub nucleosomes wrapped with 5′ FAM labelled 175 bp Widom601 DNA (FAM = fluorescein). Nucleosomes were incubated with increasing concentrations (0–8000 nM, 2x dilution series) of His-MBP-DNMT3A1 fragments and resolved by native-PAGE. Imaging for Fluorescent DNA signal was performed using blue light excitation and 532 nm emission filters. Representative gels show concentrations 16–8000 nM, experiment done in duplicate. Source data are available online for this figure.

nucleosome surface can be bound by DNMT3A accessory proteins (Xu et al, 2020a) and histone tail binding alters DNA methylation activity (Brohm et al, 2022; Guo et al, 2015; Li et al, 2011; Zhang et al, 2010), by direct recognition of multiple histone post-translational modifications. Beyond the C-terminal catalytic domain, DNMT3A contains chromatin recognition PWWP (Pro-Trp-Trp-Pro) (Dhayalan et al, 2010) and ADD (ATRX-DNMT3-DNMT3L) (Ooi et al, 2007) domains (Fig. 1A) which help to target DNMT3A to non-transcribed intergenic regions marked by H3 Lysine 36 dimethylation (H3K36me2) and H3K4me0, respectively (Weinberg et al, 2019; Xu et al, 2020b). The PWWP domain directly recognises H3K36 methylation (Rondelet et al, 2016; Weinberg et al, 2019) and disruption of H3K36 methylation or binding leads to aberrant DNMT3A1 localisation and activity (Chen et al, 2022; Hamagami et al, 2023; Heyn et al, 2019; Sendzikaite et al, 2019), with aberrant hypermethylation at facultative heterochromatin. There appear to be two competing mechanisms for DNMT3A recruitment in vivo, with enrichment at intragenic regions marked with histone H3K36me2 but also tissue-specific recruitment to H3K27me3-H2AK119ub marked facultative heterochromatin. DNMT3A1 localises to CpG island shores in both normal (Gu et al, 2022; Manzo et al, 2017) and disease state-mimic cells (Heyn et al, 2019; Sendzikaite et al, 2019; Weinberg et al, 2021). DNMT3A1-specific recruitment to Polycomb regions has been mapped to its divergent N-terminal domain, proposed to interact directly with the H2AK119ub Polycomb mark (Gu et al, 2022; Weinberg et al, 2021; Chen et al, 2024; Gretarsson et al, 2024).

Here we structurally and biochemically explore the interaction of DNMT3A1 with H2AK119ub modified nucleosomes. A single-particle cryo-EM structure reveals that an N-terminal region of DNMT3A1 facilitates interaction with the nucleosome acidic patch, H2A and H3, facilitating specific interaction with H2AK119ub. This interaction maps to a ubiquitin-dependent recruitment (UDR) region in DNMT3A1 which contains previously unexplained disease-associated mutations. We find the same UDR region can also engage the nucleosome surface in H3K36me2 modified nucleosomes, stabilising this interaction overall. Finally, despite its strong binding to nucleosomes, DNMT3A1's catalytic activity was not proportional to its recruitment. These findings highlight a disconnect between DNMT3A recruitment and its enzymatic

activity, shedding light on the complex relationship between DNMT3A and DNA methylation of chromatinized DNA.

## Results

### Structure of DNMT3A1-DNMT3L bound to a H2AK119ub-nucleosome reveals direct DNMT3A1-nucleosome interactions

To better understand how DNMT3A1 is recruited to Polycomb-enriched regions, we determined the structure of full-length DNMT3A1 in complex with a nucleosome ubiquitylated at PRC1-targeted site H2A Lys-119. We generated unmodified and H2AKc119ub nucleosomes (Burdett et al, 2023; Long et al, 2014; Wilson et al, 2016) wrapped with the strong positioning Widom-601 DNA sequence flanked by a linker containing an optimised DNMT3A binding sequence (Fig. EV1A,B) (Gao et al, 2020; Mallona et al, 2021). The main de novo methyltransferase accessory protein in somatic cells is splice isoform DNMT3B3 (Xu et al, 2020a). We purified DNMT3A1-DNMT3B3 complex, but the yield was too low for our structural approaches. However, we could robustly make a DNMT3A1-DNMT3L complex (Figs. 1A and EV1C,D), which exhibited the expected tetrameric features. DNMT3L is typically associated with early development but described to stimulate DNMT3A1 activity (Chedin et al, 2002) and complexed with DNMT3A1 in several adult tissue types (Hata et al, 2002; Xu et al, 2020a). DNMT3A1-DNMT3L complex bound preferentially to unmodified nucleosomes in an electrophoretic mobility shift assay (EMSA) over DNA alone. Even higher affinity was observed for H2AK119ub modified nucleosomes (Fig. 1B), as reported previously for smaller fragments of DNMT3A1 (Gu et al, 2022; Weinberg et al, 2021; Chen et al, 2024; Gretarsson et al, 2024).

We preformed a complex of nucleosome-H2AK119ub with DNMT3A1-DNMT3L in the presence of S-adenosyl methionine (SAM) cofactor and determined the structure by single particle cryo-EM (Figs. 1C and EV1E,F; Appendix Figs. S1 and S2). Individual nucleosome-shaped particles could be observed in the raw data and 2D class averages were reminiscent of nucleosome structures, with some additional density attributable to ubiquitin and DNMT3A1 (Appendix Fig. S2A,B). From our 3D reconstruction the assembly appeared to be

highly mobile limiting the overall resolvability of DNMT3A1-DNMT3L, with abundant flexible density on the linker DNA which may represent the catalytic core of the DNMT3A1-DNMT3L complex (Fig. EV1E,F; Appendix Figs. S1 Map 1, S3C,D). Focussed masking of the nucleosome core and adjacent density resulted in a 3.1 Å resolution map (Fig. 1C; Appendix Figs. S1 Map 3 and S2E–G). In this map, the histone core of the nucleosome and bound DNA are well ordered and similar to other nucleosome structures (RMSD 0.39 Å, PDB 7XD1 (Ai et al, 2022)), with the extended linker DNA projecting away from the nucleosome core. We could attribute contiguous density over the C-terminal tail adjacent to lysine 119 of H2A to ubiquitin, co-incident with the linker DNA and DNMT3A1-DNMT3L densities. On the nucleosome surface, continuous non-nucleosomal density snakes from the acidic patch region of the nucleosome between histone H2A and H2B to become sandwiched in a depression between H3 and H2A prior to interaction with ubiquitin tethered over the C-terminal tail of H2A. The map was at sufficient resolution to allow us to build a discontinuous model of coiled structure of DNMT3A1 corresponding to residues 166–171 and 177–194, corresponding to the proposed UDR region of DNMT3A1 (Gu et al, 2022; Weinberg et al, 2021) with a buried surface interface of 2095 Å$^2$ (Appendix Fig. S2H; Appendix Table S1). Cross-linking mass spectrometry using zero-length cross-linker (see Methods) of the DNMT3A1-DNMT3L:nucleosome-K119ub complex showed similar links between the DNMT3A1 UDR and nucleosome surface and bound ubiquitin, in addition to expected other cross-links from the rest of the complex mapping to the flexible region of our structure (Fig. EV1G,H).

## DNMT3A1 UDR maps to residues 164–219

To validate our model and further map the minimal DNMT3A1 fragment required for interaction with nucleosome-H2AK119ub, we purified fragments of the divergent DNMT3A1 N-terminal region up to the PWWP domain. Due to robust binding of the DNMT3A1 N-terminal region to non-nucleosomal DNA (Appendix Fig. S3A–C) (Suetake et al, 2011), competitor DNA was added to EMSA assays to reduce non-specific DNA interactions. A construct containing only the N-terminal region (DNMT3A$^{1-277}$) or the N-terminal region with PWWP domain (DNMT3A$^{1-427}$) of DNMT3A1 could bind to unmodified nucleosomes but showed a clear preference binding to H2AKc119ub nucleosomes (Fig. 1D; Appendix Fig. S3D, Appendix Table S2). This suggests that this N-terminal region mediates both the ubiquitin recognition as well as additional generic nucleosome interactions.

We further purified DNMT3A1 fragments lacking stepped-intervals of ~50 residues from the N-terminus. The greatest loss of binding was observed between DNMT3A1$^{142-427}$ and DNMT3A1$^{192-427}$, confirming residues built into our model are required for H2AK119ub-nucleosome interaction (Appendix Fig. S3E,F). Indeed, in isolation, the UDR region consisting of DNMT3A1$^{164-219}$ was sufficient for preferential interaction with H2AK119ub nucleosomes (Fig. 1E). The UDR fragment displayed overall lower binding affinity to both unmodified and ubiquitylated nucleosomes, suggesting additional stabilising interactions may be present in the DNMT3A1 N-terminal region.

The N-terminal region of DNMT3A1 contains a ubiquitylation-dependent recruitment motif (UDR) (Weinberg et al, 2021), located within residues 122–219 (Gu et al, 2022). We further map the minimal UDR region to residues 164–219. Intriguingly, this UDR

region is highly invariant in the healthy population, based on the absence of missense variants found in the gnomAD population database (Fig. 1A) (Deak and Cook, 2022; Karczewski et al, 2020). This underlines the importance of the DNMT3A1 nucleosome- and ubiquitin interacting region during normal development in line with the essentiality of this region in mouse postnatal neural development (Gu et al, 2022).

## The DNMT3A1 UDR recognises generic nucleosome features

We observe binding of DNMT3A1 even in the absence of histone modification (Fig. 1B,D). Indeed, our structure shows an extensive protein interaction network formed by the UDR region on the surface of the nucleosome. This starts with the DNMT3A1 UDR and a negatively charged recess formed between H2A and H2B on the nucleosome termed the acidic patch. This is a common site of interaction for chromatin binding proteins (McGinty and Tan, 2021). Indeed, DNMT3B3 methylation adaptor protein also engages with nucleosomes via the acidic patch, but via an alternate binding mode (Xu et al, 2020a). In our structure, a clear arginine anchor (McGinty and Tan, 2021) corresponding to DNMT3A1 Arg-181 projects into canonical cavity formed between H2B αC and α2/α3 helices of H2A, interacting with the acidic patch carboxylate groups of Glu-61, Asp-90, and Glu-92 in H2A (Fig. 2B left; Appendix Fig. S4A). To verify this interaction, we neutralised the charge on the acidic patch through mutation (*acidic patch*: H2A$^{E61A/E91A/E92A}$ and H2B$^{E105A}$) (Appendix Fig. S4B). This greatly reduced binding of DNMT3A1$^{1-427}$ to H2AK119ub nucleosomes (Fig. 2C,D), confirming that the acidic patch is important for the ubiquitin-dependent interaction with DNMT3A1.

Disease-associated mutations of Arg-181 and Arg-183 have been reported in clonal haematopoiesis (Jaiswal et al, 2014), Tatton-Brown-Rahman syndrome (Tatton-Brown et al, 2018) and cancer patients (Basturk et al, 2017; Dutton-Regester et al, 2013; Huang et al, 2022), with no reported mechanism. These mutations do not affect protein stability but reduce DNMT3A1's capacity to repress transcription in a reporter assay (Huang et al, 2022). We tested whether these mutations have a direct effect on nucleosome interaction and found R181C, and to a lesser extent R183W, reduced overall binding to unmodified, H2AK119ub nucleosomes (Fig. 2E; Appendix Fig. S4C,E), while having no effect on DNA binding (Appendix Fig. S4D). This may suggest a mechanistic basis for the role of these mutations in disease states mediated via disruption of arginine anchor-acidic patch nucleosome interaction.

Extra unmodelled density over the acidic patch region suggests the UDR region loops back to further stabilise the interaction through an arginine-rich region between residues 167–171 into two adjacent negatively charged depressions behind the canonical cavity in the acidic patch. Deletions in this region reduce interaction with both unmodified and H2AKc119ub nucleosomes (Fig. 2F; Appendix Fig. S4F). This acidic patch interaction is highly extensive compared to simpler arginine anchor interactions such as the LANA peptide, but not uncommon in chromatin bind proteins (McGinty and Tan, 2021).

Internal deletions in DNMT3A1$^{1-427}$ which still maintain the Arg-181 anchor ablate both H2AK119ub and unmodified nucleosome interaction (Fig. 2F), with the most pronounced effect caused by a deletion of residues 184–191. From the structure, DNMT3A1

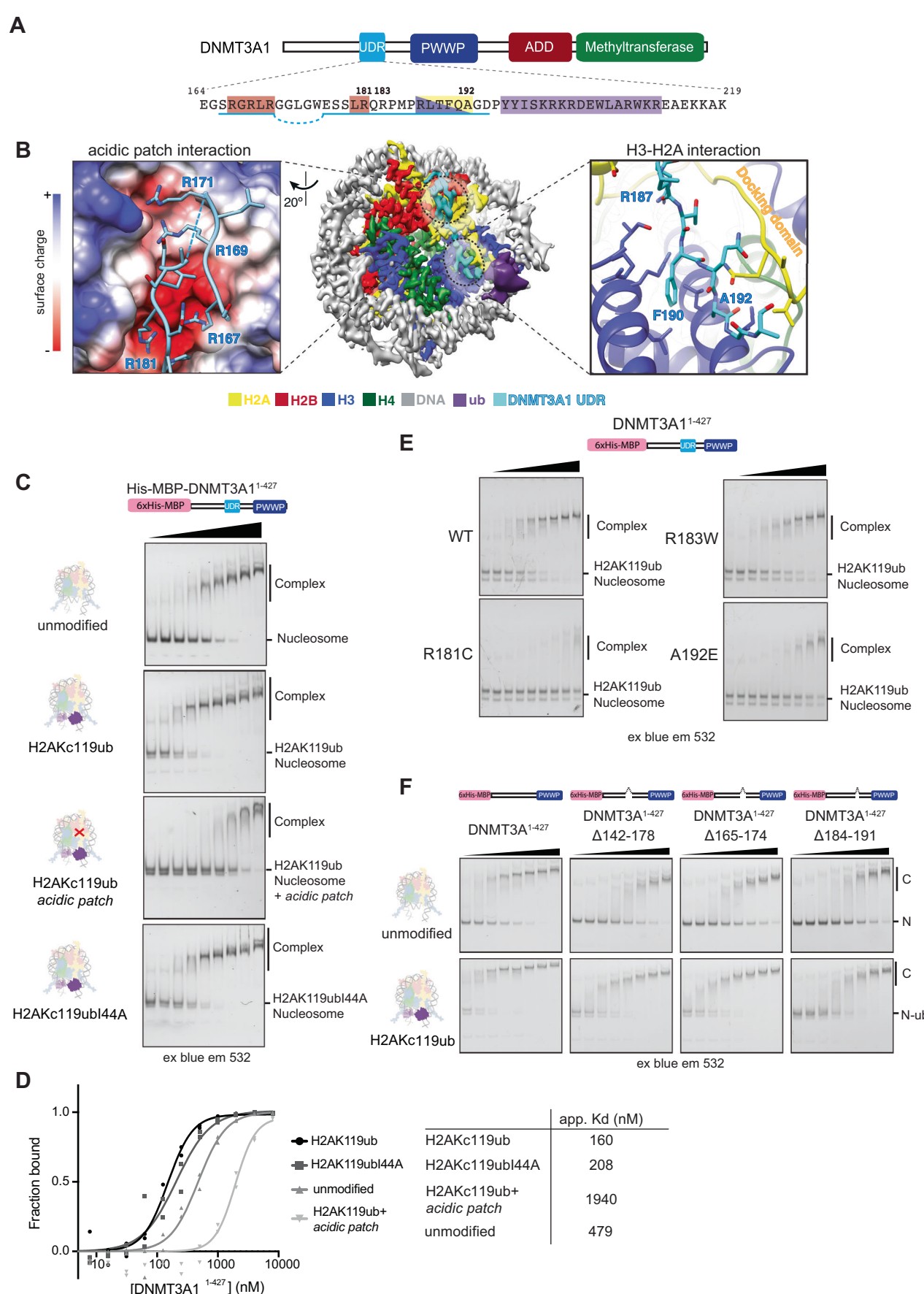

**Figure 2. The DNMT3A1 UDR region engages with nucleosome surface.**

(A) Schematic of DNMT3A1 and sequence of the UDR region. Areas underlined have been modelled into the density. Areas highlighted are involved in interaction with acidic patch (red), H2A/H3 recess (yellow/blue) and ubiquitin (purple). (B) (left) Magnified view of the H2A/H2B acidic patch-UDR interaction interface, highlighting the arginine anchor Arg181 and other positively charged residues in proximity of the acidic patch histone surface coloured according to columbic surface charge. (right) Enlarged view of H2A docking domain and H3 α2 interaction interface showing Phe190 interaction with aliphatic histone residues and H3 C-terminal interaction with backbone. (C) EMSA comparing binding of DNMT3A1$^{1-427}$ to unmodified, H2AKc119ub, H2Akc119ub acidic patch (H2A$^{E61A/E91A/E92A}$ and H2B$^{E105A}$) and H2Akc119ubI44A nucleosomes. Complexes were resolved by native-PAGE and imaged using blue light excitation and 532 nm emission filters. Gels show concentrations 32–8000 nM. (D) Quantification of (C) done with full concentration series from two experiments. Apparent Kd values calculated. (E) EMSA assay to investigate the effect of DNMT3A1 N-terminal region missense mutations in found in clinical patients R181C, R183W and A192E (0–8000 nM, 2x dilution series) on binding to H2AKc119ub nucleosomes. Gels show concentrations 31–8000 nM, experiment done in duplicate. (F) EMSA comparing binding of DNMT3A1$^{1-427}$ constructs (0–4000 nM, 1.5x dilution series) with internal deletions in the UDR to unmodified and H2AKc119ub nucleosomes. Internal deletions were created prior to (Δ142-178, Δ165-174) or after (Δ184-191) the arginine anchor Arg181 region, but still showed disruption in binding to both unmodified and ubiquitylated nucleosomes. Complexes were resolved by native-PAGE and imaged using blue light excitation and 532 nm emission filters. Gels show concentrations 234–4000 nM. Experiment performed in duplicate. Source data are available online for this figure.

UDR kinks down to pack tightly within a recess formed between the H2A-H2B dimer. Arg-187 projects into this recess, forming stabilising interactions with Asn-89 and H3 C-terminus (Appendix Fig. S4A). Interestingly, this region as well as the acidic patch is divergent in nucleosomes containing the variant H2A.Z (Suto et al, 2000) and DNMT3A1$^{1-427}$ binds drastically less well to H2A.Z nucleosomes (Appendix Fig. S4G). Even ubiquitylation at residues analogous to H2A Lys 119 (H2A.Z Kc120ub) was unable to rescue robust DNMT3A1 binding, highlighting the importance of correct nucleosome interaction for ubiquitin engagement. DNMT3A1 is absent from promoter regions (Manzo et al, 2017), which are enriched for H2A.Z. Reduced H2A.Z promoter-adjacent nucleosome binding may help this depletion along with the repulsive effect of H3K4 methylation.

Distal to the acidic patch DNMT3A1-nucleosome interactions are further stabilised by the H2A docking domain, H3 α3 helix and the C-terminal α2 helix in the opposite H3′ chain (Fig. 2B right). Multiple predicted backbone-backbone contacts are supplemented by DNMT3A1 Phe-190 making hydrophobic contacts with H3′ Leu-109, and H3 Leu-126, and Arg-129. The C-terminal carboxylate of H3 appears to form stabilising interactions with the backbone of DNMT3A1 and Arg-187. The carbonyl group of Phe-190 and sidechain of Gln-191 are aligned to form stabilising hydrogen bonds to amines in the backbone of H2A as does the sidechain of H2A Gln-112 with DNMT3A1 Ala-192 (Fig. 2B; Appendix Fig. S4A). The nucleosome interacting region of DNMT3A1 terminates with a ~60° bend at another hydrophobic region allowed by small sidechain amino acids in Ala-192 and Gly-193, directing the UDR towards the covalently attached ubiquitin. In line with this, substitution with larger sidechain amino acids, as is found in a clinically relevant mutation A192E (Giannakis et al, 2016; Huang et al, 2022; Zehir et al, 2017), reduced the DNMT3A1$^{1-427}$ interaction with H2AK119ub nucleosomes (Fig. 2E; Appendix Fig. S4C,D,E), with minimal effect on DNA binding.

## DNMT3A1 UDR-nucleosome interactions ensure specificity for H2AK119ub

The structural density attributed to ubiquitin and the section of the DNMT3A1 UDR beyond residue 194 is weaker and spread over a larger volume than other areas of the map, preventing us from confidently building this into our model. Nevertheless, there is a degree of stabilisation due to the UDR with more density on the nucleosome face with the UDR density compared to the non-bound ubiquitin on the anterior side of the nucleosome (Fig. 3A). Accordingly, ubiquitin density appears stronger than in a H2AK119ub alone structure (Ohtomo et al, 2023). The remaining portion of DNMT3A1 UDR is logically likely present in this region, forming interactions with the ubiquitin. Indeed, cross-linking mass spectrometry analysis of DNMT3A1 nucleosome H2AK119ub complex shows several cross-links between the N-terminal region and ubiquitin in addition to the structurally predicted histone surface (Fig. EV1H; Appendix Fig. S5A)

Histone ubiquitylation is a widespread modification with different signalling outcomes based on its specific position on the nucleosome surface (Fields et al, 2023; Mattiroli and Penengo, 2021). DNMT3A1 co-localises with H2AK119ub-rich sites in the genome (Gu et al, 2022; Manzo et al, 2017), rather than at other ubiquitylated histones. To assess the nature of ubiquitin-site specificity of DNMT3A1 we generated a panel of ubiquitylated nucleosomes. We included DNA damage associated H2AK15ub and H2AK127ub (Kalb et al, 2014; Mattiroli et al, 2012), active transcription mark H2BK120ub (Fleming et al, 2008; Pavri et al, 2006) and maintenance DNA methylation associated H3K18ub (Harrison et al, 2016; Ishiyama et al, 2017) (Appendix Fig. S5B). Ubiquitylation at other locations on the nucleosome did not appreciably influence the interaction of DNMT3A1$^{1-427}$; the affinity of interaction was similar to that with the unmodified nucleosome (Fig. 3B,C). This was true even for the H2A Lys-119 proximal H2AK127ub. This suggests that despite the large size of the ubiquitin post translational modification, DNMT3A1 is a specific reader for H2AK119ub. The ubiquitin site specificity is likely mediated by the interaction of DNMT3A1$^{167-193}$ with the nucleosome surface, orientating and tethering the ubiquitin interacting portion of the UDR in the correct location and orientation. We have previously seen that combining nucleosome tethering and ubiquitin recognition elements ensures a specific readout in other readers (Burdett et al, 2023; Kitevski-LeBlanc et al, 2017; Wilson et al, 2016). Indeed, while this orientation provides specificity it does not necessarily mean that the ubiquitin binding moiety is rigidly bound (Kitevski-LeBlanc et al, 2017; Rahman et al, 2022).

The density for ubiquitin is tethered over the C-terminal tail of H2A near the linker DNA as it leaves the nucleosome core (Fig. 3A). Given this proximity we tested whether interactions between linker DNA and ubiquitin may be helping to mediate

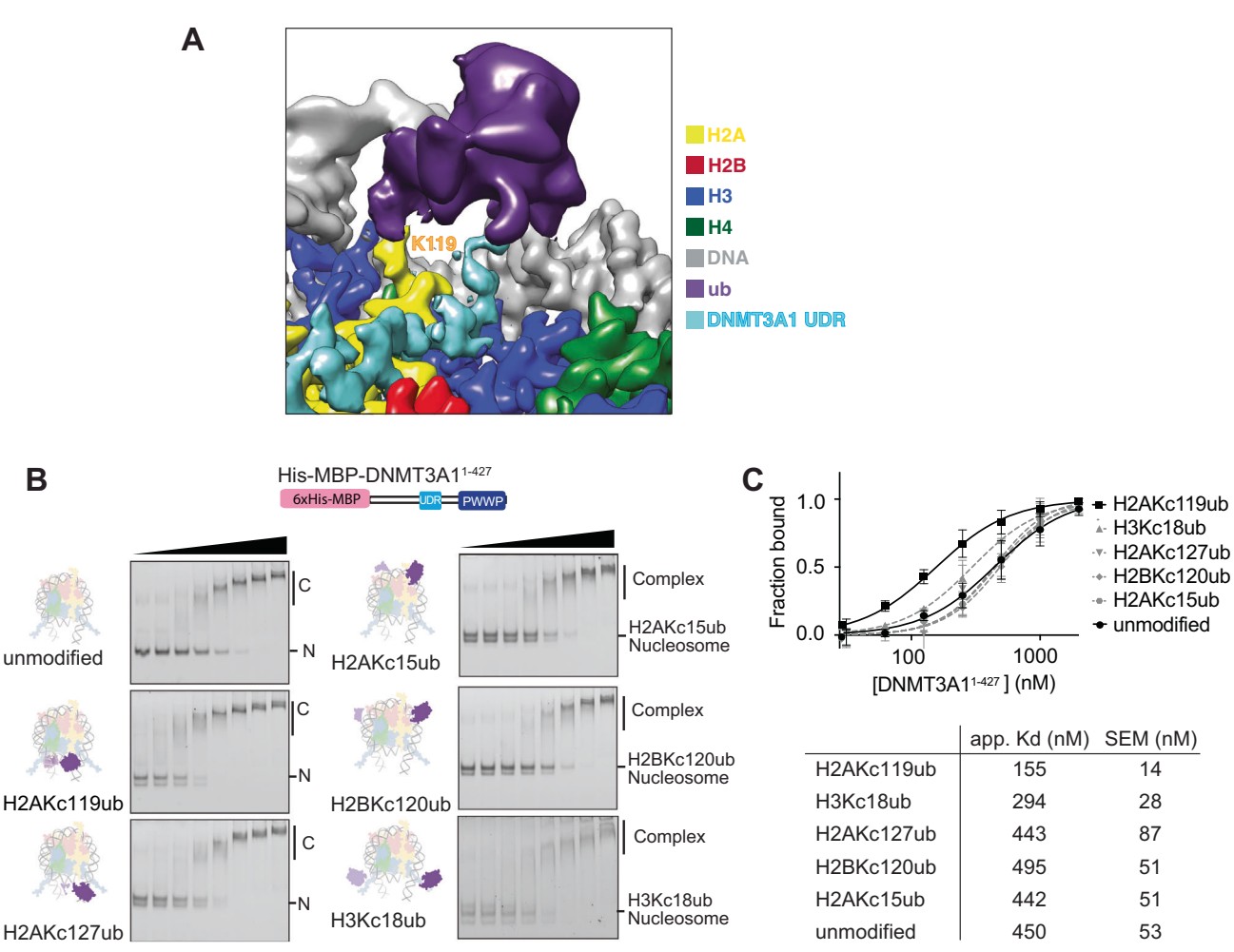

**Figure 3. DNMT3A1 shows ubiquitin specificity and binds atypically.**

(A) Magnified view of the H2AK119ub/UDR density from map 3, positioned above the C-terminal tail of H2A. Density from the nucleosome-contacting region of the UDR can be seen arcing upwards towards the density. Map displayed at 0.25 contour level apart from segmented ubiquitin density displayed at 0.08. (B) EMSA comparing binding of DNMT3A1¹⁻⁴²⁷ to nucleosomes ubiquitylated on different histone positions, leading to different sites on the nucleosome surface. Limiting amounts of nucleosomes were incubated with increasing concentrations (0–8000 nM, 2x dilution series) of His-MBP- DNMT3A1¹⁻⁴²⁷. Gels show concentrations 63–8000 nM. (C) Quantification of expert in (B) with full concentration series in triplicate. Apparent Kd values calculated with standard error of mean (SEM) listed. Source data are available online for this figure.

DNMT3A1 interaction. The UDR region contains a stretch of positively charged amino acids (Fig. 2A), which affects recruitment to H2AK119ub when mutated (Gu et al, 2022). These residues may contact DNA as part of ubiquitin interaction. If linker DNA were to be involved we would expect higher affinity to nucleosomes containing two symmetrical linkers, as both faces of the nucleosome would be bound by DNMT3A1¹⁻⁴²⁷ rather than a single asymmetric extension. However, no ubiquitin preference was observed due to DNA length, suggesting non-nucleosomal linker DNA is not important in DNMT3A1 ubiquitin specificity and interaction (Appendix Fig. S5C,D).

## DNMT3A1 engages with ubiquitin in an atypical manner

We expected that since the DNMT3A1 UDR region is involved in binding to H2AK119ub, it could function as a ubiquitin binding

domain. However, DNMT3A1¹⁴²⁻²¹⁹ did not detectably bind to ubiquitin in isolation, neither co-eluting by size exclusion chromatography (Fig. EV2A) or observable in a pull-down assay (Fig. EV2B). Indeed, ¹⁵N labelled ubiquitin shows no chemical shift perturbations upon addition of high concentrations of purified unlabelled DNMT3A1 UDR (Fig. EV2C). This suggest that the UDR region cannot be classified as a ubiquitin binding domain in isolation or as a ubiquitin interacting motif (UIM) (Komander and Rape, 2012) and can only bind within its proper nucleosomal context. Nucleosome-ubiquitin interactions mediated by short peptidic motifs or parts of domains not predicted to bind ubiquitin have been shown in several other studies (Anderson et al, 2019; Fradet-Turcotte et al, 2013; Hsu et al, 2019; Rahman et al, 2022; Valencia-Sanchez et al, 2019; Wilson et al, 2016; Worden et al, 2019; Worden et al, 2020). Similarly to DNMT3A1 UDR, these show poor to undetectable binding to ubiquitin in isolation and

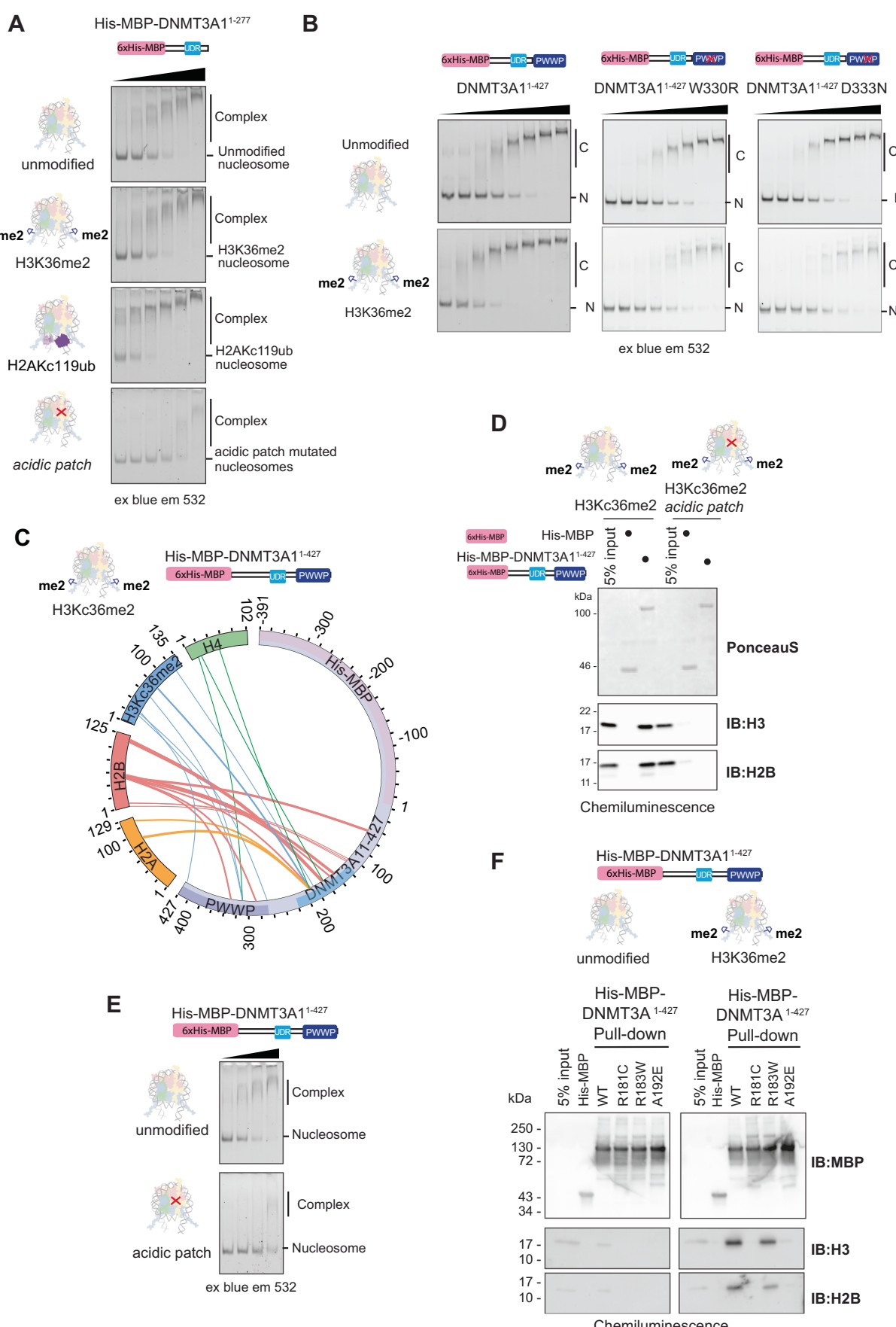

**Figure 4. The DNMT3A1 UDR region engages with nucleosome surface.**

(A) EMSAs comparing binding of His-MBP-DNMT3A1$^{1-277}$ (0–8000 nM, 2x dilution series) to unmodified, H3K36me2, H2AK119ub and acidic patch mutated nucleosomes wrapped with 5' FAM labelled 175 bp DNA (5.4 nM). Complexes were resolved by native-PAGE and imaged using blue light excitation and 532 nm emission filters. Gels show concentrations 250–8000 nM. Experiment performed in triplicate. (B) EMSA comparing binding of DNMT3A1$^{1-427}$ and Heyn-Sproul-Jackson syndrome causing mutations DNMT3A1$^{1-427}$W330R and DNMT3A1$^{1-427}$D333N to unmodified nucleosomes and H3K36me2 nucleosomes wrapped with 5' FAM labelled 175 bp Widom601 DNA. Representative gels show concentrations 32–8000 nM. EMSA of unmodified binding to DNMT3A1$^{1-427}$ also appears in Fig. 3B. (C) Cross-linking mass spectrometry of His-MBP-DNMT3A1$^{1-427}$ to H3Kc36me2 nucleosomes wrapped with 175 bp Widom-601 DNA. Circular representation shows two biological replicates combined of which one measured in triplicate. Only cross-links between DNMT3A1 and histones are shown. Cross-links weighted based on the number of times the high-confidence cross-link appears in the 4 replicates (thin = 1x, medium = 2x, wide = 3x). (D) Pull down assay comparing DNMT3A1 binding to H3Kc36me2 nucleosomes with or without mutations in the acidic patch (H2A$^{E61A/E91A/E92A}$ and H2B$^{E105A}$). DNMT3A1$^{1-427}$ or tag alone was immobilised on amylose beads and proteins were detected using PonceauS stain. Bound nucleosomes were detected using western blotting for histones H3 and H2B. (E) EMSA comparing binding of DNMT3A1$^{1-427}$ to unmodified or acidic patch mutated (H2A$^{E61A/E91A/E92A}$ and H2B$^{E105A}$) nucleosomes wrapped with 5' FAM labelled 175 bp Widom601 DNA (FAM = fluorescein). Limiting amounts (2.3 nM) of H3K36me2 nucleosomes were incubated with increasing concentrations (0–4000 nM) of His-MBP-DNMT3A1$^{1-427}$. Gels show concentrations 500–4000 nM. (F) Pull down assay to investigate the effect of DNMT3A1 N-terminal region missense mutations found in clinical patients on binding to unmodified and H3K36me2 nucleosomes. Equal amounts of wild type and mutant His-MBP-DNMT3A1$^{1-427}$ was immobilised on amylose beads and incubated with nucleosomes prior to washing and detection by western blot. Source data are available online for this figure.

leverage nucleosome surface biding to ensure specific orientation of ubiquitin interacting fragments.

Most ubiquitin binding interactions are mediated by the canonical ubiquitin hydrophobic patch, formed by Leu-8, Ile-44, Val-70 (Komander and Rape, 2012). Surprisingly, nucleosomes harbouring H2AK119ub with a I44A mutation were bound preferentially by DNMT3A1$^{1-427}$ (Fig. 2C), suggesting the canonical ubiquitin hydrophobic patch is not the main mediator of DNMT3A1 binding. Indeed, a recent study reported ubiquitin binding by DNMT3A is chiefly centred around Ile-36 and Leu-71 (Chen et al, 2024). Using alanine-scanning mutagenesis across the remaining UDR region shows that single mutations had no major effect on H2AK119ub binding (Fig. EV2D). This suggests that the ubiquitin interaction is mediated via multiple weaker interactions rather than a single critical residue. Taken together, the end of the UDR region interacts atypically with ubiquitin and only when it is in the correct nucleosomal context.

## The DNMT3A1 UDR also stabilises interaction with H3K36me2 modified nucleosomes

As we have seen that the UDR region binds to generic features of a nucleosome, we next wanted to explore whether direct nucleosome surface recognition could synergise with PWWP-mediated H3K36me2 interaction. H3K36me2 is primarily found on intergenic chromatin and directs DNMT3A methylation in these regions (Weinberg et al, 2019; Xu et al, 2020b).

We reconstituted H3K36me2 nucleosomes and compared binding in EMSA and pulldown assays to our previously characterised nucleosomes (Fig EV3A). The N-terminal fragment (DNMT3A$^{1-277}$) showed preferential binding to H2AK119ub nucleosomes over unmodified nucleosomes, which was greatly ablated by mutation of the acidic patch. As expected, the N-terminal fragment lacking the PWWP domain did not preferentially bind to H3K36me2 modified nucleosomes compared to their unmodified counterpart (Fig. 4A). However, constructs containing a PWWP domain: DNMT3A1$^{1-427}$ (Figs. 4B and EV3B) and full-length DNMT3A1-DNMT3L (Fig. EV3C) showed an increased binding to H3K36me2 containing nucleosomes. This is in line with the observed PWWP-driven localisation to H3K36-methylated regions of the genome (Dhayalan et al, 2010; Weinberg et al, 2019; Xu et al, 2020b). Furthermore, disrupting methylated-

lysine recognition using Heyn-Sproul-Jackson syndrome causing mutations in the PWWP domain (Heyn et al, 2019) removed H3K36-methylation specificity (Figs. 4B and EV3D), reducing overall binding to the same level as to unmodified nucleosomes.

The DNMT3A1 UDR region is responsible for this additional nucleosome interaction; sequential deletions of the N-terminal region confirmed that the same residues 142–192 that are required for H2AK119ub-nucleosome interaction (Appendix Fig. S3F), facilitate H3K36me2-DNMT3A1 interaction (Fig. EV3E). As previously reported (Dhayalan et al, 2010; Dukatz et al, 2019; Weinberg et al, 2019), the PWWP in isolation preferentially interacts with H3K36me2 nucleosomes, but this is relatively low affinity (Fig. EV3E–G), suggesting the UDR region helps to confer extra affinity to DNMT3A1 interaction on nucleosomes. The nucleosome surface-UDR interactions are likely maintained in this PWWP-engaged binding mode. Cross-linking mass spectrometry on DNMT3A1$^{1-427}$:H3Kc36me2-nucleosome complex (Fig. 4C; Appendix Fig. S6A,B) identified several cross-links between the N-terminal tail of H3 and PWWP domain, as expected given the H3K36me2 interaction. In addition, cross-links clustered between UDR residues of DNMT3A1 and H2A-H2B acidic patch adjacent residues (Fig. 4C; Appendix Fig. S6C). Neutralising the charge of the acidic patch on H3K36me2 (Figs. 4D and EV3C) reduced binding to DNMT3A1. Indeed, even unmodified nucleosome binding was ablated by acidic patch mutation (Fig. 4E), suggesting the DNMT3A1 N-terminal region interacts with the acidic patch independent of histone modifications. Similarly, mutation of DNMT3A1 UDR arginine anchor R181C and docking domain interactor A192E ablated binding to nucleosomes (Fig. 4F). Overall, this suggest that DNMT3A1 UDR incorporates recognition of generic nucleosome features to increase overall affinity to chromatin, in co-ordination with both H2AK119ub as well as H3K36me2 binding.

## H3K36me2 and co-incident marks do not prevent DNMT3A1 binding

We next sought to build on the complexity of the system and test marks that occur at the same genomic loci as H3K36me2. H3 Lys-27 di-methylation (H3K27me2) is found intragenically and co-occurs with H3K36me2 (Ferrari et al, 2014; Mao et al, 2015; Streubel et al, 2018; Weinberg et al, 2021). Nucleosomes bearing

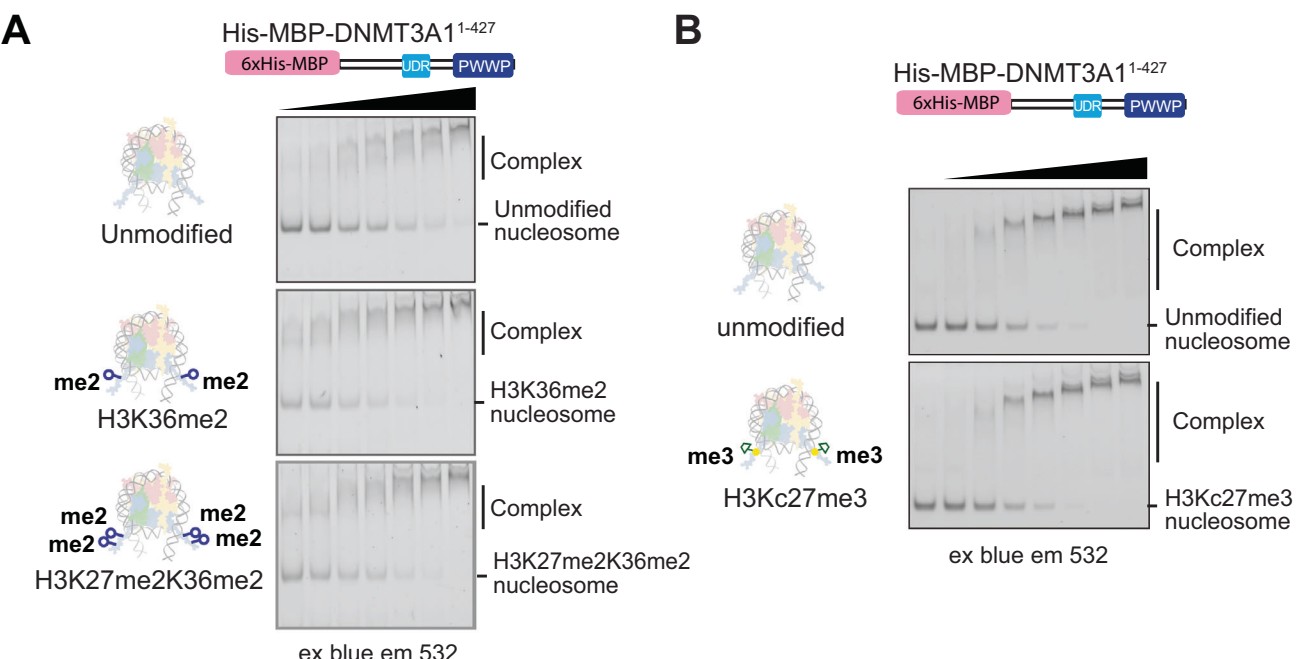

◄

**Figure 5.  DNMT3A1 can engage with multiple histone PTM marks concurrently.**

(A) EMSA comparing binding of DNMT3A1[1-427] to unmodified, H3K36me2, H3K27me2K36me2 nucleosomes. Nucleosomes were incubated with increasing concentrations (0–2000 nM, 1.5x dilution series) of His-MBP- DNMT3A1[1-427]. Gels show concentrations 175–2000 nM, experiment done in duplicate. (B) EMSA comparing binding of DNMT3A1[1-427] to nucleosomes with or without tri-methylation at H3 Lys27 (H3Kc27me3). Limiting amounts (2.3 nM) of nucleosomes were incubated with increasing concentrations (0–8000 nM) of His-MBP-DNMT3A1 constructs. Complexes were resolved by native-PAGE and imaged using blue light excitation and 532 nm emission filters. Gels show concentrations 62–8000 nM for clarity, experiment done in duplicate. (C) EMSA comparing binding of DNMT3A1[1-427] to unmodified, H2AKc119ub, H3K36me2 and dual H3K36me2/H2AK119ub nucleosomes wrapped with 5′ FAM labelled 175 bp Widom601 DNA. Nucleosomes (2.3 nM) were incubated with His-MBP-DNMT3A1[1-427] (0–4000 nM, 1.5x dilution series). Representative gels show concentrations 104–4000 nM for clarity. EMSA of unmodified, H2AKc119ub also appears in Fig. 1D. (D) Quantification of 5 C with full concentration series, from quadruplicate experiments. Source data are available online for this figure.

both H3 di-methylation marks bound with similar affinity to just H3K36me2 marked nucleosomes (Fig. 5A; Appendix Fig. S6D), suggesting H3K27me2 does not inhibit PWWP mediated interaction. However, solely modified H3 Lys-27 methylated nucleosomes were unable to promote interaction (Fig. 5B; Appendix Fig. S6E). This is in line with previous findings (Gu et al, 2022; Weinberg et al, 2019; Weinberg et al, 2021), driving home the importance of H2AK119ub in recruitment of DNMT3A1 to Polycomb-marked regions.

While H2AK119ub is found enriched at Polycomb silenced promoters, the modification is also highly abundant, widespread at low levels throughout the genome (Conway et al, 2021; Fursova et al, 2019; Fursova et al, 2021), overlapping with H3K36me2 (Weinberg et al, 2021). Doubly-modified H2AK119ub/H3K36me2 nucleosomes have a marginal additive effect on binding to DNMT3A1[1-427] suggesting that ubiquitin interaction, nucleosome surface interaction and PWWP H3K36me2 interaction can all occur concurrently on a single nucleosome (Fig. 5C,D). Taken as a whole, this points to PRC2 mediated H3K27 methylation neither directly inhibiting nor stimulating DNMT3A1 interaction, but H2AK119ub promoting interaction both in isolation and in combination with other histone modifications.

## H2AK119ub does not stimulate DNMT3A1 catalytic activity on nucleosomes

Recruitment and DNA methylation have been previously shown to go hand-in-hand (Fu et al, 2020; Weinberg et al, 2019; Xu et al, 2020b). To test what influence histone modifications had on the methyltransferase activity of DNMT3A1, we assayed turnover of the methyl donor cofactor S-adenosylmethionine (SAM) by full-length DNMT3A1-DNMT3L using the strong positioning Widom-601 DNA sequence flanked by a single 50 bp linker region as a substrate (Figs. 6A and EV4A–C). At high concentrations of DNA, activity was observed (Fig. EV4D), but substantially stimulated when the same sequence was used to wrap nucleosomes (Fig. 6A). This effect is likely due to release of the normally autoinhibitory domain of the ADD when bound to an unmodified H3 tail, observed by addition of H3 peptide in trans (Fig. EV4E (Guo et al, 2015; Li et al, 2011; Zhang et al, 2010)). Indeed, nucleosomes lacking the first 24 residues of H3 in the normally flexible tail (termed 'H3 tailless') showed low activity (Fig. 6A). As expected, methylation of H3 Lys-36 in nucleosome substrate stimulated the activity of DNMT3A1-DNMT3L.

Given the increased DNA methylation activity of H3K36me2 nucleosomes compared to unmodified nucleosomes (Fig. 6A) we presumed that the tighter binding H2AK119ub nucleosomes would

lead to even greater enzymatic activity. However, in vivo H2AK119ub-mark enriched regions are typically hypomethylated (Brinkman et al, 2012; Fu et al, 2020; Weinberg et al, 2019). Full-length DNMT3A1-DNMT3L copied the binding patterns observed for NT-PWWP alone DNMT3A1[1-427] fragment, with H3K36me2 and H2AK119ub nucleosomes bound preferentially over unmodified and acidic patch mutant (Fig. EV3C). Surprisingly, activity on H2AK119ub nucleosomes was similar to unmodified nucleosomes despite the significantly higher binding (Fig. 6A). This disconnect between affinity and catalytic activity is further exemplified when using acidic patch mutated nucleosomes, which showed equivalent activity to unmodified nucleosomes. This was also observed for nucleosomes wrapped with different strong positioning and linker DNA sequences (Fig. EV4F–H) suggesting this was not due to the DNA construct used. We purified DNMT3A2-DNMT3L complex (Fig. EV4I) to test the role of the N-terminal region of DNMT3A in nucleosome methylation assays, absent in this isoform. DNMT3A2 also showed a similar pattern of methyltransferase activity to DNMT3A1 on the different nucleosome substrates, with higher activity on H3K36me2 nucleosome and comparable activity on H2AK119ub nucleosome to their unmodified counterpart (Fig. 6B). In addition, a complex of DNMT3A1 with the catalytic domain of DNMT3B3 as accessory protein (DNMT3B3[534-770]-StrepII, Fig. EV5A) again showed a similar pattern as DNMT3A1-DNMT3L with higher activity on H3K36me2 nucleosomes and similar activity of H2AKc119ub to unmodified nucleosomes (Fig. 6C). As mentioned previously, DNMT3B3 directly binds to the nucleosome acidic patch and mutations in this region did impair activity of DNMT3A1-DNMT3B3, consistent with previous reports (Xu et al, 2020a). This suggests that under these conditions increased recruitment of DNMT3A1 does not directly result in a stimulation of downstream methylation activity.

The stimulatory activity of H3K36me2 is partly due to increased availability of DNA, caused by the H3 tail-DNA interaction being disrupted (Brohm et al, 2022), rather than direct DNMT3A1 binding. To test the relative importance of tail availability and recruitment for full-length DNMT3A1 in our assay, we made nucleosomes with different methylation states on the H3 N-terminal tail. We observed that H3Kc27me3, the DNMT3A1 non-binding methylation mark, does not stimulate catalytic activity (Fig. EV5B,C), so this is not a generic feature of H3 tail methylation. Nucleosomes containing lysine methyl analogues created by cysteine alkylation (Simon et al, 2007) bind less well to DNMT3A1[1-427] than closer-to-native chemically ligated nucleosomes (Fig. EV5D), as has been described for other methyl-lysine interactors due to differences in C-S bond length and angle (Fig. EV5E, right) (Seeliger et al, 2012). While this slight difference in chemical structure of alkylated histone affects methyl-specific reading it is unlikely to affect DNA-histone interactions. In the

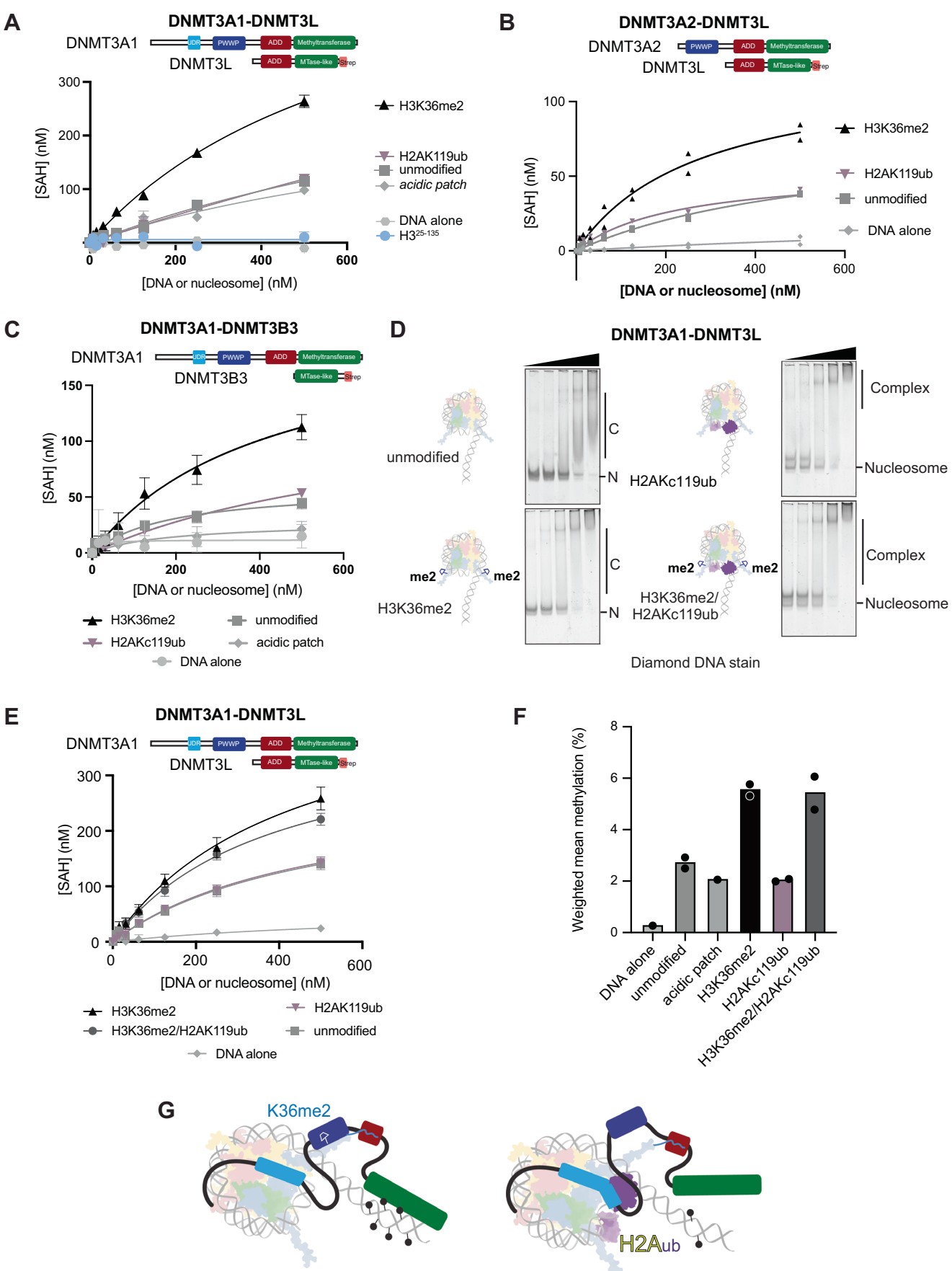

**Figure 6. DNMT3A1 shows ubiquitin specificity and DNMT3A1 recruitment and catalytic activity are disconnected.**

(A) Methyltransferase activity of full-length DNMT3A1-DNMT3L-StrepII on unmodified, H3K36me2, H2AKc119ub, acidic patch (H2A$^{E61A/E91A/E92A}$ and H2B$^{E105A}$), H3tailles (H3$^{A25C-136}$) wrapped with 195 bp Widom601 DNA and free 195 bp Widom601 DNA. DNMT3A1-DNMT3L was incubated with increasing concentrations of nucleosome/ DNA substrate for 1 h at 37 °C. Methyltransferase activity was detected using Promega MTase-Glo™ Methyltransferase Assay. Combined two experiments done in duplicate. Michaelis–Menten curves were fit using GraphPad Prism 10 error bars show $+/-$ standard error of mean. (B) Methyltransferase activity of DNMT3A2-DNMT3L-StrepII on unmodified, H3K36me2, H2AKc119ub nucleosomes wrapped with 195 bp Widom601 DNA and free 195 bp DNA Widom601 DNA. Methyltransferase activity was detected using Promega MTase-Glo™ Methyltransferase Assay. Experiment performed in duplicate, with all points shown. Michaelis–Menten curves were fit using GraphPad Prism 10. (C) Methyltransferase activity of DNMT3A1-DNMT3B3-StrepII on unmodified, H3K36me2, H2AKc119ub, acidic patch (H2A$^{E61A/E91A/E92A}$ and H2B$^{E105A}$) nucleosomes wrapped with 195 bp Widom601 DNA and free 195 bp DNA Widom601 DNA. Methyltransferase activity was detected using Promega MTase-Glo™ Methyltransferase Assay. Combined two experiments done in duplicate. Michaelis–Menten curves were fit using GraphPad Prism 10 error bars show $+/-$ standard error of mean. (D) EMSA comparing binding of full-length DNMT3A1-DNMT3L-StrepII to 195bp-wrapped unmodified, H3K36me2, H2AKc119ub and dual H3K36me2/ H2AK119ub nucleosomes. Limiting amounts (8 nM) of nucleosomes or DNA were incubated with increasing concentrations (0–2666 nM, 2x dilution series) of full-length DNMT3A1-DNMT3L-StrepII. Complexes were resolved by native-PAGE, stained with diamond stain and imaged using blue light. Representative gels show concentrations 41–667 nM of one of two independent experiments. (E) Methyltransferase activity of full-length DNMT3A1-DNMT3L-StrepII on unmodified, H3K36me2, H2AKc119ub and dual H3K36me2/H2AK119ub nucleosomes wrapped with 195 bp Widom601 DNA and free 195 bp Widom601 DNA. DNMT3A1-DNMT3L was incubated with increasing concentrations of nucleosome/DNA substrate for 1 h at 37 °C. Methyltransferase activity was detected using Promega MTase-Glo™ Methyltransferase Assay. Combined two experiments done in duplicate. Michaelis–Menten curves were fit using GraphPad Prism 10 error bars show $+/-$ standard error of mean. (F) Barplot of weighted mean methylation for linker region CpGs derived from Nanopore sequencing of in vitro methyltransferase assays of (A, E). Bar heights are the mean of two independent methyltransferase experiments in duplicate for unmodified, H3K36me2, H2AKc119ub and H3K36me2/H2AKcK119ub nucleosomes, one duplicate was sequenced for acidic patch mutated nucleosomes and DNA alone. (G) A model of proposed DNMT3A1 function at intergenic H3K36me2 sites (left), and facultative heterochromatin/bivalent domains (right). On intergenic chromatin, the interaction of the PWWP and ADD domains with H3K36me2 and unmodified H3K4 respectively, stimulate methylation activity leading to high DNA methylation levels. On facultative chromatin, H2AK119ub leads to recruitment of DNMT3A1 through the UDR motif. This motif is also responsible for interacting with the acidic patch on nucleosomes, which is interrupted by disease mutations implicated in cancer. Lower methyltransferase activity stimulation by H2AK119ub leads to lower levels of methylation that may be increased due to increased recruitment, exacerbated by disease mutations inhibiting PWWP-H3K36me2 interaction. Source data are available online for this figure.

methyltransferase assay, this reduced direct binding to the PWWP of alkylated histones H3Kc36me2, H3Kc36me3, H3Kc27me3Kc36me3 also reduces catalytic activity compared to more native H3K36me2 (Fig. EV5E). In addition, performing the assay on DNA alone in the presence of H3K36me2 peptides showed no direct stimulation by methylation compared to unmodified peptides (Fig. EV5F). This suggest PWWP binding and H3 tail availability both contribute to the stimulatory effect of H3K36me2.

We next tested if H2AK119ub could inhibit DNMT3A activity, potentially through sequestering DNMT3A1 on the nucleosome in a poorly active conformation. DNMT3A1 showed similar affinity for dual modified H3K36me2/H2AKc119ub nucleosomes as to H2AK119ub in binding assays (Figs. 5C and 6D). However, in methyltransferase assays DNMT3A1-DNMT3L showed an increase in activity on H3K36me2/H2AKc119ub nucleosomes compared to H2AK119ub alone (Fig. 6E), suggesting that binding of DNMT3A1 to H2AK119ub still allows H3K36me2 stimulation. This agrees well with the co-occurrence of these marks and high CpG methylation in intragenic regions (Weinberg et al, 2019). To test whether the methylation was being incorporated in DNA we used nanopore sequencing of the reactions and observed that the extent of methylation of linker CpG sites agreed with the measured activities (Fig. 6F).

Overall, DNMT3A1-DNMT3L activity is stimulated by H3K36me2, but not by H2AK119ub. Indeed, DNMT3A1 is the predominant isoform present in somatic tissues while H2AK119ub sites are commonly hypomethylated (Brinkman et al, 2012; Fu et al, 2020; Weinberg et al, 2019), suggesting the lower enzymatic activity may underlie the otherwise unexplained observation of H2AK119ub hypomethylation.

## Discussion

Here we have shown that the DNMT3A1 UDR found in the divergent N-terminal region (Gretarsson et al, 2024; Gu et al, 2022;

Heyn et al, 2019; Weinberg et al, 2021) multivalently engages with nucleosomes. The same UDR region also improves interaction within the context of PWWP-mediated H3K36me2 interaction, by adding stabilising nucleosome surface recognition (Fig. 6G).

The striking absence of missense mutations in the UDR region in the healthy population (Fig. 1A) and presence of disease mutations (Basturk et al, 2017; Dutton-Regester et al, 2013; Huang et al, 2022; Jaiswal et al, 2014; Tatton-Brown et al, 2018) suggests this region is essential for proper DNMT3A1 function. Indeed, DNMT3A1 is the predominant isoform of DNA methyltransferase in the postnatal brain required for neural development (Gu et al, 2022; Hamagami et al, 2023; Sendzikaite et al, 2019), which undergoes DNA methylation reconfiguration during postnatal neuronal development (Lister et al, 2013).

The UDR interacts extensively with the nucleosome surface. Interestingly, the spumavirus GAG protein binds to nucleosomes (Lesbats et al, 2017) with a similar overall interaction interface and fold despite a different amino acid sequence (RMSD 1.25 Å), evidently an example of convergent evolution. An acidic patch interaction was also observed in the structure of DNMT3A2-DNMT3B3 (Xu et al, 2020a). A splice-isoform-specific feature of the DNMT3B3 catalytic domain contacts the acidic patch, rigidly orientating the rest of the methyltransferase complex on adjacent DNA (Xu et al, 2020a). In contrast, we see low ordering for the DNMT3A1-DNMT3L methyltransferase domains on adjacent DNA suggesting much greater degrees of freedom, presumably due to the flexible sequences between UDR and rest of DNMT3A1-DNMT3L. Intriguingly, mutations blocking acidic patch binding in DNMT3B3 lead to increased reliance on histone tail modifications for methylation targeting (Xu et al, 2020a) and we see lower activity of a DNMT3A1-DNMT3B3 complex on acidic patch mutated nucleosomes. How DNMT3A1 and DNMT3B3 co-ordinate their nucleosome and chromatin binding features would be a fascinating avenue of future study.

DNMT3A1 is highly specific to H2AK119ub within a nucleosome. H2AK119ub is the most prevalent histone ubiquitin mark (Fursova et al, 2019; Lee et al, 2015) and multiple readers of this mark have been recently structurally described. This includes two subunits of PRC2 (Blackledge et al, 2014; Kasinath et al, 2021), RYBP subunit of PRC1 (Ciapponi et al, 2024) and SSX1 (McBride et al, 2020; Tong et al, 2024). All readers of H2AK119ub described structurally to date share common features and likely would compete for the same interfaces but display highly variable binding modes. Indeed, PRC2 component JARID2 and DNMT3A1 UDR bind mutually exclusively (Chen et al, 2024). Unfortunately, the flexibility in our system prevented us from visualising the nature of the ubiquitin-DNMT3A1 interaction, but the UDR region contains a stretch of positively charged amino acids (Fig. 2A), which affects recruitment to H2AK119ub when mutated (Gu et al, 2022). These residues likely contact DNA as part of the ubiquitin interaction, but based on our biochemistry this is not via linker DNA (Appendix Fig. S5D) but due to DNA interaction at the dyad (Chen et al, 2024; Gretarsson et al, 2024). Future work will determine the balance of affinities between these different readers and how they may integrate in a cellular setting.

While this work was under review two complementary studies have been published (Chen et al, 2024; Gretarsson et al, 2024). These reported similar structures of the UDR on H2AK119ub-nucleosomes, using minimal UDR fragments of DNMT3A1 and glutaraldehyde cross-linking. The structure reported here used the full-length DNMT3A1-DNMT3L, but the majority of the complex was not readily observable due to structural flexibility. We were unable to obtain structures using smaller fragments in the absence of cross-linker, suggesting that the extra DNA and histone stabilisation conferred by the remainder of the DNMT3A1-DNMT3L complex was required to aid our complex stability on grids. These studies further validated structures in cells highlighting the role of the UDR region in recruitment and activity in vivo (Chen et al, 2024; Gretarsson et al, 2024).

Intriguingly, the affinity for H2AK119ub nucleosomes appears higher than for H3K36me2 nucleosomes. This agrees with the prior observation that DNMT3A1 localisation correlates with H2AK119ub-rich CpG island shores (Manzo et al, 2017). However, Polycomb-mark enriched regions are typically found to be hypomethylated (Fu et al, 2020). This hypomethylation pattern though is less clear for more differentiated cells where DNMT3A1 expression is higher (Di Croce et al, 2002; Mohn et al, 2008; Schlesinger et al, 2007; Weinberg et al, 2019). It should be noted that genomic abundance of H3K36me2 is higher but spread more evenly across the genome, as such may work to outcompete the sparser H2AK119ub mark (Weinberg et al, 2019) and H2AK119ub is also spread widely across the genome beyond Polycomb-enriched peaks (Conway et al, 2021; Fursova et al, 2019; Fursova et al, 2021). Disease mutations in the PWWP domain responsible for H3K36me2 binding leads to redistribution of DNMT3A1 predominantly to H3K27me3/H2AK119ub enriched regions, which is mediated via DNMT3A1's UDR-H2AK119ub recognition (Gretarsson et al, 2024).

Surprisingly in vitro DNA methylation of H2AK119ub nucleosomes is similar to that on unmodified nucleosomes, in contrast to the higher activity seen on H3K36me2 marked nucleosomes. Our structure shows DNMT3A1-DNMT3L density concurrent with linker DNA but is too low resolution in this region to observe if the enzyme is in a non- or poorly productive state. Indeed, H2AK119ub-reading by DNMT3A1 does not inhibit catalytic activity in doubly marked H3K36me2/H2AK119ub nucleosomes (Fig. 6E). Demethylating TET enzymes actively remove methylation at H2AK119ub marked sites leading them to be hypomethylated (Gu et al, 2018; Manzo et al, 2017; Neri et al, 2013), but this may also be in part due to a disconnect between recruitment and enzymatic activity described here. Hypermethylation of Polycomb-marked CpG islands is a hallmark of HESJAS syndrome (Heyn et al, 2019; Kibe et al, 2021; Sendzikaite et al, 2019; Weinberg et al, 2021) as well as observed in cancers and during normal ageing (Gretarsson et al, 2024; Hannum et al, 2013; Teschendorff et al, 2010). While DNMT3A1 may be less active at H2AK119ub marked chromatin, pathogenic mutant-driven aberrant hyperaccumulation or longer opportunity for overall residence will conceivably lead to hypermethylation seen at CpG-islands in disease states.

# Methods

## Methods and protocols

### Generation of plasmid constructs
A list of expression constructs used in this study can be found in Appendix Table S3. Human DNMT3A1, DNMT3L and DNMT3B3 constructs were PCR amplified from cDNA expression constructs (Addgene #35521, #35523) and cloned by ligation dependant cloning into x6His-MBP-TEV or 6xHis-MBP-GFP expression vectors. DNMT3L was cloned with a StrepII tag on the C terminus. For DNMT3B3, a 20 amino acid linker was added between the C terminus and the StrepII tag (GTENSKGLEVLFNGPSGSSVWSHPQFEK). Histone expression vectors have been described previously (Burdett et al, 2023; Salguero et al, 2019; Wilson et al, 2019), purchased from Addgene originated by the Landry lab. All histones are derived from human sequences, throughout all H3 constructs are based on H3.1 containing C96S C110A mutation.

Template DNA for 195 bp DNA was synthesised as a double-strand gBlock fragments (Integrated DNA Technologies), prior to Gibson assembly into a pUC57 backbone.

DNMT3A and histone mutations were mutated either using site directed mutagenesis or direct cloning of synthesised double-strand gBlock fragments containing mutations (Integrated DNA Technologies) using Gibson assembly (Gibson et al, 2009).

## Protein purification

### Histone expression and purification
Histones were expressed and purified as previously described (Burdett et al, 2023; Deak et al, 2023; Wilson et al, 2019). Briefly, histones were expressed in BL21 (DE3 RIL) cells and resolubilised from inclusion bodies. Histones were further purified by cation exchange chromatography prior to dialysis in 1 mM acetic acid and lyophilisation.

Histone concentrations were determined via absorbance at 280 nm using a Nanodrop One spectrophotometer (Thermo Scientific), followed by SDS-PAGE and InstantBlue (Expedeon) or Coomassie colloidal blue staining with comparison to known amounts of control proteins.

**Reagents and tools table**

| Reagent/Resource | Reference or Source | Identifier or Catalog Number |
|---|---|---|
| **Recombinant DNA** | | |
| pLIC-His6-MBP-TEV | Addgene | pMDW89 |
| **DNMT3A1** | | |
| pLIC-His-MBP-DNMT3A1-427 | This study | pMDW208 |
| pLIC-His-MBP-DNMT3A1-277 | This study | pMDW238 |
| His-MBP-DNMT3A1-427 R181C | This study | pMDW539 |
| His-MBP-DNMT3A1-427 R183W | This study | pMDW551 |
| His-MBP-DNMT3A1-427 A192E | This study | pMDW541 |
| His-MBP-DNMT3A1-427 W330R | This study | pMDW225 |
| His-MBP-DNMT3A1-427 D333N | This study | pMDW226 |
| His-MBP-DNMT3A1-427 P195A | This study | pMDW723 |
| His-MBP-DNMT3A1-427 Y197A | This study | pMDW708 |
| His-MBP-DNMT3A1-427 I198A | This study | pMDW724 |
| His-MBP-DNMT3A1-427 K200A | This study | pMDW678 |
| His-MBP-DNMT3A1-427 R201A | This study | pMDW709 |
| His-MBP-DNMT3A1-427 K202A | This study | pMDW726 |
| His-MBP-DNMT3A1-427 R203A | This study | pMDW727 |
| His-MBP-DNMT3A1-427 D204A | This study | pMDW728 |
| His-MBP-DNMT3A1-427 E205A | This study | pMDW729 |
| His-MBP-DNMT3A1-427 L207A | This study | pMDW710 |
| His-MBP-DNMT3A58-427 | This study | pMDW249 |
| His-MBP-DNMT3A115-427 | This study | pMDW250 |
| His-MBP-DNMT3A142-427 | This study | pMDW209 |
| His-MBP-DNMT3A192-427 | This study | pMDW210 |
| His-MBP-DNMTA164-219 | This study | pMDW486 |
| His-MBP-DNMT3A1-427 Δ142-178 | This study | pMDW395 |
| His-MBP-DNMT3A1-427 Δ165-174 | This study | pMDW390 |
| His-MBP-DNMT3A1-427 Δ184-191 | This study | pMDW396 |
| His-MBP-DNMT3A1 | This study | pMDW271 |
| **DNMT3A2** | | |
| His-MBP-DNMT3A2 | This study | pMDW272 |
| His-MBP-DNMT3A21-238 | This study | pMDW243 |
| **DNMT3L** | | |
| His-GFP-DNMT3L-StrepII | This study | pMDW285 |
| **DNMT3B3** | | |
| His-GFP-DNMT3B3 534-770-StrepII | This study | pMDW814 |
| **Histones** | | |
| unmodified H3.1 (No cys) | (Wilson et al, 2019) | pMDW31 |
| H3.3 T45CΔ1-44 | (Bryan et al, 2021) | pMDW401 |
| H3.1 K36C | This study | pMDW40 |
| H3.1 K27C | This study | pMDW146 |
| H2A | Addgene | pMDW1 |
| H2A acidic patch | (Belotserkovskaya et al, 2020) | pMDW8 |
| H2B | Addgene | pMDW17 |

| Reagent/Resource | Reference or Source | Identifier or Catalog Number |
|---|---|---|
| H2B acidic patch | (Belotserkovskaya et al, 2020) | pMDW20 |
| H4 | Addgene | pMDW45 |
| H2A K13C | (Burdett et al, 2023) | pMDW236 |
| H2A K15C | (Belotserkovskaya et al, 2020) | pMDW10 |
| H2A K119C | (Burdett et al, 2023) | pMDW13 |
| H1A K127C | (Burdett et al, 2023) | pMDW12 |
| H2B K120C | (Burdett et al, 2023) | pMDW368 |
| H3 K18C | (Burdett et al, 2023) | pMDW35 |
| H2AK119C acidic patch | This study | pMDW537 |
| H3tailless | (Belotserkovskaya et al, 2020) | pMDW34 |
| H3K27CK36C | This study | pMDW147 |
| **Ubiquitin** | | |
| His-TEV-Ubiquitin G76C | (Burdett et al, 2023) | pMDW56 |
| His-TEV-Ubiquitin I44A | This study | pMDW202 |
| **DNA** | | |
| Widom601 145 or 175 bp DNA | (Burdett et al, 2023) | pMDW290 |
| 195 bp DNA | This study | pMDW327 |
| 193 bp DNA | This study | pMDW306 |
| 207 bp DNA | (Bryan et al, 2021) | pMDW380 |
| **Antibodies** | | |
| Anti-H2B | Abcam ab1790 | |
| Anti-H3 | Abcam ab1791 | |
| Anti-MBP | NEB e8032s | |
| Anti-H2A | Abcam ab18255 | |
| **Oligonucleotides and other sequence-based reagents** | | |
| H3K27C fw | CTGGCCACCAAGGCGGCTCGCtgcAGCGCTCCGGCCACCGGTGGC | MDW200 |
| H3K27C rev | GCCACCGGTGGCCGGAGCGCTgcaGCGAGCCGCCTTGGTGGCCAG | MDW201 |
| H3 K36C fw | CTCCGGCCACCGGTGGCGTCtgcAAGCCCCACCGCTACCGCCc | MDW87 |
| H3 K36C rev | gGGCGGTAGCGGTGGGGCTTgcaGACGCCACCGGTGGCCGGAG | MDW88 |
| DNMT3A1-277fw | TACTTCCAATCCAATGCAATGCCCGCCATGCCCTCC | HW9 |
| DNMT3A1-277rev | TTATCCACTTCCAATGTTATTAtcaGCCTGCTTTGGTGGCATTCTTG | HW10 |
| DNMT3A1-427fw | TACTTCCAATCCAATGCAATGCCCGCCATGCCCTCCAG | MDW258 |
| DNMT3A1-427rev | TTATCCACTTCCAATGTTAtcaTTCTTCTGGTGGCTCCAGGC | MDW262 |
| DNMT3A1-427 D194A fw | GCTCACCTTCCAGGCGGGGGCCCCCTACTACATCAGCAAGCGCAA | HW49 |
| DNMT3A1-427 D194A rev | TTGCGCTTGCTGATGTAGTAGGGGGCCCCCGCCTGGAAGGTGAGC | HW50 |
| Y196A | CTTCCAGGCGGGGGACCCCGCCTACATCAGCAAGCGCAAGCGG | HW51 |
| Y196A | CCGCTTGCGCTTGCTGATGTAGGCGGGGTCCCCCGCCTGGAAG | HW52 |
| Y197A | CAGGCGGGGGACCCCTACGCCATCAGCAAGCGCAAGCGGG | HW53 |
| Y197A | CCCGCTTGCGCTTGCTGATGGCGTAGGGGTCCCCCGCCTG | HW54 |
| I198A | GGCGGGGGACCCCTACTACGCCAGCAAGCGCAAGCGGGAC | HW55 |
| I198A | GTCCCGCTTGCGCTTGCTGGCGTAGTAGGGGTCCCCCGCC | HW56 |
| K200A | GGCGGGGGACCCCTACTACATCAGCGCGCGCAAGCGGGACGAGTGGC | HW57 |
| K200A | GCCACTCGTCCCGCTTGCGCGCGCTGATGTAGTAGGGGTCCCCCGCC | HW58 |
| R201A | GGGGACCCCTACTACATCAGCAAGGCCAAGCGGGACGAGTGGCTG | HW59 |

| Reagent/Resource | Reference or Source | Identifier or Catalog Number |
|---|---|---|
| R201A | CAGCCACTCGTCCCGCTTGGCCTTGCTGATGTAGTAGGGGTCCCC | HW60 |
| R203A | CCCCTACTACATCAGCAAGCGCAAGGCGGACGAGTGGCTGGCACGCT | HW61 |
| R203A | AGCGTGCCAGCCACTCGTCCGCCTTGCGCTTGCTGATGTAGTAGGGG | HW62 |
| E205A | TCAGCAAGGCCAAGCGGGACGCGTGGCTGGCACGCTGGAAAAG | HW65 |
| E205A | CTTTTCCAGCGTGCCAGCCACGCGTCCCGCTTGGCCTTGCTGA | HW66 |
| L207A | CGCAAGCGGGACGAGTGGGCGGCACGCTGGAAAAGGGAGGC | HW67 |
| L207A | GCCTCCCTTTTCCAGCGTGCCGCCCACTCGTCCCGCTTGCG | HW68 |
| R209A | GCGGGACGAGTGGCTGGCAGCCTGGAAAAGGGAGGCTGAGAAGAAAGC | HW69 |
| R209A | GCTTTCTTCTCAGCCTCCCTTTTCCAGGCTGCCAGCCACTCGTCCCGC | HW70 |
| R212A | GTGGCTGGCACGCTGGAAAGCGGAGGCTGAGAAGAAAGCCAAGGTC | HW71 |
| R212A | GACCTTGGCTTTCTTCTCAGCCTCCGCTTTCCAGCGTGCCAGCCAC | HW72 |
| R216A | CGCTGGAAAAGGGAGGCTGAGGCGAAAGCCAAGGTCATTGCAGGAATGA | HW73 |
| R216A | TCATTCCTGCAATGACCTTGGCTTTCGCCTCAGCCTCCCTTTTCCAGCG | HW74 |
| 3A delta 196-219 fw | CCGAGGCTCACCTTCCAGGCGGGGGACCCCGTCATTGCAGGAATGAATGCTGTGG | HW75 |
| 3A delta 196-219 rev | TTCTTCCACAGCATTCATTCCTGCAATGACGGGGTCCCCCGCCTGGAA | HW76 |
| delta 196-207 fw | CCGAGGCTCACCTTCCAGGCGGGGGACCCCGCACGCTGGAAAAGGGAGGC | HW77 |
| delta 196-207 rev | TTTCTTCTCAGCCTCCCTTTTCCAGCGTGCGGGGTCCCCCGCCTGGAA | HW78 |
| delta 208-219 fw | ATCAGCAAGCGCAAGCGGGACGAGTGGCTGGTCATTGCAGGAATGAATGCTGTGG | HW79 |
| delta 208-219 rev | TTCTTCCACAGCATTCATTCCTGCAATGACCAGCCACTCGTCCCGCTT | HW80 |
| DNMT3A1-427 P195A fw | CACCTTCCAGGCGGGGGACGCGTACTACATCAGCAAGCGCAAGCG | HW84 |
| DNMT3A1-427 P195A rev | CGCTTGCGCTTGCTGATGTAGTACGCGTCCCCCGCCTGGAAGGTG | HW85 |
| DNMT3A1-427 S199A Fw | GCGGGGGACCCCTACTACATCGCGAAGCGCAAGCGGGACGAG | HW86 |
| DNMT3A1-427 S199A rev | CTCGTCCCGCTTGCGCTTCGCGATGTAGTAGGGGTCCCCCGC | HW87 |
| DNMT3A1-427 K202A Fw | CCCCTACTACATCAGCAAGCGCGCGCGGGACGAGTGGCTGGC | HW88 |
| DNMT3A1-427 K202 rev | GCCAGCCACTCGTCCCGCGCGCGCTTGCTGATGTAGTAGGGG | HW89 |
| DNMT3A1 1-427 D204A fw | ACATCAGCAAGCGCAAGCGGGCCGAGTGGCTGGCACGCTGGA | HW90 |
| DNMT3A1 1-427 D204A rev | TCCAGCGTGCCAGCCACTCGGCCCGCTTGCGCTTGCTGATGT | HW91 |
| **Chemicals, Enzymes and other reagents** | | |
| MTase-Glo™ Methyltransferase Assay | Promega | V7601 |
| NEB Blunt/TA Ligase Master Mix | NEB | #M0367 |
| Native Barcoding Kit 24 V14, | Oxford Nanopore Technologies | #SQK-NBD114.24 |
| NEBNext FFPE Repair Mix | NEB | #M6630 |
| NEBNext Ultra II End repair/dA-tailing Module (NEB, #E7546) | NEB | #E7546 |
| NEBNext Quick Ligation Module | NEB | #E6056 |
| **Software** | | |
| Cryosparc v4.2.0 | | |
| Relion 4 | | |
| GraphPad Prism v10 | | |
| **Instrumentation** | | |
| Oxford Nanopore MinION Mk1b | R10.4.1 flongle flow cell | |
| TitanKrios G3 | Thermofisher scientific | |
| Orbitrap Fusion Lumos | Thermo Fisher Scientific | |

### Expression and purification of DNMT3 constructs

Full-length His-MBP-DNMT3A1 or DNMT3A2 and His-GFP-DNMT3L-StrepII or DNMT3B3[534-770]-StrepII were co-expressed from two separate plasmids in BL21 (DE3 RIL) in 2xTY medium. Cell pellets were resuspended in lysis buffer (25 mM sodium phosphate pH 7.5, 400 mM NaCl, 0.1% (v/v) Triton, 10% (v/v) glycerol, 2 mM β-mercaptoethanol, 1 mM AEBSF, 1X protease inhibitor cocktail (2.2 mM PMSF, 2 mM benzamidine HCl, 2 μM leupeptin, 1 μg.mL$^{-1}$ pepstatin A), 4 mM MgCl$_2$, 5 μg.mL$^{-1}$ DNAse and 500 μg.mL$^{-1}$ lysozyme) and lysed using a Constant Systems 1.1 kW TS Cell Disruptor. Cell debris was spun down at 39,000 × g for 25 min and the supernatant was filtered through a 0.45 μm filter. Lysate was then loaded on a nickel sulphate charged HiTrap chelating column (Cytiva) equilibrated with 20 mM Tris pH 7.5, 400 mM NaCl, 15 mM imidazole, 10% (v/v) glycerol, 2 mM β-mercaptoethanol, 0.5 mM AEBSF, washed 20 CV with the same buffer and eluted in 12 CV 20 mM Tris pH 7.5, 400 mM NaCl, 400 mM imidazole, 10% (v/v) glycerol, 2 mM β-mercaptoethanol, 0.5 mM AEBSF. This was loaded on a StrepTrap HP column (Cytiva) equilibrated in streptrap buffer (20 mM HEPEs pH 7.5, 150 mM NaCl, 4 mM sodium citrate, 10% (v/v) glycerol, 2 mM DTT, 0.5 mM AEBSF), washed with 15CV streptrap buffer and eluted in 2.5 mM D-desthiobiotin in streptrap buffer. The His-MBP and His-GFP tags were cleaved using TEV protease (1:25 ratio) at 4 °C overnight. Cleaved proteins were loaded on a Heparin column (Cytiva) equilibrated in 20 mM HEPES pH 7.5, 150 mM NaCl, 1 mM DTT, 5% (v/v) glycerol, 0.5 mM AEBSF, washed with 15 CV buffer and eluted in 15–80% 150 mM–1 M NaCl, 20 mM HEPES pH 7.5, 1 mM DTT, 5% (v/v) glycerol, 0.5 mM AEBSF. DNMT3A1-DNMT3L-StrepII eluted at 32.5% (426 mM NaCl). The pure protein complex was concentrated using a spin concentrator and buffer exchanged to 150 mM NaCl. Purity was analysed by SDS-PAGE. An analytic amount (20 μg) was loaded on a Superdex 200 Increase 3.2/300 (Cytiva), fractions were analysed by SDS-PAGE. The purified complex was flash frozen in liquid nitrogen and stored at −80 °C.

His-MBP tagged DNMT3A and DNMT3B non-full-length constructs were expressed in BL21 (DE3 RIL) E. coli cells with 400 μM IPTG at 18 °C overnight in 2xYT medium (16 g/l tryptone, 10 g/l yeast extract, 5 g/l NaCl, pH 7.0). Cell pellets were resuspended in in lysis buffer (25 mM sodium phosphate pH 7.5, 400 mM NaCl, 0.1% (v/v) Triton, 10% (v/v) glycerol, 2 mM β-mercaptoethanol, 1 mM AEBSF, 1X protease inhibitor cocktail (2.2 mM PMSF, 2 mM benzamidine HCl, 2 μM leupeptin, 1 μg/ml pepstatin A), 4 mM MgCl$_2$, 5 μg.mL$^{-1}$ DNAse and 500 μg/ml lysozyme) and stirred at 4 °C before additional lysis using a sonicator (2 s on, 2 s off for total 20 s at 50% amplitude. Cell debris was spun down at 39,000 × g for 25 min and the supernatant was filtered through a 0.45 μm filter. Lysate was then loaded on Ni-NTA beads (1 ml per litre culture), washed with 25 column volume (CV) 15 mM sodium phosphate, 500 mM NaCl, 10% glycerol, 15 mM imidazole, 2 mM β-mercaptoethanol and eluted with 5CV 20 mM Tris pH 7.5, 400 mM NaCl, 300 mM imidazole, 10% glycerol, 2 mM β-mercaptoethanol. The eluted protein was concentrated using a 30 kDa MWCO centrifugal filter and analysed for purity on SDS-PAGE. Where necessary, an ion exchange step was added. For this, the eluted fraction of the Nickel NTA was diluted to 100 mM NaCl using 20 mM HEPES pH 7.5, 10% glycerol, 1 mM DTT and loaded on 5 mL HiTrap Q HP cation exchange chromatography column

(Cytiva), washed with 15 CV 20 mM HEPES pH 7.5, 100 mM NaCl, 10% glycerol, 1 mM DTT and eluted with 0–50% 20 mM HEPES pH 7.5, 1 M NaCl, 10% glycerol 1 mM DTT. Fractions containing the correct protein were pooled. All proteins were further purified by size exclusion chromatography. Pooled fractions from Nickel NTA or ion exchange were concentrated to less than 5 ml and loaded on a HiLoad Superdex 200 16/600 (Cytiva) equilibrated with 15 mM HEPES pH 7.5, 150 mM NaCl, 1 mM DTT, 5% (v/v) glycerol. Fractions were analysed by SDS-PAGE and those containing pure protein were pooled, concentrated in a 30 kDa MWCO centrifugal filter, flash frozen in liquid nitrogen and stored at −80 °C. His-MBP-DNMT3A1[1-277] was cleaved with His-tagged TEV protease, purified using NiNTA affinity chromatography and buffer exchanged back to the original buffer. 100% of the starting amount (molar) of protein was recovered after TEV cleavage. For the NT-PWWP fragment, His-MBP-DNMT3A1[1-427] was cleaved with His-tagged TEV protease, purified using NiNTA affinity chromatography and buffer exchanged back to the original buffer. Only 26% of the starting amount (molar) of protein was recovered after TEV cleavage. To remove tag form PWWP alone construct, His-MBP-DNMT3A1 278-247 was cleaved with His-tagged TEV protease and purified using Q HP ion exchange (cytiva) and size-exclusion (HiLoad 16/600 Superdex 75 pg) chromatography.

For NMR experiments His-MBP-DNMTA1[164-219] was expressed and purified as above with additional steps to remove the tag. The His-MBP tag was cleaved using His-tagged Recombinant Tobacco Etch Virus (TEV). A ratio of 14:1 (w/w) DNMTA1[164-219]: TEV was added to the eluted fraction of the Nickel NTA purification and dialysed in 2 L 20 mM HEPES pH 7.5, 150 mM, 4 mM Sodium citrate, 10% (v/v) glycerol, 1 mM DTT, 0.5 mM AEBSF using 3.5 MWCO SnakeSkin™ Dialysis Tubing at 4 °C for 18 h. The cleaved mixture was spun for 10 min at 40,000 × g and supernatant was diluted to 100 mM NaCl with 20 mM HEPES pH 7.5, 10% (v/v) glycerol, 1 mM DTT. The mixture was loaded on a 5 mL HiTrap SP HP cation exchange chromatography column (Cytiva), washed with 10CV 20 mM HEPES pH 7.5, 100 mM NaCl, 10% (v/v) glycerol, 1 mM DTT, 0.5 mM AEBSF and eluted with a gradient of 0–80% 20 mM HEPES pH 7.5, 1 M NaCl, 10% (v/v) glycerol, 1 mM DTT, 0.5 mM AEBSF. The fraction containing DNMTA1[164-219] were pooled and purified further by size exclusion chromatography using a HiLoad Superdex 75 16/600 (Cytiva) in gel filtration buffer (15 mM HEPES pH 7.5, 150 mM NaCl, 1 mM DTT, 5% (v/v) glycerol). Fractions were analysed by SDS-PAGE and those containing pure protein were pooled and concentrated using a 1 mL HiTrap SP FF cation exchange chromatography column (Cytiva) as follows. The pooled fractions were diluted to 100 mM NaCl using 20 mM sodium phosphate pH 7.5, 5% glycerol, 1 mM DTT, washed with 5CV 20 mM sodium phosphate pH 7.5, 100 mM NaCl, 5% glycerol, 1 mM DTT and eluted with 60% 20 mM sodium phosphate pH 7.5, 1 M NaCl, 5% glycerol, 1 mM DTT. Fractions with the highest concentration protein were pooled and dialysed into 1 L 20 mM sodium phosphate pH 7.5, 150 mM NaCl, 5% glycerol, 1 mM DTT using 0.5–3 mL 3.5 MWCO Slide-A-Lyzer™ Dialysis Cassette (Thermo Fisher Scientific) at 4 °C for 4 h. The purified protein was flash frozen in liquid nitrogen and stored at −80 °C.

### Expression and purification of 6xHis-ubiquitinG76C

6xHis-ubiquitin G76C and 6xHis-ubiquitin I44A G76C were expressed and purified as previously described (Burdett et al,

2023). Briefly, ubiquitin proteins were expressed in BL21 (DE3 RIL) cells, lysed and purified using nickel sulphate charged HiTrap chelating column (Cytiva) and size exclusion chromatography (HiLoad Superdex S75 16/600 Cytiva) prior to dialysis in 1 mM acetic acid and lyophilisation.

## Histone chemical modification

### Native chemical ligation
Native chemical ligation was performed essentially as described (Bartke et al, 2010; Bryan et al, 2021), with some modifications. Peptides corresponding to human H3.1 residues 1–43 containing H3K36me2 and H3K27me2K36me2 with C-terminal thioesters were synthesised by Peptide Synthetics. H3 Δ1-44 T45C C110A histone was resuspended at 14 mg/ml in degassed 300 mM sodium phosphate, 6 M Guanidine, 100 mM TCEP (Tris(2-carboxyethyl) phosphine hydrochloride) pH 7.9. Peptide was resuspended in degassed 300 mM sodium phosphate, 6 M Guanidine, 120 mM MPAA (4-Mercaptophenylacetic acid) pH 7.9. Equal volume of the peptide and histones were mixed and incubated for 24 h at 25 °C with gentle agitation. The reaction was dialysed extensively into 7 M Urea, 25 mM Tris pH 7.5, 20 mM NaCl, 1 mM EDTA, 2 mM β-mercaptoethanol and reacted products separated from unreacted histones and peptide by cation exchange chromatography eluting using a salt gradient. Reaction product was confirmed by SDS-PAGE and 1D intact weight ESI mass spectrometry (SIRCAMs, School of Chemistry, University of Edinburgh).

### Cysteine alkylation
Histone H3K36C, H3K27C and H3K27CK36C were alkylated as previously described (Simon et al, 2007; Simon and Shokat, 2012). Briefly, histones were resuspended in denaturing buffer and (2-chloroethyl)-dimethylammonium chloride reagent was added and incubated at 20 °C for 2 h. The reaction was quenched with ~650 μM β-mercaptoethanol and desalted using PD-10 columns (GE Healthcare). The extent of reaction was checked using 1D intact weight ESI mass spectrometry (SIRCAMs, School of Chemistry, University of Edinburgh).

### Histone ubiquitylation
Histones were chemically ubiquitylated as previously described (Burdett et al, 2023; Long et al, 2014; Wilson et al, 2016). Briefly, lyophilized 6xHis-ubiquitinG76C or 6xHis-ubiquitinI44AG76C and histones H2AK119C, H2AK13C, H2AK15C, H2AK127C, H2BK120C or H3K18C were resuspended, mixed, added to a solution of 1,3-dibromoacetone (DBA) in 100 mM Tris pH 7.5 and incubated for 1 h on ice before quenching with 20 mM β-mercaptoethanol. The ubiquitylated histones were purified by ion exchange chromatography (HiTrap SP HP column Cytiva) followed by nickel sulphate charged HiTrap chelating column (Cytiva) and dialysed in 1 mM β-mercaptoethanol, then lyophilized and stored at −20 °C.

## Nucleosome formation

### Octamer refolding
Octamers were refolded as previously described (Dyer et al, 2004; Wilson et al, 2016). Briefly, histones were resuspended in 20 mM Tris pH 7.5, 6 M guanidine, 10 mM DTT and mixed in a ratio of 1:1:1.5:1.5 H3, H4, H2A, H2B and diluted to a total concentration of 2 mg/ml. The histone mixture was dialysed to 15 mM Tris pH 7.5, 2 M NaCl, 5 mM β-mercaptoethanol, 1 mM EDTA. In case of His-tagged histones EDTA was omitted from this buffer. After dialysis, his-tagged octamers were purified on a 1 mL nickel sulphate charged HiTrap chelating column (Cytiva) and eluted in 15 mM Tris pH 7.5, 2 M NaCl, 5 mM β-mercaptoethanol, 300 mM imidazole. Then, 1 mM EDTA and 1/25 (w/w) TEV protease was added and dialysed into 15 mM Tris pH 7.5, 2 M NaCl, 1 mM EDTA, 5 mM β-mercaptoethanol for 18 h to cleave the his-tag. All octamers were purified using size exclusion chromatography (HiLoad Superdex 200 16/600 or Superdex 200 Increase 10/300 GL Cytiva) in 15 mM Tris pH 7.5, 2 M NaCl, 1 mM EDTA, 5 mM β-mercaptoethanol. Octamer fractions were pooled, concentrated and stored in 50% (v/v) glycerol at −20 °C.

### PCR amplification of nucleosome DNA
All DNA fragments for nucleosome reconstitution were generated by PCR amplification and purified as previously described (Burdett et al, 2023; Lowary and Widom, 1998; Wilson et al, 2019). A table for DNA sequences can be found in Appendix Table S4. Fluorescent dyes were incorporated in the primers (IDT technologies, HPLC pure). PCR reactions using Pfu polymerase and oligonucleotides were pooled, filtered through a 0.4 μm filter, and applied to a 6 ml ResourceQ column (Cytiva) pre-equilibrated with 10 mM Tris pH 7.5 and 1 mM EDTA. The column was then washed extensively with 500 mM NaCl, before eluting across a 12 CV gradient from 500 mM NaCl to 900 mM NaCl. Fractions were analysed by native-PAGE, and fractions containing the desired product were pooled, concentrated by ethanol precipitation and resuspended in 10 mM Tris pH 8.

### Nucleosome wrapping
Nucleosomes were reconstituted as previously described (Burdett et al, 2023; Dyer et al, 2004), with some minor modifications. Purified octamers were incubated with DNA in 1:1.2 molar ratio and wrapped using an 18 h exponential salt reduction gradient (2 M KCl to 0.2 M KCl in 15 mM HEPES pH 7.5, 1 mM DTT, 1 mM EDTA) and then dialysed to 15 mM HEPES pH 7.5, 100 mM NaCl, 1 mM DTT, 1 mM AEBSF. Free DNA was removed from mono-nucleosomes by partial PEG precipitation, using 9% (w/v) PEG-6000 for 145 bp DNA and for 175 bp DNA and 9.5% PEG-6000 (w/v) and 150 mM NaCl was added. Pellets were resuspended in 15 mM HEPES pH 7.5, 100 mM NaCl, 1 mM DTT, 1 mM AEBSF. The extent and purity of nucleosomes wrapping was checked by native-PAGE and SDS-PAGE analysis.

## Transmission electron microscopy sample preparation and data collection

A complex of DNMT3A1-DNMT3L-StrepII was formed with H2AKc119ub nucleosomes wrapped with 195 bp Widom601 DNA in a 2.5:1 molar ratio protein:protomer for 1 h on ice.

Holey carbon R2/2 300 mesh grids (Quantifoil Micro Tools GmbH) were glow-discharged for 90 s at 25 mA using a PELCO easiGlow glow discharge cleaning system. The complex was diluted with s-adenosyl methionine (SAM) in 15 mM HEPES pH 7.5, 1 mM DTT to a final concentration of 110–130 ng/μl (DNA concentration), 65 mM NaCl and 100 μM SAM and immediately vitrified by

applying 3.5 µl to the glow-discharged Quantifoil grids, followed by immediate blotting (blot force = 0 N, blot time = 8 s) and plunge-freezing in liquid ethane cooled by liquid nitrogen, using a FEI Vitrobot IV (ThermoFisher) at 100% relative humidity and with a chamber temperature set at 4 °C. Grids were screened for ice quality and a small dataset was collected and processed to 2D classes on a TF20 microscope (University of Edinburgh, Cryo-transmission EM facility).

### Data collection of DNMT3A1-DNMT3L-StrepII on H2AKc119ub nucleosomes

Two separate datasets from two grids were collected on a FEI Titan Krios transmission electron microscope (ThermoFisher) operating at 300 keV equipped with a K3 camera (Gatan), using a magnification of 105,000× and a pixel size of 0.829 Å/pixel. Movies were recorded using the EPU automated acquisition software in counting super resolution mode and a total dose of 61 e⁻/Å² over 65 frames. One dataset was 6969 movies, the second dataset was 8522 movies, defocus values ranged from −1.5 µm to −3.0 µm. Detailed information on data collection and structure refinement of DNMT3A1-DNMT3L-StrepII on H2AKc119ub nucleosomes is shown in Appendix Table S1.

## Cryo-EM image processing

A schematic of the data processing pipeline in shown in Appendix Fig. S1. All movies were motion corrected and dose weighted with MotionCor2 (RELION). For Map1 (Fig. 4F; Appendix Fig. S2), data was processed in RELION 3 and 4 (Kimanius et al, 2021; Zivanov et al, 2018), except for the initial model (generated in cryoSPARC) (Punjani et al, 2017). The 6969 movies of the first dataset were CTF estimated using GCTF in RELION. 1,824,711 particles were picked using 2D nucleosome templates, and extracted with a box size of 384, binned by 2. Two rounds of 2D classification were performed yielding 446,680 particles in nucleosome-like classes. 3D classification was done with 8 classes based on the ab initio model generated from cryoSPARC processing of this dataset. One class showed high quality nucleosomes with added density (203,793 particles) and this map and particles were refined using RELION 3D auto-refine. The extra density on the linker DNA belonging to DNMT3A1-DNMT3L-StrepII clashed with the side of the box. To dela with this issue the nucleosome, rather than the whole particle, was re-cantered by re-extraction displacing all particles by 40 pixels in y direction (y −40) and a smaller box size of 320, binned by 2. An initial model was generated from these particles in RELION to check the centring of the particles and a 3D classification with three classes was done. Two classes (203,347 particles) showed good quality nucleosomes, and these were refined using RELION 3D auto-refine. A mask was created in RELION from the refined density and used for postprocessing. Postprocessing in RELION using B-factor −50, resulted in a 5.1 Å density map of DNMT3A1-DNMT3L-StrepII on a nucleosome. The local resolution across the map was estimated using RELION local resolution algorithm using a B-factor of −50 (Map 1 Appendix Fig. S1). Segmentation (Fig. EV1E) was performed on the filtered map from the local resolution estimation using Segger v2.5.3 in UCSF Chimera (Pettersen et al, 2004).

For Maps 2–4, the two DNMT3A1-DNMT3L-StrepII:H2AKc119ub nucleosomes datasets were initially treated independently.

Micrographs were imported into cryoSPARC (Punjani et al, 2017) and CTF parameters were estimated using patch CTF. Approximately 1000 particles were picked manually and 2D classified to produce templates for template-based automated picking in cryoSPARC. Two rounds of 2D classification were performed to discard poorly averaged particles. The most promising classes were pooled (527,966 particles & 529,296 particles) and used for ab initio reconstruction and separated into three (dataset 1) or two (dataset 2) ab initio classes. The best class comprising 214,344 and 330,169 particles were merged and re-extracted with a 384-pixel voxel size. Initial ab initio job led to 391,167 particles contributing to a high-resolution class, which yielded a map with overall 3.0 Å resolution from non-uniform refinement (Punjani et al, 2020), with density for ubiquitin and DNMT3A1 features, albeit weaker than core nucleosome features. To separate sample and structural hetero-geneity we created a loose mask either covering the UDR region or the ubiquitin-distal DNA region. We used these as focused masks in cryoSPARC 3D classification, with 2 classes and 0.85 similarity score (UDR density) or 4 classes and 0.5 similarity score (Ub-DNA density). For the UDR based mask on map corresponding to 191,397 particles showed the clearest density for this region and homogenous refinement using a wide dynamic mask of 10–18 Å, produced Map 2 showing density for UDR, ubiquitin, extra-nucleosomal DNA and some poor extra density expected to be from DNMT3A1-DNMT3L. With the same particles non-uniform refinement with local CTF optimisation produced a 3.1 Å Map 3. Focused classification of the Ubiquitin density revealed poorer overall density and no discrete ubiquitin states. Pooling 2 classes followed by non-uniform refinement produced map 4. Local resolution was estimated in cryoSPARC and final maps filtered using a B factor of −50.

## Model building

An initial model of DNMT3A1 and nucleosome was generated using ModelAngelo (Jamali et al, 2024), using the protein sequence for human histones, 195 bp DNA and residues 164–219 of DNMT3A1 with final 3.1 Å map 3 sharpened using a B factor of −50. This reliably built density into the unassigned UDR region with register that fit with biochemical observations. For model building the cryo-EM structure of human nucleosomes (PDB ID: 7XD1) (Ai et al, 2022) was combined with the DNMT3A1 UDR model and rigid-body docked into the reconstructed density in Chimera (Pettersen et al, 2004). DNA ends were removed on one side of the nucleosome, to represent the available density. The model was adjusted using Coot (Emsley et al, 2010) and ISOLDE (Croll, 2018) and extra residues added to unmodelled density. The model was iteratively refined using Phenix real space refine (Liebschner et al, 2019). Where density was lacking side chains were removed past the Cβ position. The overall quality of the model was assessed using MolProbity (Williams et al, 2018) and Phenix validation tools. All figures were prepared in Chimera or ChimeraX (Pettersen et al, 2004; Pettersen et al, 2021).

## Electrophoretic mobility shift assay

Nucleosomes wrapped with 6-carboxyfluorescein (5′ 6-FAM) labelled 175 bp Widom601 DNA or free 6-carboxyfluorescein (5′ 6-FAM) labelled DNA were incubated at a concentration of 2.4 nM

with various concentration ranges of proteins, as specified in figure legends, in EMSA buffer 15 mM HEPES pH 7.5, 75 mM NaCl, 0.05% (v/v) Triton X-100, 0.05 mg/mL BSA, 10% (v/v) glycerol, 1 mM DTT, 8% (w/v) sucrose, 0.01% (w/v) bromophenol blue) to a final volume of 12 μl. Competitor DNA was used in EMSAs with nucleosomes (0.5 mg/mL salmon sperm DNA, low molecular weight 31149-10g-f, Sigma-Aldrich). Samples were incubated on ice for 1 h and products were separated on 5% 19:1 acrylamide native PAGE gels using 1xTris Glycine as running buffer for 90 min at 4 °C. Gels were imaged for FAM signal (Excitation Blue light, Emission 532 nm) using Bio-Rad ChemiDoc MP. Quantification was done using Image Lab (Bio-Rad) and binding curves were analysed in GraphPad Prism 10 using non-linear regression-specific binding with hill slope.

For full-length protein nucleosomes wrapped with 195 bp Widom601 DNA or free 195 bp Widom601 DNA as used in methyltransferase assays were incubated at a concentration of 2.4 nM with a concentration range of 0–2.6 μM, 2x dilution series, in 17.5 mM HEPES pH 8, 150 mM NaCl, 2.5% (v/v) glycerol, 1 mM DTT, 0.1 mg/mL BSA, 1.5 mM MgCl$_2$, 8% (w/v) sucrose, 0.01% (w/v) bromophenol blue. Samples were incubated on ice for 1 h and products were separated on 5% 19:1 acrylamide native PAGE gels using 1xTris Glycine as running buffer for 90 min at 4 °C. Gels were stained with Diamond stain (Promega) and imaged on a Bio-Rad ChemiDoc MP.

## Nuclear magnetic resonance (NMR)

$^1$H-$^{15}$N HSQC NMR of $^{15}$N-labelled ubiquitin was performed with and without DMT3A$^{164-219}$ in identical normalised buffer (20 mM sodium phosphate pH 7.5, 150 mM NaCl, 5% (v/v) glycerol, 1 mM DTT). Spectra were taken of $^{15}$N ubiquitin (300 μM) and then DNMT3A1$^{164-219}$ was titrated into a final concentration of 300 μM DNMT3A1$^{164-219}$ and 125 μM of ubiquitin, at a molar ratio of 2.4:1. Titrations of DNMT3A1$^{164-219}$ were also tested to monitor any dose response shifts. All spectra were measured using a Bruker Avance NEO 800 MHz standard bore NMR spectrometer with Topspin (Bruker) software, at 298 K, with 32 scans, sweep width 12 ppm in the $^1$H dimension and 35 ppm in $^{15}$N dimension. Spectra were analysed using CCPN mr3.1.1 AnalysisAssign and the crosspeaks were assigned based on BMRB17769 (Cornilescu et al, 1998) and previous ubiquitin spectra.

## Nucleosome pull-down assay

Amylose resin (NEB) was equilibrated with amylose pull down buffer (50 mM Tris pH 7.5, 150 mM NaCl, 0.02% (v/v) NP-40, 10% (v/v) glycerol, 1 mM EDTA, 0.1 mg/mL BSA and 2 mM β-mercaptoethanol) and blocked with 1 mg/mL BSA in amylose pull down buffer. After washing 3x with amylose pull down buffer, His-MBP tagged proteins (6 μg unless differently specified) were immobilised on amylose beads for 1 h rotating at 4 °C and washed 3x with amylose pull down buffer. Nucleosomes (0.5 μg unless differently specified) were added and incubated for 2 h at 4 °C, washed with amylose pull down and resuspended in 30 μl 2x SDS loading buffer. Samples (6 μl) and 5% input were loaded on SDS-PAGE gels. Nucleosomes bound were detected using western blot followed by detection with histone antibodies (anti-H3 Abcam ab1791; anti-H2A Abcam ab18255, anti-H2B Abcam ab1790) and HRP conjugated secondary antibodies (anti-mouse IgG-HRP

Vector Labs PI2000, anti-rabbit IgG-HRP Vector Labs PI1000). Proteins were detected using anti-MBP antibody (anti-MBP NEB e8032s) and HRP conjugated secondary antibody, PonceauS (3% trichloroacetic acid, 3% sulfosalicylic Acid, 0.2% Ponceau) or stainfree gels (Mini-PROTEAN TGX Stain-Free Precast Gels).

For the alanine scanning experiment partially purified His-MBP-DNMT3A1$^{1-427}$ and variants were expressed in small scale and enriched from *E. coli* extract with a Ni-NTA bead purification step, as described above under protein purification. Protein was eluted and concentration determined, prior to loading on amylose beads at ~5 μg per pulldown.

## DNA methyltransferase assays

Methyltransferase activity of full-length DNMT3A1-DNMT3L and DNMT3A2-DNMT3L was measured using Promega MTase-Glo™ Methyltransferase Assay following the small volume (10 μl) protocol with minor adjustments. Nucleosomes wrapped with 195 bp Widom 601 DNA (0–0.5 μM) and S-adenosyl methionine (SAM, 20 μM) were incubated with full-length DNMT3A1-DNMT3L (0.5 μM), DNMT3A2-DNMT3L (0.5 μM) or DNMT3A1-DNMT3B3 (1 μM) in 20 mM HEPES pH 8, 150 mM NaCl, 3 mM MgCl$_2$, 0.1 mg/ml BSA, 1 mM DTT, 0.25% glycerol for 1 h at 37 °C. 5x reaction mixture (1x final, Promega MTase-Glo™ Methyltransferase Assay) was added and the mixture was incubated for 30 min at room temperature followed by 30 min incubation with detection reagent (Promega MTase-Glo™ Methyltransferase Assay). Luminescence was detected using a SpectraMax iD5 (Molecular Devices) plate reader. Methyltransferase assay with high concentrations of DNA (Fig. S10D) was performed with 0–3300 nM DNA and 0–250 nM nucleosomes in 20 mM Tris pH 8, 50 mM NaCl, 3 mM MgCl$_2$, 0.1 mg/ml BSA, 1 mM DTT. Methyltransferase activity in presence of H3 (1-20) peptide was performed with 0.5 μM 195 bp DNA, S-adenosyl methionine (SAM, 20 μM), and 0–10 μM H3 peptide (eurogentec/anaspec, AS-62753, H-ARTKQTARKSTGGKAPRKQL-OH). Methyltransferase activity in presence of peptides H3 K4me0 (eurogentec/anaspec, AS-62753, H-ARTKQTARKSTGGKAPRKQL-OH), H3K36me2 (Anaspec, RKAA-PATGGV - K(Me2) - KPHRYRPGTV - K(BIOTIN) and H3K36me0 (Anaspec, H-ATKAARKSAPATGGVKKPHRYRPGGK(biotin)-OH) was performed with 0.5 μM 195 bp DNA, S-adenosyl methionine (SAM, 20 μM), and 10 μM peptides.

A SAH standard curve was made (0–1 μM SAH) in 20 mM HEPES pH 8, 150 mM NaCl, 3 mM MgCl$_2$, 0.1 mg/ml BSA, 1 mM DTT, 0.25% (v/v) glycerol following the same steps as the methyltransferase reaction. Raw data were baseline subtracted and converted using the slope of the SAH standard curve. Michaelis-Menten curves were fit using Graphpad Prism 10.

## Nanopore sequencing

Two technical replicates of DNA methylated in in vitro methyltransferase assays (nucleosomes or DNA (0.5 μM), S-adenosyl methionine (SAM, 20 μM), full-length DNMT3A1-DNMT3L (0.5 μM)) was pooled and treated with proteinase K (NEB) and purified using Monarch PCR & DNA cleanup kit (NEB). 20 ng of each DNA sample was repaired, end-prepped and barcoded before the ligation of sequencing adapters using the Native Barcoding Kit 24 V14 (Oxford Nanopore Technologies, #SQK-NBD114.24) in conjunction with 4 NEBNext modules: NEB Blunt/TA Ligase

Master Mix (NEB, #M0367), NEBNext FFPE Repair Mix (NEB, #M6630), NEBNext Ultra II End repair/dA-tailing Module (NEB, #E7546) and NEBNext Quick Ligation Module (NEB, #E6056). Sequencing was performed on the Oxford Nanopore MinION Mk1b (ONT, #MIN-101B) using a Flongle adapter (ONT, #ADP-FLG001) and a R10.4.1 flongle flow cell (ONT #FLO-FLG114). Ten total barcoded samples equating to 2 biological replicates of each sample (4 total replicates each) and a single replicate for acidic patch mutation and DNA alone (2 total replicates each) were pooled. Approximately 15 fmol of the sequencing library was loaded onto a single flow cell.

Raw POD5 files were then processed using *Dorado* (*v0.7.2*) to call bases and modifications (5mCG and 5hmCG), trim barcodes and sequencing adapters and demultiplex the samples (settings: *basecaller –trim all* and *demux* with default settings, basecalling model: dna_r10.4.1_e8.2_400bps_hac@v5.0.0_5mC_5hmC@v1 and settings). Reads were aligned to the 195 bp DNA sequence using Dorado *aligner* with integrated *Minimap2-2.28* (*r1209*, and settings: *-k 5 -w 3*). Read alignment statistics can be found in Appendix Table S5. Following alignment, methylation data was extracted at reference CpG sites using *modkit pileup* (*v0.3.1*, settings: *--cpg*). The resulting bedMethyl files were then analysed using R (*v4.4.1*). We then calculated the weighted mean percentage methylation for CpGs within the linker region from the sum of the modified 5mC count and valid coverage for those CpGs.

## Missense variants analysis

Missense variant analysis was performed as described before (Deak and Cook, 2022). Briefly, data on missense variants associated with the human DNMT3A1 gene (transcript ENST00000264709.3, genome build GRCh37/hg19) were retrieved from the gnomAD v2.1.1 dataset (Karczewski et al, 2020). Plot Protein Converter (Deak and Cook, 2022) was used to filter the data for non-deleterious variants and format it for the Plot Protein program (Turner, 2013), enabling visualisation of variants on the DNMT3A1 protein sequence.

## Ubiquitin pull down assays

Amylose resin (NEB) was equilibrated with amylose pull down buffer (50 mM Tris pH 7.5, 150 mM NaCl, 0.02% (v/v) NP-40, 10% (v/v) glycerol, 1 mM EDTA, 0.1 mg/mL BSA and 2 mM β-mercaptoethanol) and blocked with 1 mg/ml BSA in amylose pull down buffer. After washing 3x with amylose pull down buffer, His-MBP and His-MBP-DNMT3A$^{1-427}$ (50 µg) were immobilised on amylose beads for 1 h rotating at 4 °C and washed 3x with amylose pull down buffer. His-TEV-UbiquitinG76C (50 µg) was added and incubated for 2 h at 4 °C, washed with amylose pull down and resuspended in 30 µl 2x SDS loading buffer. Inputs of ubiquitin, His-MBP and His-MBP-DNMT3A1$^{1-427}$ (2%) were loaded as control. Proteins were detected using stainfree gels (Mini-PROTEAN TGX Stain-Free Precast Gels), ubiquitin was detected using anti-ubiquitin antibody (Santa Cruz sc-8017) and HRP conjugated secondary antibody.

## Analytical Ubiquitin and DNMT3A$^{1-277}$ coelution assay

DNMT3A1$^{1-277}$ (20 µg) with or without ubiquitin in a molar ratio of 1:5 was incubated on ice for 1 h at a concentration of 4 mg/ml. The

protein or protein complex was diluted in size exclusion buffer to 2 mg/mL and loaded on a Superdex increase S200 3.2/300 column equilibrated with 20 mM HEPES pH 7.5, 150 mM NaCl, 1 mM DTT, 5% (v/v) glycerol. Eluted peak fractions were analysed on SDS-PAGE gels.

## Cross-linking mass spectrometry

For cross-linking mass spectrometry of a complex of full-length DNMT3A1-DNMT3L and H2AK119ub modified nucleosomes, a 2.5:1 molar ratio (DNMT3A-3L:nucleosomes) was cross-linked using 1-ethyl-3-(3-dimethylaminopropyl)carbodiimide hydrochloride (EDC) and N-hydroxysulfosuccinimide in a w/w ratio of 1:7.5–30:15–60 nucleosome:EDC:sulfo-NHS in 50 mM HEPES pH 7.5, 150 mM NaCl, 1 mM DTT. The complex was cross-linked for 3 h on ice then quenched with 50 mM Tris pH 7.5 and 50 mM ammonium bicarbonate.

For cross-linking mass spectrometry of a complex of DNMT3A1$^{1-277}$ and H2AK119ub modified nucleosomes, a 2.5:1 molar ratio (DNMT3A:nucleosome) was cross-linked using 1-ethyl-3-(3-dimethylaminopropyl)carbodiimide hydrochloride (EDC) and N-hydroxysulfosuccinimide in a w/w ratio of 1:7.5:15 nucleosome:EDC:sulfo-NHS in 50 mM HEPES pH 7.5, 150 mM NaCl, 1 mM DTT. The complex was cross-linked for 3 h on ice then quenched with 50 mM Tris pH 7.5 and 50 mM ammonium bicarbonate.

For cross-linking mass spectrometry of a complex of DNMT3A1$^{1-427}$ and H3Kc36me2 modified nucleosomes, a 2.5:1 molar ratio (DNMT3A:nucleosomes) was cross-linked using 1-ethyl-3-(3-dimethylaminopropyl) carbodiimide hydrochloride (EDC) and N-hydroxysulfosuccinimide in a w/w ratio of 1:15–30:30–60 nucleosome:EDC:sulfo-NHS in 50 mM HEPES pH 7.5, 150 mM NaCl, 1 mM DTT. The complex was cross-linked for 4 h on ice then quenched with 50 mM Tris pH 7.5.

Cross-linked complexes were separated on an SDS PAGE gel. The bands running at a higher molecular weight than DNMT3A1 were excised and the proteins were digested following previously established protocol (Maiolica et al, 2007). Briefly, proteins were reduced with 10 mM DTT for 30 min at 37 °C, alkylated with 55 mM iodoacetamide for 20 min at room temperature and digested using 13 ng/µl trypsin (Promega) overnight at 37 °C. Digested peptide were desalted using C18-StageTips (Rappsilber et al, 2003; Rappsilber et al, 2007) for LC-MS/MS analysis.

LC-MS/MS analysis was performed using Orbitrap Fusion Lumos (Thermo Fisher Scientific) with a "high/high" acquisition strategy. The peptide separation was carried out on an EASY-Spray column (50 cm × 75 µm i.d., PepMap C18, 2 µm particles, 100 Å pore size, Thermo Fisher Scientific). Mobile phase A consisted of water and 0.1% v/v formic acid. Mobile phase B consisted of 80% v/v acetonitrile and 0.1% v/v formic acid. Peptides were loaded at a flow rate of 0.3 µl/min and eluted at 0.2 µl/min using a linear gradient going from 2% mobile phase B to 40% mobile phase B over 139 min (each sample has been running three time with different gradient), followed by a linear increase from 40% to 95% mobile phase B in 11 min. The eluted peptides were directly introduced into the mass spectrometer. MS data were acquired in the data-dependent mode with 3 s acquisition cycle. Precursor spectrum was recorded in the Orbitrap with a resolution of 120,000. The ions with a precursor charge state between 3+ and 8+ were isolated with a window size of 1.6 *m/z* and fragmented using high-energy collision dissociation (HCD) with collision energy 30. The fragmentation spectra were recorded in the Orbitrap with a resolution

of 30,000. Dynamic exclusion was enabled with single repeat count and 60 s exclusion duration.

The mass spectrometric raw files were processed into peak lists using ProteoWizard (version 3.0) (Kessner et al, 2008), and cross-linked peptides were matched to spectra using Xi software (version 1.7.6.1) (Mendes et al, 2019) with in-search assignment of monoisotopic peaks (Lenz et al, 2018). Search parameters were MS accuracy, 3 ppm; MS/MS accuracy, 10 ppm; enzyme, trypsin; cross-linker, EDC; max missed cleavages, 4; missing mono-isotopic peaks, 2; fixed modification, carbamidomethylation on cysteine; variable modifications, oxidation on methionine and phosphorylation on threonine for phosphorylated sample; fragments, b and y ions with loss of $H_2O$, $NH_3$ and $CH_3SOH$. K36Cme2 at H3.1 sequence using Customer setting modification to define.

## Data availability

The cryo-EM density map and associated meta data for the NCP deposited at the Electron Microscopy Data Bank at www.ebi.ac.uk/emdb/EMD-18778 & www.ebi.ac.uk/emdb/EMD-18793, raw data at EMPIAR www.ebi.ac.uk/empiar/EMPIAR-12359 and the structural model at www.rcsb.org/structure/8QZM. Cross-linking mass spectrometry data has been uploaded to PRIDE database under www.ebi.ac.uk/pride/archive/projects/PXD046529 & www.ebi.ac.uk/pride/archive/projects/PXD056554.

The source data of this paper are collected in the following database record: biostudies:S-SCDT-10_1038-S44319-024-00306-3.

## Peer review information

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

## Acknowledgements

We thank Adrian Bird and members of the Wilson lab for helpful discussions and critical reading of the manuscript. MDW's work is supported by the Sir Henry Dale Fellowship from the Wellcome Trust (210493/Z/18/Z), Medical Research Council (T029471/1), Scottish high-field NMR centre seed funding and University of Edinburgh. HW work is part supported by Institutional Strategic Support Fund (IS3-R1.37 22-23). Work in the Voigt lab was supported by the Wellcome Trust ([104175/Z/14/Z], Sir Henry Dale Fellowship to PV) and the UK Biotechnology and Biological Sciences Research Council (BBS/E/B/000C0421). Work in the DS laboratory is supported by an MRC university grant to the MRC Human Genetics Unit. DV, AD and DK are supported by PhD studentships from the Darwin Trust of Edinburgh. WR and JAW are funded by the Wellcome trust integrative cellular mechanisms PhD program (218470). MP is supported by the MRC Human Genetics Unit PhD program. GD's work is supported by BBSRC EASTBIO [BB/M010996/1]. This work was supported by the Edinburgh Protein Production Facility (EPPF), which receives funding from a

core grant (203149) to the Wellcome Centre for Cell Biology at the University of Edinburgh. Grid screening was performed in the cryo-EM facility in School of Biological Sciences at the University of Edinburgh, we are grateful to Maarten Tuijtel and Martin Singleton for their support. The cryo-EM facility was set up with funding from the Wellcome Trust (087658/Z/08/Z) and SULSA. We are grateful to the Rappsilber lab for access to the Xi server for mapping protein cross-links. We thank Arthur Riggs, Joe Landry (via Addgene) Duncan Sproul and Frank Sicheri for gifts of plasmids. We are very grateful to David Owen, Julika Radecke and Matt Byrne at Diamond for access and support of the Cryo-EM facilities at the UK national electron bio-imaging centre (eBIC), proposal EM-BI24557 & EM-BI31827, funded by the Wellcome Trust, MRC and BBSRC. We are grateful to the Edinburgh Clinical Research Facility Genetics Core and Martin Taylor for insights to nanopore sequencing. This work was supported by the Wellcome Centre for Cell Biology Mass spectrometry facility which receives funding from a core grant to the Wellcome Centre for Cell Biology (203149) and a Multi-User Equipment grant(s) (108504). This work was supported by funding for the Wellcome Discovery Research Platform for Hidden Cell Biology [226791] and we gratefully acknowledge support from the Proteomics and Structural Biology cores. We thank Logan Mackay in SIRCAMS school of chemistry, university of Edinburgh for mass spec analysis. We thank Juraj Bella in the NMR facility, School of Chemistry, University of Edinburgh for help with NMR experiments.

## Author contributions

**Hannah Wapenaar**: Conceptualization; Data curation; Formal analysis; Investigation; Methodology; Writing—original draft; Project administration; Writing—review and editing. **Gillian Clifford**: Data curation; Formal analysis; Investigation; Methodology. **Willow Rolls**: Data curation; Formal analysis; Investigation; Methodology; Writing—review and editing. **Moira Pasquier**: Data curation; Formal analysis; Writing—review and editing. **Hayden Burdett**: Data curation; Formal analysis; Investigation; Methodology; Writing—review and editing. **Yujie Zhang**: Data curation; Formal analysis; Investigation; Methodology; Writing—review and editing. **Gauri Deák**: Data curation; Formal analysis; Investigation; Methodology; Writing—review and editing. **Juan Zou**: Data curation; Formal analysis; Investigation; Methodology; Writing—review and editing. **Christos Spanos**: Formal analysis. **Mark R D Taylor**: Data curation; Formal analysis; Investigation; Methodology; Writing—review and editing. **Jacquie Mills**: Data curation; Formal analysis; Investigation; Methodology; Writing—review and editing. **James A Watson**: Data curation; Formal analysis; Investigation; Methodology; Writing—review and editing. **Dhananjay Kumar**: Investigation; Methodology. **Richard Clark**: Formal analysis; Investigation. **Alakta Das**: Investigation; Methodology. **Devisree Valsakumar**: Methodology. **Janice Bramham**: Data curation; Formal analysis; Investigation; Methodology; Writing—review and editing. **Philipp Voigt**: Funding acquisition; Methodology. **Duncan Sproul**: Data curation; Formal analysis; Supervision; Funding acquisition; Methodology; Writing—review and editing. **Marcus D Wilson**: Conceptualization; Resources; Data curation; Formal analysis; Supervision; Funding acquisition; Validation; Investigation; Methodology; Writing—original draft; Project administration; Writing—review and editing.

Source data underlying figure panels in this paper may have individual authorship assigned. Where available, figure panel/source data authorship is listed in the following database record: biostudies:S-SCDT-10_1038-S44319-024-00306-3.

## Disclosure and competing interests statement

The authors declare no competing interests.

# Expanded View Figures

▶

**Figure EV1.  Preparation of full-length DNMT3A1-3L and nucleosomes for enzymology and Cryo-EM.**

(A) Sequence of 195 bp DNA optimised for the DNMT3A1 catalytic domain, used for nucleosomes in cryo-EM and enzymology experiments. The sequence comprises strong positioning sequence Widom 601 followed by a 50 bp linker. The linker is engineered for optimum DNMT3 catalytic activity (Gao et al, 2020; Mallona et al, 2021) with flanking sites marked in green box and methylated CpG highlighted green. (B) Native gels and SDS-PAGE gels of 195 bp DNA, unmodified and H2AKc119ub nucleosomes used in Fig. 1C and for cryo-EM experiments. Nucleosomes (100 ng based on DNA concentration) were loaded on 5% 19:1 acrylamide native gels using native loading buffer (8% sucrose, 0.02 mg/mL BSA, 0.01% bromophenol blue) and run 100 V for 90 min, then stained with Diamond stain. Native gels show shift in mobility of DNA when wrapped into a nucleosome. H2AK119ub nucleosome appears as two bands on gel due to conformation flexibility of the ubiquitin altering mobility. For SDS-PAGE, 900 ng nucleosomes were loaded on 17% 37.5:1 acrylamide gels using SDS-PAGE loading buffer (56 mM Tris pH 6.8, 11% glycerol, 2.4%SDS, 0.016% bromophenol blue) and run 100 V for 10 min then 200 V for approximately 60 min, then stained with colloidal Coomassie blue stain. Denatured gels show equal loading of all four histones and reduced mobility of ubiquitylated H2A. (C) Schematic of the purification strategy of full-length DNMT3A1-DNMT3L-StrepII complex used. Full-length His-MBP-DNMT3A1 and His-GFP-DNMT3L-StrepII were co-expressed in *E. coli* BL21 RIL. Cells were lysed and purified with Nickel NTA affinity purification followed by StrepTrap. His-MBP and His-GFP tags were cleaved using TEV protease followed by a further purification using a heparin column. The salt was reduced to 150 mM NaCl and complex was flash frozen and stored at −80 °C. SDS-PAGE gel of purified full-length DNMT3A1-DNMT3L-StrepII complex. Asterisk indicates degradation product. (D) A sample of 40 µg of DNMT3A1-DNMT3L-StrepII (1.77 mg/mL) complex was run on analytical size exclusion chromatography to assess monodispersity. Two overlapping peaks (0.9 ml and 1.05 ml elution volume) corresponding to rough sizes for heterotetramer and heterodimeric species. An SDS-PAGE gels was run of all fractions across the peaks. (E) Surface rendering of the 5.1 Å DNMT3A1-3L: H2AK119ub Nucleosome at 0.02, 0.01 and 0.0065 threshold. (F) Segmented surface rendering of the 5.1 Å DNMT3A1-3L: H2AK119ub Nucleosome. Locally filtered segmented map with nucleosome density displayed at 0.0065 threshold and ubiquitin and UDR density displayed 0.05. Rigid body fitting of nucleosome model (PDB 7VVU (Qu et al, 2022)), guided segmentation to identify non-nucleosome features including ubiquitin/UDR density (purple), nucleosome contacting UDR density (cyan) and linker DNA adjacent DNMT3A1-DNMT3L density (cyan). (G) Cross-linking mass spectrometry of full-length DNMT3A1-DNMT3L-strepII to H2AKc119ub nucleosomes wrapped with 195 bp DNA. (H) Circular representations of combined detected cross-links over a range of EDC ratios (7.5–30:1 molar ratio) measured in two separate experiments. (left) Only cross-links with DNMT3A1 shown. (middle) Cross-links to DNMT3A1 mapped on a nucleosome model (PDB 1AOI (Luger et al, 1997)) with ubiquitin docked close to H2AK119 and DNA extended to 195 bp. More abundant cross-linked residues shown as spheres, less as stick representation. Model coloured, Yellow, H2A; Red, H2B; Blue, H3; Green, H4. (right) all inter- and intra-protein cross-links detected in the two experiments. Source data are available online for this figure.

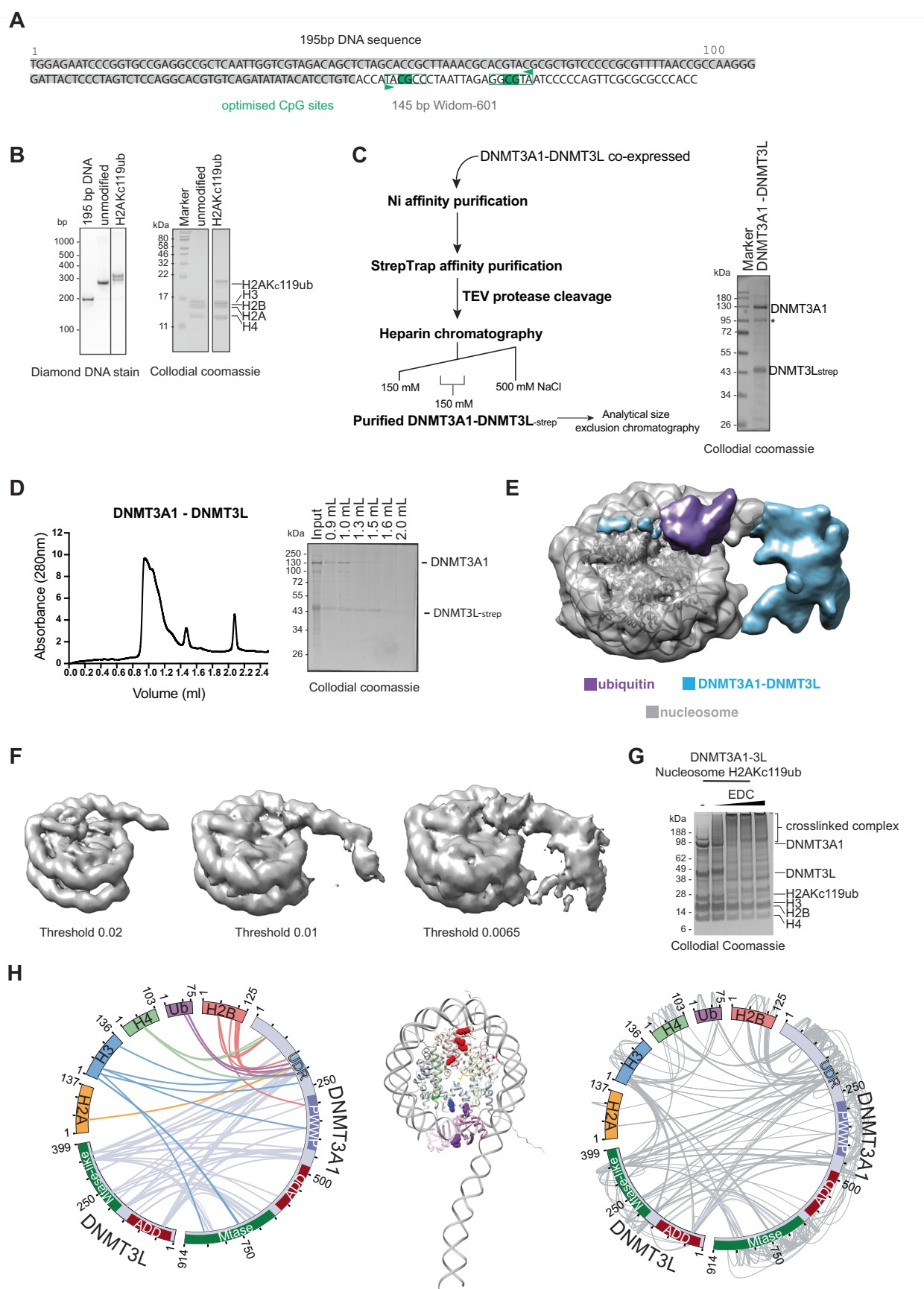

**A**

195bp DNA sequence

optimised CpG sites          145 bp Widom-601

**B** Diamond DNA stain / Collodial coomassie

**C** DNMT3A1-DNMT3L co-expressed → Ni affinity purification → StrepTrap affinity purification → TEV protease cleavage → Heparin chromatography → 150 mM / 500 mM NaCl → Purified DNMT3A1-DNMT3L-strep → Analytical size exclusion chromatography

Collodial coomassie

**D** DNMT3A1 - DNMT3L

Collodial coomassie

**E** ubiquitin  DNMT3A1-DNMT3L  nucleosome

**F** Threshold 0.02  Threshold 0.01  Threshold 0.0065

**G** DNMT3A1-3L Nucleosome H2AKc119ub

Collodial Coomassie

**H**

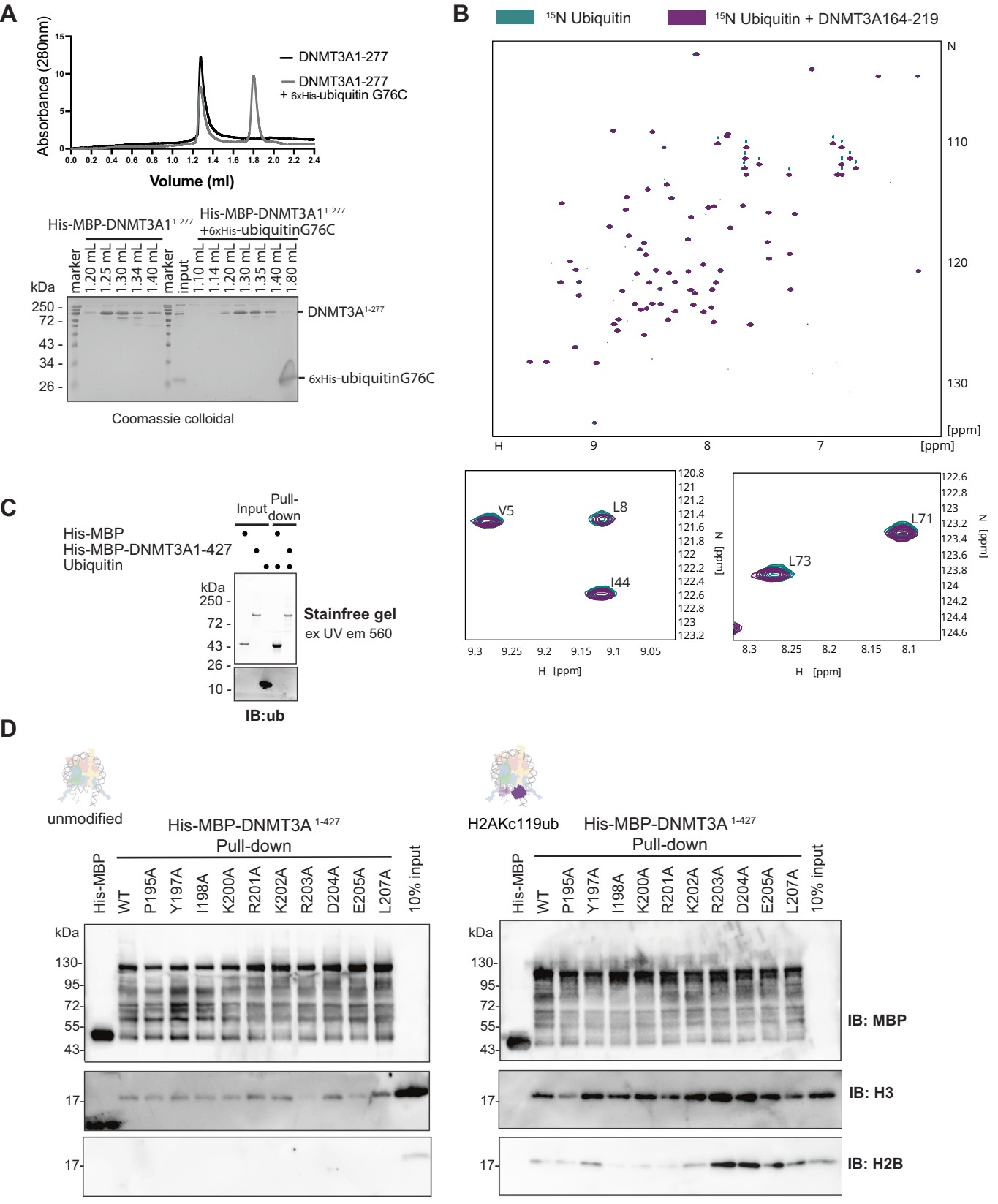

**Figure EV2.  DNMT3A1 UDR region alone does not interact with ubiquitin.**

(A) Size exclusion chromatography of DNMT3A1$^{1-277}$ with and without 6xHis-ubiquitin. DNMT3A1$^{1-277}$ (20 μg) was loaded on a Superdex 200 Increase 3.2/300 (Cytiva) column with (grey) or without (black) first incubating with 5x molar excess of 6xHis-ubiquitin (20 μg). An SDS-PAGE gel was run of the fractions of each peak. (B) HSQC NMR of $^{15}$N ubiquitin with and without DMT3A$^{164-219}$. Spectra were taken of $^{15}$N ubiquitin (green), then DNMT3A1$^{164-219}$ was added (purple). (bottom) magnified view of assigned residues, highlighting crosspeaks for canonical hydrophobic patch residues (left) and C-terminal hydrophobic patch region (right). (C) Pull down assay measuring binding of 6xHis-ubiquitin to His-MBP-DNMTA1$^{1-427}$. His-MBP-DNMTA1$^{1-427}$ (50 μg) was immobilised on amylose beads (50 μL) and 6xHis-ubiquitin (50 μg) was added. Bound proteins were detected using stainfree UV detection of Mini-PROTEAN TGX Stain-Free Precast Gels. 6xHis-ubiquitin was detected using western blot followed by detection with antibodies against ubiquitin. (D) Pull-down assay using partially purified His-MBP, His-MBP-DNMT3A1$^{1-427}$ and His-MBP-DNMT3A1$^{1-427}$ alanine scanning mutants in the proposed ubiquitin interacting region of the UDR. Proteins were expressed and purified on Nickel affinity beads, eluted and bound to amylose beads prior to incubation with unmodified nucleosomes (left) and H2AKc119ub nucleosomes (right), prior to washing and detection by western blot. Source data are available online for this figure.

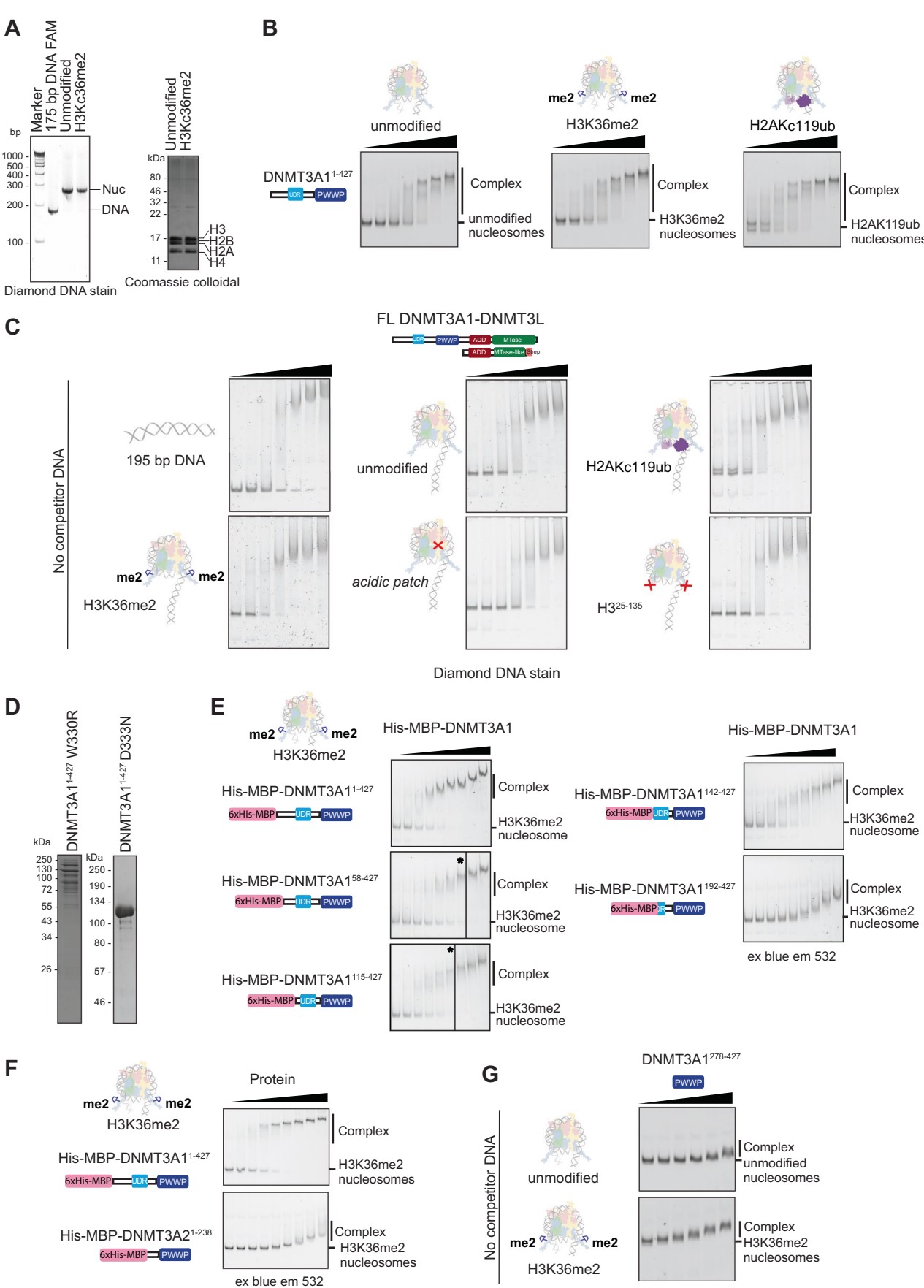

**Figure EV3. DNMT3A1-H3K36me2 nucleosome interaction is localised in N-terminal region and PWWP domains.**

(A) Native gels and SDS-PAGE gels of H3Kc36me2 nucleosomes. (B) EMSAs showing binding of DNMT3A1 1-427 after cleaving His-MBP to nucleosomes wrapped with FAM labelled 175 bp DNA. DNMT3A1 1-427 (0–10 μM, 2x dilution series) were mixed with unmodified, H3K36me2 and H2AKc119ub nucleosomes wrapped with 5′ 6-FAM labelled 175 bp DNA (5.4 nM) and incubated for 1 h on ice. Complexes were resolved by native-PAGE and imaged using blue light excitation and 532 nm emission filters. Gels show concentrations 9.8–1250 nM. (C) EMSA comparing binding of full-length DNMT3A1-DNMT3L-StrepII to 195 bp-wrapped unmodified, H3K36me2, H2AKc119ub, acidic patch (H2A$^{E61A/E91A/E92A}$ and H2B$^{E105A}$) and H3$^{25-135}$ nucleosomes and free 195 bp DNA. Limiting amounts (8 nM) of nucleosomes or DNA were incubated with increasing concentrations (0–2666 nM, 2x dilution series) of full-length DNMT3A1-DNMT3L-StrepII. Complexes were resolved by native-PAGE, stained with diamond stain and imaged using blue light. Representative gels show concentrations 41–2666 nM of one of two independent experiments. EMSA of unmodified, H2AKc119ub and DNA alone also appears in Fig. 1B. (D) SDS-PAGE gel of purified DNMT3A proteins used in Fig. 4B. (E) EMSA comparing binding of DNMT3A1 constructs with different lengths of the N-terminal region to H3K36me2 nucleosomes wrapped with 5′ FAM labelled 175 bp Widom601 DNA. Limiting amounts (2.3 nM) of H3K36me2 nucleosomes were incubated with increasing concentrations (0–8000 nM) of His-MBP-DNMT3A1 constructs. Complexes were resolved by native-PAGE and imaged using blue light excitation and 532 nm emission filters. Representative gels show concentrations 25–8000 nM of one of three experiments. (F) EMSA comparing DNMT3A splice isoform fragments DNMT3A1$^{1-427}$ and DNMT3A2$^{1-238}$. Limiting amounts (2.3 nM) of H3K36me2 nucleosomes were incubated with increasing concentrations (0–8000 nM) of protein. Complexes were resolved by native-PAGE and imaged using blue light excitation and 532 nm emission filters. Gels show concentrations 32–8000 nM. (G) EMSAs showing binding of DNMT3A1 278-427 (PWWP domain alone) after cleaving His-MBP to nucleosomes wrapped with FAM labelled 175 bp DNA. DNMT3A1 278-427 (0–25 μM, 1.5x dilution series) were mixed with unmodified and H3K36me2 nucleosomes wrapped with 5′ 6-FAM labelled 175 bp DNA (5.4 nM) and incubated for 1 h on ice. Complexes were resolved by native-PAGE and imaged using blue light excitation and 532 nm emission filters. Gels show concentrations 650–5000 nM. Source data are available online for this figure.

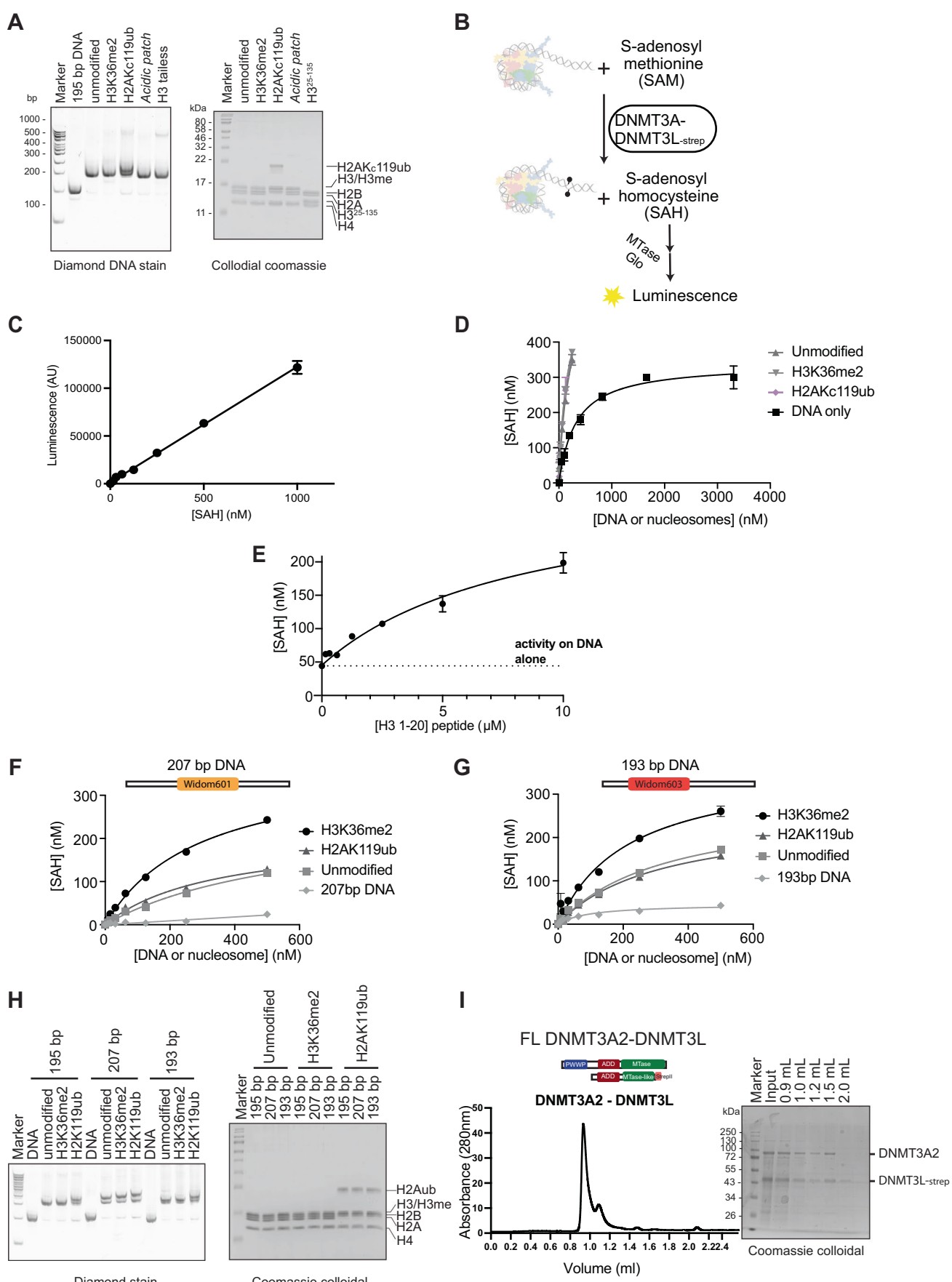

**Figure EV4.  DNMT3A1 activity differs on different nucleosome substrates.**

(A) Native gels and SDS-PAGE gels of modified and unmodified nucleosomes wrapped with 195 Widom601 DNA used in used for enzymology in Fig. 6 and EMSA experiments in Fig. EV3B. The unmodified, H2AK119ub and DNA SDS-PAGE and Native gels are duplicated from Fig. EV1B. (B) Schematic of Promega MTase-Glo™ Methyltransferase Assay. (C) Representative S-Adenosyl-L-homocysteine (SAH) standard curve used to convert luminescence units to SAH concentration. (D) Methyltransferase assay showing activity of full-length DNMT3A1-DNMT3L-StrepII on high concentrations (0–3300 nM) of 195 bp DNA. Activities on unmodified, H3K36me2 and H2AK119ub nucleosomes (0–250 nM) are shown as comparison. DNMT3A1-DNMT3L was incubated with increasing concentrations of DNA or nucleosomes for 1 h at 37 °C. Methyltransferase activity was detected using Promega MTase-Glo™ Methyltransferase Assay. Experiment done in duplicate. Michaelis-Menten curves were fit using GraphPad Prism 10. (E) Methyltransferase assay showing activity of full-length DNMT3A1-DNMT3L-StrepII on free 195 bp DNA in the presence of H3 (H1-20) peptide. The level of activity in the absence of H3 peptide is marked by a dotted line. Methyltransferase activity was detected using Promega MTase-Glo™ Methyltransferase Assay. Experiment done in duplicate. Michaelis-Menten curves were fit using GraphPad Prism 10. (F) Methyltransferase activity of full-length DNMT3A1-DNMT3L-StrepII on unmodified, H3K36me2, H2AKc119ub wrapped with 207 bp Widom601 DNA and free 207 bp DNA. DNMT3A1-DNMT3L was incubated with increasing concentrations of nucleosomes for 1 h at 37 °C. Methyltransferase activity was detected using Promega MTase-Glo™ Methyltransferase Assay. Experiment done in duplicate. Michaelis-Menten curves were fit using GraphPad Prism 10. (G) Methyltransferase activity of full-length DNMT3A1-DNMT3L-StrepII on unmodified, H3K36me2, H2AKc119ub nucleosomes wrapped with 193 bp Widom603 DNA and free 193 bp DNA. DNMT3A1-DNMT3L was incubated with increasing concentrations of nucleosomes for 1 h at 37 °C. Methyltransferase activity was detected using Promega MTase-Glo™ Methyltransferase Assay. Experiment done in duplicate. Michaelis-Menten curves were fit using GraphPad Prism 10. (H) Native gels and SDS-PAGE gels of modified and unmodified nucleosomes used in Fig. EV4F,G. (I) Analytical size exclusion chromatography of full-length DNMT3A2-DNMT3L-StrepII. 40 μg protein at 2 mg/ml was loaded on a Superdex 200 Increase 3.2/300 (Cytiva) column. An SDS-PAGE gels was run of all fractions across the peaks. Source data are available online for this figure.

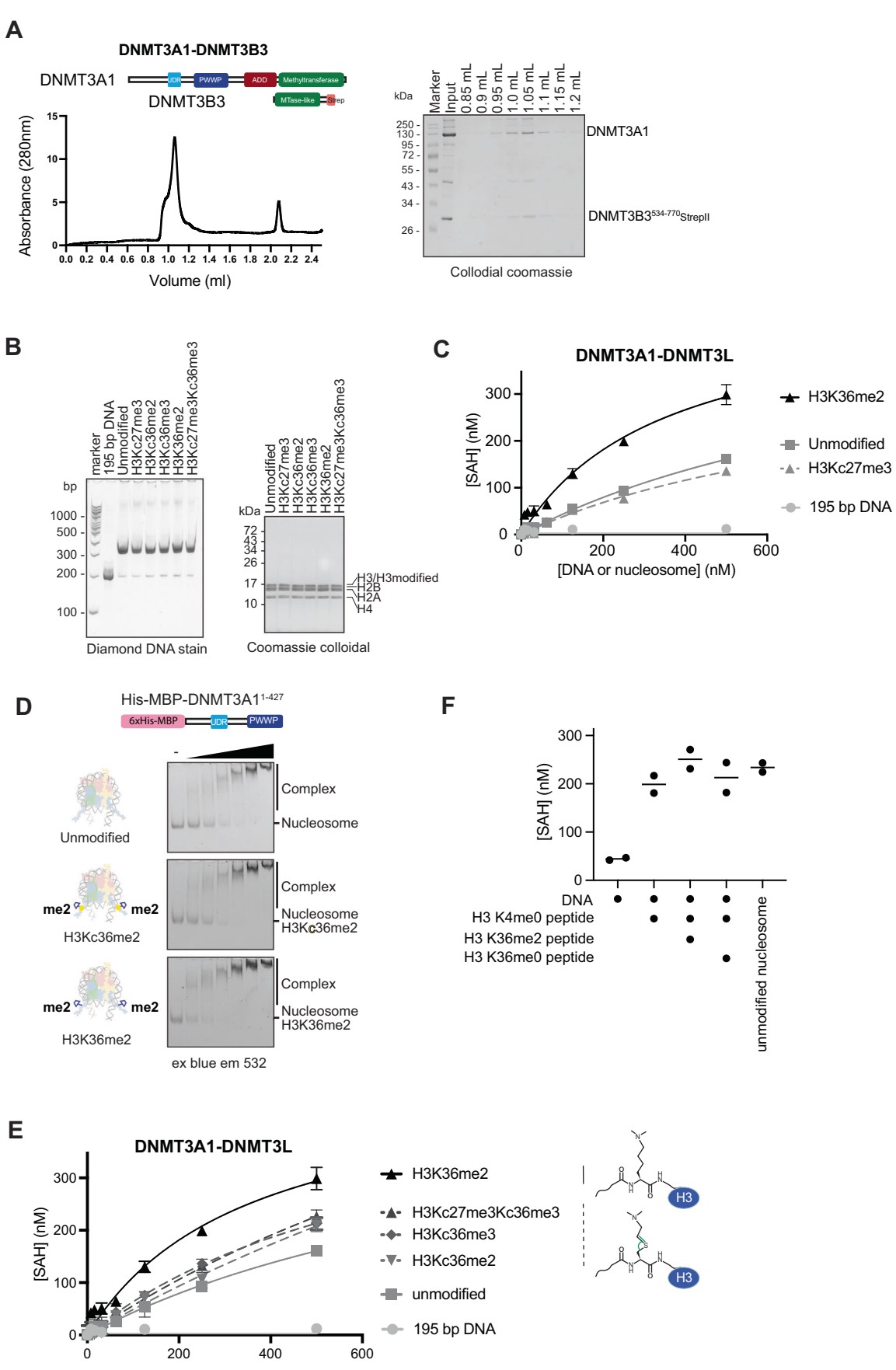

**Figure EV5. DNMT3A activity is only weakly affected by combinatorial PWWP-UDR status readout.**

(A) Analytical size exclusion chromatography of DNMT3A1-DNMT3B3$^{534-770}$-StrepII. 18 μg protein at 1.8 mg/ml was loaded on a Superdex 200 Increase 3.2/300 (Cytiva) column. An SDS-PAGE gels was run of all fractions across the peaks. (B) Native gels and SDS-PAGE gels of nucleosomes used for enzymology experiments in Fig. EV5C,E. (C) Methyltransferase activity of full-length DNMT3A1-DNMT3L-StrepII on unmodified, H3K36me2, H3Kc27me3 nucleosomes wrapped with 195 bp Widom601 DNA and free 195 bp DNA. DNMT3A1-DNMT3L was incubated with increasing concentrations of nucleosomes for 1 h at 37 °C. Methyltransferase activity was detected using Promega MTase-Glo™ Methyltransferase Assay. Experiment done in duplicate. Michaelis-Menten curves were fit using GraphPad Prism 10. (D) EMSA assay comparing DNMT3A1$^{1-427}$ interaction with unmodified, H3K36me2 and Methyl-lysine analogue H3Kc36me2 nucleosomes. Methyl-lysine analogues bind less well compared to true native methylated bond due too differences in the gamma-position on the modified sidechain. (E) Methyltransferase assay comparing activity of DNMT3A1-DNMT3L on nucleosomes with different H3 methylation states. Methyltransferase activity was detected using Promega MTase-Glo™ Methyltransferase Assay. Experiment done in duplicate. Michaelis-Menten curves were fit using GraphPad Prism 10. (left) Schematic of the H3 methyl lysine analogues and methylated lysine, green highlights carbon-sulphur bond differences in bond angle and bond length compared to carbon-carbon bond. (F) Methyltransferase assay showing activity of full-length DNMT3A1-DNMT3L-StrepII on free 195 bp DNA in the presence of differently methylated peptides (H3 1-20, H3$^{26-46}$K36me2, H3$^{21-44}$K36me0. Methyltransferase activity was detected using Promega MTase-Glo™ Methyltransferase Assay. Experiment done in duplicate. Michaelis-Menten curves were fit using GraphPad Prism 10. Source data are available online for this figure.

