## [Peer Review File · EMBO Reports]

The N-terminal region of DNMT3A engages the nucleosome surface to aid chromatin recruitment.

Hannah Wapenaar, Gillian Clifford, Willow Rolls, Moira Pasquier, Hayden Burdett, Yujie Zhang, Gauri Deak, Juan Zou, Christos Spanos, Mark Taylor, Jacquie Mills, James Watson, Dhananjay Kumar, Richard Clark, Alakta Das, Devisree Valsakumar, Janice Bramham, Philipp Voigt, Duncan Sproul, and Marcus Wilson

Corresponding author(s): Marcus Wilson (marcus.wilson@ed.ac.uk)

Review Timeline:

Submission Date:	30th Sep 24
Editorial Decision:	2nd Oct 24
Revision Received:	8th Oct 24
Editorial Decision:	16th Oct 24
Revision Received:	18th Oct 24
Accepted:	22nd Oct 24

Editor: Esther Schnapp

Transaction Report: A revised version of this manuscript was transferred to EMBO reports following peer review at the EMBO Journal.

Referee #1:

The article from Wapenaar et al. presents biochemical and structural analysis of the interaction between DNMT3A1 and the nucleosome. The authors use EMSA, pulldowns and SAH activity assays together with cryo-EM analysis to put forward a model in which they identify a UDR domain in the N-terminus of DNMT3A1 that binds the nucleosomal surface and favors ubiquitinated nucleosomes. The identification of a new interface between DNMT3A1 and nucleosomes is of high impact and interest. While this is addressed structurally and biochemically, I have some concerns that should be addressed before publication.

The structure is a strong piece of evidence for this new interaction, and it is somewhat a pity that it is presented relatively late in the manuscript. Some clarifications could help:

- o Fig 4D: Please show the fit of the PDB within the EM density for the DNMT3A1 peptide.
- o The ubiquitin density isn't resolved at high resolution. It is therefore not possible to determine based on these results whether this is ubiquitin. What is the evidence that this density is ubiquitin, rather than e.g. the PWWP or anything else? This should be addressed by crosslinking mass spec or comparable approaches.
- o Fig. S6. It's not always clear which maps these figures belong too. The euler angle plot of the other maps are missing. In addition, the particles still seem to have a preferential orientation. What do the authors think the effect of this could be on the final map? It seems like especially the DNMT3A1 containing part is overrepresented, could this have an effect on the extra density visible here?
- o Could the authors add images of the initial models and provide information about the low-pass filtering of these classes?
- o For dataset 1, a 3D refinement in RELION using a B-factor of -50 is performed. However, with datasets below 10Å, usually an automated B-factor estimation is preferred, to prevent the amplification of noise (Rosenthal & Henderson 2003). Could the authors clarify this choice?
- o Map 1: why did the authors choose a contour level of 0.0638? This seems quite high. Even though an extra density becomes visible this quite noisy and difficult to interpret.

The paper starts with suggesting that the N-terminal region in DNMT3A1 (aa 1-227) is critical for nucleosome binding (more important than the PWWP). A control with just DNA should be included in Figure 1B (these controls are run in Figure 2 but the 1-227 construct is not included). While the structure disproves that this construct binds DNA, a DNA control is critical to trust all the EMSA that are used in this work. Construct 1-227 should also be included in the EMSA in Figure 4A, 5C and 5D, to confirm the authors statements.

Similarly, Figure 1D requires DNA as control, and the 1-227 construct should be included. This will clarify the performance of these pull-down experiments, which not always align with the EMSA experiments (e.g. binding to unmodified nucleosomes).

The role of the acidic patch is critical, Figure S2C should be moved to main to make this point very obvious from the beginning. Moreover, In the experiment in Figure 5C, binding to the acidic patch mutant nucleosome (without K119Ub) is missing and should be included.

The authors quantify their binding experiments and activity assay by extrapolating binding curves, but no quantitative values are included (apparent K_d and K_{cat}/K_m). This would be helpful in comparing the different assays and the many figures. It would also help to estimate actual effects of some of the mutations/deletions. Could the authors include these and perhaps a table that summarizes all the binding and activity observations, so to aid into the interpretations?

As the authors state, the binding and activity data are not aligning. The authors should include activity assays with DNMT3A1-L mutants (e.g. deletion/mutations on the UDR or PWWP), to at least test if the UDR is important for activity at all.

The final model where the authors suggest different structural conformations depending on the histones PTMs is intriguing, but I don't see how this is supported by the data directly. For example, there is no evidence that the UDR is not engaging the acidic patch when the nucleosome are H3K36me2. It's quite the opposite actually, as shown already in Figure 1. Therefore, the model does not match the data and needs to be revisited.

Some experiments have been performed once (e.g. stated in Figure legends 5E). Replicates should be included during revision.

Regarding the crosslinking experiments: no intra-molecular crosslinks are shown in Figure 2A and S2B. This seems unlikely, so it would be helpful to share the complete results of these crosslinking mass spec experiments.

Finally, the story could benefit from some streamlining, both at the text and figures level, to get the key messages out more clearly.

Referee #2:

The authors report the identification and analysis of novel histone binding capabilities of the N-terminal part of DNMT3A1. The paper reports a large piece of technically demanding data, but it lacks novelty, because key literature findings have not been cited and considered accordingly. Moreover, several findings are not in good agreement with literature data. Hence, I cannot recommend publication of this work in EMBO J. and do not see that a revision could mitigate these problems. In my assessment, this paper needs to be completely rewritten, considering (and describing) all relevant literature. Then, finding must be critically discussed in the light of the literature data. Based on the outcome of this, the paper may be publishable, but I do not see it in a high-end journal.

Comments:

1) The lack of novelty is illustrated already in the abstract but reflected also in the detailed description of results:

"Using a combination of biochemical and structural approaches we find that DNMT3A interacts

using multiple interfaces with chromatin; directly binding generic nucleosome features as well as site-specific post-translational histone modifications." - This claim is not novel, as the chromatin interaction mediated by the ADD and PWWP domains are well known since more than a decade. "The N-terminal region, unique to the DNMT3A1 isoform, is essential for these interactions and stabilises H3K36me₂-nucleosome recruitment. Intriguingly, in the same region critical for nucleosome binding we also map a ubiquitylation-dependent recruitment motif (UDR). The UDR binds specifically to ubiquitylated H2AK119, explaining the previously observed recruitment to Polycomb-occupied heterochromatin." - This part rediscovers published findings reported in Weinberg et al. 2021 Nat Genet 53(6):794-800 and Gu et al. 2022 Nat Genet 54(5):625-636, which are not appropriately mentioned.

Together, these parts comprise 2/3 of the entire abstract.

This problem is also already appearing in the title, which makes statements one can find in numerous reviews on DNMT3A written after 2015.

2) In the first chapter of the results, the authors report that the DNMT3A1(1-427) region is essential for chromatin engagement and H3K36me_{2/3} binding. This claim is not in agreement with data reported in Weinberg et al. 2019, showing that the isolated PWWP domain specifically binds H3K36me_{2/3} containing nucleosomes and that DNMT3A2 expressed in cells delivers DNA methylation to chromatin regions containing H3K36me₂.

3) In the second chapter of the results (starting on p. 7), the author study by XL analysis the interaction of DNMT3A1(1-427) with nucleosomes and observe an interaction with the acidic patch of nucleosomes. However, the information that DNMT3A/3B3 heterotetrameric complexes bind to the acidic patch is available in the literature since 2020, when a cryo-EM paper reporting this finding was published (Xu et al., Nature 586(7827):151-155). Again, this important part of literature is not mentioned. However, the authors mapped the acidic patch interaction to the N-terminal part, while Xu et al. show the ability of the C-terminal domain of DNMT3B3. This discrepancy can be explained because the authors used the DNMT3A1(1-427) fragment in the XL analysis. Hence, it remains open, which of these two interaction modes is more important under which condition.

4) In the next results chapter (starting on p. 8), the authors studied methylation of nucleosomal DNA. Again, similar studies with better CpG site resolution have already been carried out, e.g. Brohm et al. Commun Biol. 2022 5(1):192. It is unclear in Fig. 3C, why the DNA alone was not methylated at all, suggesting something went wrong with the experiment. They later conclude that DNMT3A1-DNMT3L can engage linker DNA, which again was not novel (see for example Brohm et al.).

5) The cryo-EM of the ubiquitylated nucleosome presented in Fig. 4 and 5 is novel, but to me the interaction is not very convincing (see for example Fig. 4C). More experiments need to be conducted testing the relevance of critical residues to validate this structural model.

6) Finally, the authors report that the H2AK119ub interaction does not stimulate DNMT3A activity. However, based on their own observations hinting towards a close connection of H2AK119ub and H3K36me₂ binding, a nucleosome carrying the double mark, would have been needed, which was

not used. Instead, the H3K27me3/K36me3 double mark was tested, although it was shown before that H3K27me3 does not have an effect.

Referee #3:

The manuscript outlines the potential functionalities of the DNMT3A1 N-terminal region (before the PWWP domain). This includes the binding of the nucleosome acidic patch and/or binding of ubiquitylated H2AK119 (as reported already by two recent studies). The N-terminal region, in conjunction with the PWWP domain's H3K36me2 binding, could guide DNMT3A1 to chromatin domains marked with intergenic H3K36me2 or heterochromatin H2AK119ub. In addition, the authors demonstrated that DNMT3A1 activity could be stimulated by H3K36me2 but not by H2AK119ub (despite strong binding). The study comprises two parts: in vitro binding/activity assays using modified reconstituted nucleosomes and a cryo-EM structural study of DNMT3A1-DNMT3L in complex with ubiquitylated nucleosomes.

A critical concern regarding experimental design is the choice of using DNMT3L in complex with DNMT3A1. Since DNMT3L primarily functions in the germline and is silenced upon differentiation, whereas DNMT3A1 is the predominant somatic isoform, the use of DNMT3A1-DNMT3L may be an artifact. The authors should provide a justification for choosing 3A1-3L over DNMT3A1-3B3, for instance. Additionally, one needs to consider the competitive binding of 3A1 and 3B3 to the same acidic patch, as demonstrated by a recent 3A2-3B3 structure.

The cryo-EM structure of 3A1-3L raises concern about its poor quality, lacking representation of any tetrameric features of 3A-3L. To address this, I recommend focusing on the structure of the N-terminal fragment (residues 1-427) in complex with the nucleosome (with either modifications of H3K36me2 or H2AK119ub), considering that all binding data were obtained with this fragment. Currently, there is a disconnect between the binding data and structural data in the manuscript. For instance, Figure 4A (and 4E) presents binding data with His-MBP-DNMT3A1 fragments, while the corresponding structures are depicted as DNMT3A1-DNMT3L. Focusing on the same fragment will enhance the manuscript's coherence and strengthen the overall argument.

Other concerns:

"6xHis-MBP tagged fragments" - a positive control should be included for untagged fragment

"PWWP domain alone bound poorly, likely due to the low affinity of the interaction with the histone mark ... and affinity for competitor DNA used in the assay buffer" - Does the PWWP domain bind nucleosome in the absence of competitor DNA?

Page 7, line 3, The N-terminal region alone showed considerable interaction with nucleosomes irrespective of their modification state (Fig 1A), - wrong figure panel?

Response to reviewers comments Wapenaar et al., The N-terminal region of DNMT3A combines multiple chromatin reading motifs to guide recruitment.

Referee #1:

The article from Wapenaar et al. presents biochemical and structural analysis of the interaction between DNMT3A1 and the nucleosome. The authors use EMSA, pulldowns and SAH activity assays together with cryo-EM analysis to put forward a model in which they identify a UDR domain in the N-terminus of DNMT3A1 that binds the nucleosomal surface and favors ubiquitinated nucleosomes. The identification of a new interface between DNMT3A1 and nucleosomes is of high impact and interest. While this is addressed structurally and biochemically, I have some concerns that should be addressed before publication.

We thank the reviewer for their positive comments and appraisal.

The structure is a strong piece of evidence for this new interaction, and it is somewhat a pity that it is presented relatively late in the manuscript. Some clarifications could help:

We agree and based on comments by this and other reviewers have streamlined the manuscript, changing the order in which the results are presented. We now present the structure in the first figure and discuss this interaction with H2AK119ub before moving onto implications for H3K36me2 and activity.

- Fig 4D: Please show the fit of the PDB within the EM density for the DNMT3A1 peptide.

the full fit of the modelled sequence is now shown in Supplementary 3H (expansion of Supplementary 6H)

- The ubiquitin density isn't resolved at high resolution. It is therefore not possible to determine based on these results whether this is ubiquitin. What is the evidence that this density is ubiquitin, rather than e.g. the PWWP or anything else? This should be addressed by crosslinking mass spec or comparable approaches.

The reviewer makes a good point, we ascribed this density to ubiquitin and the ubiquitin contacting UDR. This was based on continuous density stemming from the tail of H2A around position 119 as expected and the UDR fragment of DNMT3A (old Figure 6A, now Figure 3A). We have amended the text to make the uncertainty of the annotation of this density clearer. As suggested, we have performed cross-linking mass spec for H2AK119ub nucleosome which shows crosslinks between DNMT3A1 N-terminal region and the nucleosome surface and ubiquitin. This is shown in new Supplemental Fig S6A.

Furthermore, biochemically we found the UDR region DNMT3A1¹⁶⁴⁻²¹⁹ (old figure 4E, new figure 1E) and the longer construct DNMT3A1¹⁻²⁷⁷ (adapted figure 4A) is sufficient for this interaction. We note that two complementary studies have become available since our initial submission (Gretarsson et al., bioRxiv 2024; Chen et al., *Nature communications* 2024). These both determine the structure of just a short fragment of DNMT3A1 (with no other domains present) on H2AK119ub nucleosomes in the presence of crosslinking agent. The density assigned to ubiquitin in both these studies is similarly localised to the density we described as ubiquitin.

- Fig. S6. It's not always clear which maps these figures belong too. The euler angle plot of the other maps are missing. In addition, the particles still seem to have a preferential orientation. What do the authors think the effect of this could be on the final map? It seems like especially the DNMT3A1 containing part is overrepresented, could this have an effect on the extra density visible here?

We apologise that this was not clear. We have relabelled these plots and the figure legends to clarify in new Figure S3. We have also included the Euler plots for the other maps in Fig S2.

We were concerned about preferred orientation given the issue in the H3K36me2-DNMT3A1-3L data. However, this does not cause significant affects for the H2AK119ub-DNMT3A-3L data (Map 3), with complete overall angular coverage in the Euler plots. We do not see any observable anisotropy in the maps and 3DFSC plot shows good overlap of directional FSC and global FSC (Fig S3F).

In addition, we have now included more quantitative measures of sample distribution in supplemental Table S1. The measured sphericity for 3D-FSC is 0.966 (out of 1). The sample compensation factor (SCF) is 0.94 (out of 1) (greater than 0.81 considered to be no significant anisotropy) (Baldwin, P. R., & Lyumkis, D. (2020).

The DNMT3A1 fragment is slightly lower resolution than the adjacent histones octamer (Supp Fig S2), which we believe explains the difference in representation seen at higher threshold (i.e overrepresentation). We do not believe this is due to masking as density is similar in looser masked Map 2.

- Could the authors add images of the initial models and provide information about the low-pass filtering of these classes?

This has been included in figure S2 (previously S6). Initial models were derived from the data using cryoSPARC *ab initio* approach. For the reion map, an initial cryoSPARC *ab initio* model of a subset of the data was used and low pass filtered to 50Å.

- For dataset 1, a 3D refinement in RELION using a B-factor of -50 is performed. However, with datasets below 10Å, usually an automated B-factor estimation is preferred, to prevent the amplification of noise (Rosenthal & Henderson 2003). Could the authors clarify this choice?

This map was to show our working methodology and confirm some additional density outside of the nucleosome was present in the data to show we were on the correct track.

Automated B factor assignment from linear fitting to Guinier fitting gave an estimated b factor of -203. Global sharpening B factor for this map is not the best approximation for this data given the large resolution range across the map (supp fig old s4 new S2) due to the structural flexibility observed. For this reason we used a lower global B factor to under sharpen the low-resolution features, leading to loss of interpretable detail rather than over sharpen, to avoid the interpretation and amplification of map noise.

- Map 1: why did the authors choose a contour level of 0.0638? This seems quite high. Even though an extra density becomes visible this quite noisy and difficult to interpret.

We agree this is a high threshold, we thought it appropriate to display the additional non-nucleosomal density present in the data.

We have now included Supplemental Figure S1E, showing this map at three representative threshold levels with no sharpening applied. The high structural heterogeneity in the data leads to

delocalisation of the DNMT3A1-3L signal due to particle averaging. This is expected as the nucleosome tethering elements of the structure contain long flexible linkers, while the DNA methyltransferase we believe can engage in multiple orientations on the flexible linker DNA. The lower threshold clearly shows the nucleosome and linker DNA projecting away from the nucleosome core. Flexible DNMT3A and ubiquitin densities are only observable at the higher thresholds. We mainly focus on the nucleosome surface and ubiquitin interaction of the DNMT3A1 N-terminal region shown in Map 3 and do not draw strong conclusions from this map beyond the co-incidence of DNMT3A1-3L features.

- The paper starts with suggesting that the N-terminal region in DNMT3A1 (aa 1-227) is critical for nucleosome binding (more important than the PWWP). A control with just DNA should be included in Figure 1B (these controls are run in Figure 2 but the 1-227 construct is not included). While the structure disproves that this construct binds DNA, a DNA control is critical to trust all the EMSA that are used in this work. Construct 1-227 should also be included in the EMSA in Figure 4A, 5C and 5D, to confirm the authors statements.

This is a good point as we did see relatively high DNA binding affinity mapped to the N-terminal region (old figure 2), as reported previously (Suetake et al., Biochem J 2011). Indeed, based on this observation, all binding assays (EMSA and Pulldown) were performed in the presence of competitor DNA in a manner analogous to adding BSA as a background eliminating/crowding agent. We have made this observation more prominent; showing DNA binding of the full length complex (new figure 1B), the N-terminal PWWP construct (Figure S4C), and 1-277 fragment as suggested to H2AK119ub, H3K36me2, acidic patch mutated and unmodified nucleosomes (figure 4A).

We also note that the PWWP is highly additive and essential for H3K36me2 mediated interactions explaining the in vivo recruitment of DNMT3A (old figure 4 and new figure 4A, new supp fig S8G).

Similarly, Figure 1D requires DNA as control, and the 1-227 construct should be included. This will clarify the performance of these pull-down experiments, which not always align with the EMSA experiments (e.g. binding to unmodified nucleosomes).

Old Figure 1D was performed with competitor DNA present so we would not expect any binding to be apparent to DNA alone. However, we have included DNA binding and the 1-277 construct (fig S4C, fig 4A) as suggested.

The pulldown assays are complementary and consistent to the EMSA assays, but overall less informative. Pulldown experiments are performed at a single concentration point with washing amplifying the effect of koff, whereas EMSA's are across a broad concentration range. By using higher amounts of nucleosome/DNMT3A in pull-downs we do see binding to unmodified nucleosomes (Reviewer Figure 1) but due to the binary nature of the signal differences due to modification state are lower (effectively saturating signal as in higher concentrations in EMSAs,).

Reviewer figure 1: comparison of pull-down assay conditions using increasing amounts of DNMT3A and nucleosome (right).

- The role of the acidic patch is critical, Figure S2C should be moved to main to make this point very obvious from the beginning. Moreover, in the experiment in Figure 5C, binding to the acidic patch mutant nucleosome (without K119Ub) is missing and should be included.

We agree this is an essential result and have now moved this experiment (old Figure S2C) is now Fig 4E. The acidic patch is clearly involved in binding both within the context of an unmodified nucleosomes, H3K36me2 nucleosome and H2AK119ub nucleosome and is present in the revised manuscript (Fig 4D, 4E, 2C respectively).

- The authors quantify their binding experiments and activity assay by extrapolating binding curves, but no quantitative values are included (apparent K_d and K_{cat}/K_m). This would be helpful in comparing the different assays and the many figures. It would also help to estimate actual effects of some of the mutations/deletions. Could the authors include these and perhaps a table that summarizes all the binding and activity observations, so to aid into the interpretations?

We agree and have included this where relevant in the revised manuscript Figures and a complete Table S2. As suggested Apparent K_d values have been added to the main figures rather than absolute numbers in part due to the presence of competitor DNA allowing comparison between experiments.

We chose not to report K_{cat} , V_{max} and K_m values to avoid suggesting kinetic characteristics for DNMT3A1. for a few reasons:

- DNMT3A1 is a bi-substrate enzyme including two substrates: SAM and DNA. To determine V_{max} and K_m values more accurately, it would be required to change the concentrations of both substrates and determine the mechanism of action. As in our assays neither SAM, DNMT3A1 or the DNA sequence change and this requires a lot of material, we decided to use our assay comparatively, but not report V_{max} or K_m values
- Due to limited amounts of nucleosomes, we do not reach full V_{max} (requiring more material than we could feasibly make), making estimations of the V_{max} and K_m values uncertain.

3. The Mtase Glo kit by Promega uses two additional enzymatic steps. Although these are in excess (activity is linear far beyond the concentrations we present), we prefer to be careful with presenting values that can be interpreted as enzyme specific.
- As the authors state, the binding and activity data are not aligning. The authors should include activity assays with DNMT3A1-L mutants (e.g. deletion/mutations on the UDR or PWWP), to at least test if the UDR is important for activity at all.

This is an important conclusion and one we address through using a version of the DNMT3A protein that lacks the UDR region (old figure 7B, currently figure 6B) and through performing activity assays on acidic patch mutated nucleosomes. Both of these have no measurable influence on enzymatic activity, but do have an effect on recruitment. We therefore conclude that that the UDR is not involved in direct catalytic activity in vitro. We are still interested in understanding this disconnect and believe it may be due to enzyme processivity

Additional Revision experiment: mapping DNA methylation position and quantity across the in vitro methyltransferases assays using sequencing based approaches

We therefore conclude that increased recruitment rather than an increased activity is responsible for hypermethylation observed in disease and ageing states on H2AK119ub marked chromatin. We have expanded the discussion of this point in the manuscript.

- The final model where the authors suggest different structural conformations depending on the histones PTMs is intriguing, but I don't see how this is supported by the data directly. For example, there is no evidence that the UDR is not engaging the acidic patch when the nucleosome are H3K36me2. It's quite the opposite actually, as shown already in Figure 1. Therefore, the model does not match the data and needs to be revisited.

We agree this is an important point and was the conclusion we were trying to suggest, the UDR region is important in both K119ub and H3K36me2 binding modes. We aimed to show this with an arrow going from the UDR to the acidic patch concurrently with arrows to the H3K36me2 but agree this was unclear. We have modified this figure to make this point clearer as well as including a new section in the manuscript as this was an important conclusion from the paper.

Some experiments have been performed once (e.g. stated in Figure legends 5E). Replicates should be included during revision.

This was a typographical error, all experiments were repeated prior to submission. Further replicates have now also been included.

- Regarding the crosslinking experiments: no intra-molecular crosslinks are shown in Figure 2A and S2B. This seems unlikely, so it would be helpful to share the complete results of these crosslinking mass spec experiments.

For the sake of clarity, we removed these from the representation, but agree it is good to show them for experts. We have re-added these in figure S8F.

All crosslinking data has been deposited and is available at the PRIDE database under code PXD046529.

- Finally, the story could benefit from some streamlining, both at the text and figures level, to get the key messages out more clearly.

We thank the reviewer for this comment and agree. As suggested we have restructured the paper with more emphasis on the H2AK119ub-DNMT3A1 structure, interaction followed by the H3K36me2 and enzymology which has greatly improved readability

Referee #2:

The authors report the identification and analysis of novel histone binding capabilities of the N-terminal part of DNMT3A1. The paper reports a large piece of technically demanding data, but it lacks novelty, because key literature findings have not been cited and considered accordingly. Moreover, several findings are not in good agreement with literature data. Hence, I cannot recommend publication of this work in EMBO J. and do not see that a revision could mitigate these problems. In my assessment, this paper needs to be completely rewritten, considering (and describing) all relevant literature. Then, findings must be critically discussed in the light of the literature data. Based on the outcome of this, the paper may be publishable, but I do not see it in a high-end journal.

We regret that the reviewer took umbrage at our representation of our data and in retrospect we agree our choice of wording was poor at points. It was unclear what we were stating as novel and what was exciting and new about our manuscript. We however strongly disagree with the comments of the reviewer. We had fully cited, credited and considered the literature referenced by the reviewer throughout the manuscript. These were cited throughout, but we purposefully left the discussion of our results to the appropriate Discussion section of the manuscript.

We have rewritten and re-ordered the manuscript with extensive consideration to avoid a similar miscommunication with the audience to better reflect and highlight the novelty of our study.

Comments:

1) The lack of novelty is illustrated already in the abstract but reflected also in the detailed description of results:

"Using a combination of biochemical and structural approaches we find that DNMT3A interacts using multiple interfaces with chromatin; directly binding generic nucleosome features as well as site-specific post-translational histone modifications." - This claim is not novel, as the chromatin interaction mediated by the ADD and PWWP domains are well known since more than a decade.

We were not intending to suggest multivalent chromatin interaction of DNMT3A has not been previously reported. Of course the PWWP and ADD domain recognise specific genomic features (H3K36me2/3 & H3K4me0 respectively, outlined in the introduction section). We were mistaken for not explicitly naming the generic nucleosome feature in the abstract: we were referring to the nucleosome acidic patch and H3-H2A interface not previously reported to be engaged by DNMT3A1. This has now been amended.

"The N-terminal region, unique to the DNMT3A1 isoform, is essential for these interactions and stabilises H3K36me2-nucleosome recruitment. Intriguingly, in the same region critical for nucleosome binding we also map a ubiquitylation-dependent recruitment motif (UDR). The UDR binds specifically to ubiquitylated H2AK119, explaining the previously observed recruitment to Polycomb-occupied heterochromatin." - This part rediscovers published findings reported in Weinberg et al. 2021 Nat Genet 53(6):794-800 and Gu et al. 2022 Nat Genet 54(5):625-636, which are not appropriately mentioned.

This is not a fair representation of our manuscript. These excellent studies have been referred to in the introduction to introduce the importance of the N terminal region for recruitment to chromatin regions marked by H2AK119ub. They were also mentioned in the results section under "The N-

terminal region of DNMT3A1 binds to H2A Lys119ub within a nucleosome”, “Structure of DNMT3A1-DNMT3L bound to a H2AKc119ub-nucleosome reveals DNMT3A1 interactions on the nucleosome surface” and discussed extensively throughout, but most specifically in the discussion section (referred to 22 times throughout the original manuscript).

In our manuscript we have biochemically mapped the region that interacts with K119ub nucleosomes further than these studies and also show that this is both necessary and sufficient for recruitment. We have been working on this problem since the discovery by Edinburgh colleagues of relocalisation of DNMT3A to polycomb regions (Heyn et al 2019), independently finding direct DNMT3A1-H2AK119ub-nucleosome interaction in 2019. We were unaware of Weinberg et al. 2021 and Gu et al. 2022 until their publication and presented our data from the historical perspective in which we performed the experiments.

We have since re-written the manuscript to better highlight the novelty. Importantly, we show that this same region in addition to polycomb recognition aids H3K36me2 interaction as well and present a structure explaining molecular details of this interaction on nucleosomes. The observation on H3K36me2 is entirely novel as too was the structure and we extend the results proposed by Gu et al., and Weinberg et al., extensively. We have amended the abstract, introduction and discussion to make these points even more explicit.

Together, these parts comprise 2/3 of the entire abstract.

This problem is also already appearing in the title, which makes statements one can find in numerous reviews on DNMT3A written after 2015.

We have refined the abstract and title to better represent the key conclusion from our paper to:

The N-terminal region of DNMT3A engages the nucleosome surface to guide recruitment.

2) In the first chapter of the results, the authors report that the DNMT3A1(1-427) region is essential for chromatin engagement and H3K36me2/3 binding. This claim is not in agreement with data reported in Weinberg et al. 2019, showing that the isolated PWWP domain specifically binds H3K36me2/3 containing nucleosomes and that DNMT3A2 expressed in cells delivers DNA methylation to chromatin regions containing H3K36me2.

We agree the PWWP is clearly important as reported by many studies (Weinburg 2019; Heyn 2019; Sendzikate 2019; Kibe 2021, Lue et al., 2023 and many others) and indeed our own discussed results. We showed HESJAS mutations that ablate histone methyl recognition greatly reduce binding (original supp fig s1E). We were not intending to suggest otherwise, simply that PWWP-H3K36me2 interactions are stabilised by additional interactions DNMT3A1 including the UDR region.

We do see preferential binding of just the PWWP to nucleosomes in the absence of competitor DNA (now included as Fig S8G). We have changed the way this is discussed in the manuscript to better reflect this.

3) In the second chapter of the results (starting on p. 7), the author study by XL analysis the interaction of DNMT3A1(1-427) with nucleosomes and observe an interaction with the acidic patch of nucleosomes. However, the information that DNMT3A/3B3 heterotetrameric complexes bind to the acidic patch is available in the literature since 2020, when a cryo-EM paper reporting this finding

was published (Xu et al., Nature 586(7827):151-155). Again, this important part of literature is not mentioned. However, the authors mapped the acidic patch interaction to the N-terminal part, while Xu et al. show the ability of the C-terminal domain of DNMT3B3. This discrepancy can be explained because the authors used the DNMT3A1(1-427) fragment in the XL analysis. Hence, it remains open, which of these two interaction modes is more important under which condition.

Xu et al., 2020 was cited in the introduction and results section and the structure was discussed in the discussion. There is no discrepancy in our study: DNMT3A2-DNMT3B3 lacks the N-terminal region which is the main focus of our study. Throughout all observations on DNMT3A1(1-427) fragment are also validated for full length DNMT3A1-DNMT3L (lacking DNMT3B3 acidic patch interactor). The structure published by Xu et al. is obviously relevant, we originally discussed this in the discussion section when comparing our structure and directly addressing the comment above on how these two acidic patch binding modules may integrate and fascinating area of future study, we have now moved this to earlier in the manuscript. We note that there would be two acidic patch regions on either face of one nucleosome so it is conceivable that both could be engaged, one by DNMT3B3 one by DNMT3A1. We are in the process of expanding to include DNMT3B3 in our analysis

Additional Revision experiment: We have co-expressed and purified DNMT3A1-DNMT3B3 complex and will perform relevant enzymatic assays and binding assays (reviewer Figure 2 below) and we are awaiting full characterisation of tetrameric nature of this complex.

4) In the next results chapter (starting on p. 8), the authors studied methylation of nucleosomal DNA. Again, similar studies with better CpG site resolution have already been carried out, e.g. Bröhm et al. Commun Biol. 2022 5(1):192.

The Bröhm study is a very nice approach that we cited throughout. However, it was performed using murine DNMT3A2 at a single concentration point. Given the focus of our manuscript was on the role of multiple post-translational modifications (not reported in Bröhm) that interact with an N-terminal region that are lacking from DNMT3A2 the studies are very different. Our assay are performed across a range of substrate concentrations. We also utilised a greater array of nucleosome substrates including closer to native chemistry and this appears to be important based on the additional activity over the methyl-lysine analogues used previously (old figure 7C, now Figure 6C). This therefore provides a great deal of new insight beyond what has been previously reported.

We agree, better CpG resolution would be informative and we are in the process of analysing sequencing data to probe site(s) of methylation and understand the processivity of DNMT3A1 in our assays.

Additional Revision experiment: methyl-sequencing of our in vitro methylation reactions to determine the site(s) and relative abundance of DNA methylation on differently modified nucleosomes.

It is unclear in Fig. 3C, why the DNA alone was not methylated at all, suggesting something went wrong with the experiment.

The observation of low activity on DNA alone is entirely expected and consistent with the literature. This is due to the auto-inhibitory nature of the ADD domain of DNMT3A1 catalytic activity, which is alleviated upon engagement of the H3 tail (Guo et al., 2015; Li et al, 2011; Zhang et al, 2010) and referenced in the original manuscript. We indeed see significantly more activity on nucleosomes over DNA alone, but this is reduced in nucleosomes lacking the N-terminal tail in line with the H3 tail activation, which was discussed in the manuscript in results section “H3K36me2 stimulates DNMT3A1-DNMT3L complex activity on nucleosomes”. To make this more explicit we have now included our initial control experiments. DNA alone can of course be methylated, but this required far higher concentration of enzyme, way beyond the dynamic range of activity we see on chromatinized templates (now added in Fig S10D). We also used H3 peptides in trans to observe activation as has been previously reported (now added in Fig S11D).

They later conclude that DNMT3A1-DNMT3L can engage linker DNA, which again was not novel (see for example Brohm et al.).

In no way did we mean to suggest linker DNA engagement was novel we reported this when discussing our structure, in a similar vein to the overall structure of nucleosome is not novel we were simply explaining the context of our results.

5) The cryo-EM of the ubiquitylated nucleosome presented in Fig. 4 and 5 is novel, but to me the interaction is not very convincing (see for example Fig. 4C). More experiments need to be conducted testing the relevance of critical residues to validate this structural model.

It is unclear what further validation the reviewer would require to be convinced:

- The main map (Map 3) is at sufficient resolution to allow unambiguous modelling into the density and determine sequence register. Old Figure 4C (Map1) is a lower resolution map and we do not draw any conclusions based on this data beyond the presence of additional density concurrent with density on the nucleosome surface.
- We have made point mutations on DNMT3A1 in this region that ablate binding to both H3K36me2 and H2AK119ub nucleosomes.
- Conversely we have made point mutations on the nucleosome that ablate binding to DNMT3A1-427 (old figures 2C, 5C S2C; new figures 4D, 4A and 4E) and DNMT3A1-DNMT3L complex (Fig S8C). Crosslinking mass spectrometry corroborates the structure showing clear crosslinks between the region of interest in DNMT3A and H3K36me2 modified nucleosomes. We also show that DNMT3A1 shows less affinity for H2A.Z, which is in agreement to the observed interaction of the UDR region with the interface between the H2A and H2B dimer, where H2A.Z is divergent.
- We note our structure is in line with two others using a crosslinked smaller fragment of DNMT3A1 on H2AK119ub nucleosome core particle that have subsequently become available after our submission (Gretarsson et al., bioRxiv 2024; Chen et al., *Nature communications* 2024)

6) Finally, the authors report that the H2AK119ub interaction does not stimulate DNMT3A activity. However, based on their own observations hinting towards a close connection of H2AK119ub and H3K36me2 binding, a nucleosome carrying the double mark, would have been needed, which was not used. Instead, the H3K27me3/K36me3 double mark was tested, although it was shown before that H3K27me3 does not have an effect.

The experiment using a double mark is a good suggestion, especially given the co-occurrence of these marks (Weinberg 2021; Conway et al, 2021; Fursova et al, 2019; Fursova et al, 2021). We have performed binding and enzymology assays using nucleosomes containing both H3K36me2 and H2AK119ub (Fig 5C/D, 6D,E). We see enzymatic activity in these nucleosomes is similar to singly modified H3K36me2, suggesting that H2AK119ub is not in itself reducing catalytic activity.

Additional Revision experiment: methyl-sequencing of H3K36me2/H2AK119ub in vitro methylated nucleosome and comparison to other modified nucleosomes to test CpG site usage.

Referee #3:

The manuscript outlines the potential functionalities of the DNMT3A1 N-terminal region (before the PWWP domain). This includes the binding of the nucleosome acidic patch and/or binding of ubiquitylated H2AK119 (as reported already by two recent studies). The N-terminal region, in conjunction with the PWWP domain's H3K36me2 binding, could guide DNMT3A1 to chromatin domains marked with intergenic H3K36me2 or heterochromatin H2AK119ub. In addition, the authors demonstrated that DNMT3A1 activity could be stimulated by H3K36me2 but not by H2AK119ub (despite strong binding). The study comprises two parts: in vitro binding/activity assays using modified reconstituted nucleosomes and a cryo-EM structural study of DNMT3A1-DNMT3L in complex with ubiquitylated nucleosomes.

We thank the reviewer for the positive appraisal and summary

A critical concern regarding experimental design is the choice of using DNMT3L in complex with DNMT3A1. Since DNMT3L primarily functions in the germline and is silenced upon differentiation, whereas DNMT3A1 is the predominant somatic isoform, the use of DNMT3A1-DNMT3L may be an artifact. The authors should provide a justification for choosing 3A1-3L over DNMT3A1-3B3, for instance. Additionally, one needs to consider the competitive binding of 3A1 and 3B3 to the same acidic patch, as demonstrated by a recent 3A2-3B3 structure.

This is an important concern and one we did not address fully in the initial submission and we have added extensive discussion in the revised manuscript (page 3). We chose DNMT3L as the activating substrate adaptor originally a technical necessity as we struggled to express and co-purify DNMT3A1 with DNMT3B3. We agree using a DNMT3A1-DNMT3B3 would hold more direct relevance and we have since been successful with optimised constructs to produce full-length DNMT3A1-DNMT3B3 complex (see reviewer figure 2 below).

We do note that DNMT3L appears to be present with DNMT3A1 in some select tissues. Furthermore for our experiments we find using DNMT3L as an accessory factor to ensure activity without adding another nucleosome interacting region that may confuse our assays. DNMT3B3 and DNMT3A1 acidic patch interaction, while mutually exclusive on a single nucleosome face, would not be so in the context of two acidic patches per full nucleosome. As such comparing 3L and 3B3 will be interesting and we are excited to test the binding and enzymology in the revised manuscript.

Reviewer Figure 2: **A** schematic of full-length DNMT3A1-DNMT3B3 proteins. **B** Co-purification full-length DNMT3A1 with full-length DNMT3B3, final SEC trace. Full-length His-MBP-DNMT3A1 was co-expressed with full-length His-GFP-DNMT3B3, co-lysed and purified on a Ni-NTA column (5 mL, Cytiva). His-MBP and His-GFP tags were cleaved using TEV protease and removed using Ni-NTA column (5 mL, Cytiva). The protein complex was further purified using Heparin HP (5 mL, Cytiva) and size-exclusion chromatography (Superdex S200 increase 10/300 gl, 24 mL, Cytiva). **C.** SDS PAGE gel of final protein complex. additional bands correspond to DNMT degradation products. **D.** Methyltransferase activity of full-length DNMT3A1-DNMT3B3 on unmodified and H3K36me2 nucleosomes wrapped with 195bp Widom601 DNA and free 195bp Widom601 DNA. Experiment done in duplicate.

Additional Revision experiments: we are validating the oligomerisation and binding DNMT3A1-DNMT3B3 complex and plan to perform relevant enzymatic assays using this. Preliminary results are promising (reviewer Figure 1) and we are awaiting full characterisation of tetrameric nature of this complex.

The cryo-EM structure of 3A1-3L raises concern about its poor quality, lacking representation of any tetrameric features of 3A-3L. To address this, I recommend focusing on the structure of the N-terminal fragment (residues 1-427) in complex with the nucleosome (with either modifications of H3K36me2 or H2Ak119ub), considering that all binding data were obtained with this fragment. Currently, there is a disconnect between the binding data and structural data in the manuscript. For instance, Figure 4A (and 4E) presents binding data with His-MBP-DNMT3A1 fragments, while the corresponding structures are depicted as DNMT3A1-DNMT3L. Focusing on the same fragment will enhance the manuscript's coherence and strengthen the overall argument.

Based on this comment and those of reviewer 1 we have shifted the focus of the revised manuscript more onto the DNMT3A1 N-terminal region, which we believe helps with the overall readability. Our map and model is high quality and we are confident the lower resolution observed for additional DNMT3A1-DNMT3L regions is biological rather than to do with technical limitations of the data, caused by structural heterogeneity inherent in the complex. We included the low-resolution structure of DNMT3A1-DNMT3L more to show our pipeline of investigation and confirm that the additional density over the acidic patch was concurrent with additional density on the linker DNA.

We have moved the lower-resolution structure of the full complex to supplementary figure S1F and include representations of this structure at multiple threshold levels (supp fig S1E). As shown, this did a disservice to the data, at lower thresholds the structure clearly shows features such as turn of DNA and separation between secondary structure elements in alpha helical octamer core indicative of the resolution we report. While the nucleosome core is overall at good resolution, the large mass of the nucleosome drives alignment during data processing. Due to the nature of averaging in single particle cryo-EM processing we often observe a range of resolutions indicative of flexibility across a map. Extensive structural heterogeneity of the untethered DNA linker combined with multiple possible DNMT conformations on substrate DNA leads to overall lower resolution represented in the local resolution estimation in old supp fig 5 new supp fig 2 (bottom left).

We did attempt structural studies using smaller fragments of DNMT3A1, but these were unsuccessful we believe due to the additional stabilising interactions from the ADD and methyltransferase domain present in the full-length protein in our study. Indeed we observed that the full length DNMT3A1-3L complex (Fig 1B and supp fig S8C) binds with higher affinity, but displays the same overall nucleosome binding patterns as NT-PWWP fragments. We note that a similar article that reports a complementary structure has recently been deposited in bioRxiv (Gretarsson et al., bioRxiv 2024) and has been published (Chen et al., *Nature communications* 2024). These use a short fragment of DNMT3A1 but required cross-linker to observe the density, not required in our structure.

As commented the tetramerization is in catalytic domains, not the region of interest in this manuscript. Nevertheless, it was important for us to use a well characterised and high-quality complex for our studies. DNMT3A1 and DNMT3L are equimolar and co-purified using a tandem affinity purification strategy with a large hydrodynamic radius indicative of tetrameric assembly (old supp fig S 3A & B, current fig S1C & D). We obtained 2D classes that were reminiscent of DNMT3A1-DNMT3L tetramer in isolation in the DNMT3A1-DNMT3L NCPH2AK119ub data, presumably from free DNMT complex not interacting with nucleosomes (reviewer figure 3, added to supplemental). Multiple views could be obtained that match well with projections generated from tetramer crystal structures, but not single DNMT3A1 or DNMT3A1-DNMT3L dimers. These are also reminiscent of DNMT3A2-DNMT3B3 negative stain class (Zeng, Ren & Kaur et al 2020 genes dev), which thoroughly characterised these complexes and inspired our work. Multiple bi-lobed orientations on the linker region that only match DNMT3-DNMT3L tetramers was also observed for the DNMT3A1-DNMT3L di-nucleosome-H3K36me2 data (old figure 3D and projection matched representations in reviewer figure 3). We are conscious to not overinterpret projection matching but this data and biochemistry suggests that we have purified a predominantly tetrameric of DNMT3A1-DNMT3L complex.

Reviewer Figure 3: Comparison of tetrameric DNMT3A1-DNMT3L crystal structure projections with 2D classes from our cryo-EM dataset. Crystal structure projections were made of two copies of PDB 4u7p, each containing DNMT3A1 catalytic domain and ADD domain and DNMT3L catalytic domain. To ensure correct tetramer orientation, PDB 4u7p was aligned with PDB 5yx2, containing tetrameric DNMT3A1-3L catalytic domain. The tetrameric model was then converted to a volume using UCSF Chimera 1.16. The volume was imported into Cryosparc v4.4.1 and converted to projections by generating templates. Templates were compared with 2D classes of protein-only particles found in the dataset of DNMT3A1-3L on H2AK119ub nucleosomes.

Reviewer Figure 4: Comparison of tetrameric DNMT3A1-DNMT3L crystal structure projections with 2D classes from cryo-EM dataset of H3Kc36me2 di-nucleosomes with DNMT3A1-DNMT3L. Crystal structure projections were made by combining PDB 4u7t, containing DNMT3A1 catalytic domain and ADD domain, DNMT3L catalytic domain and an H3 tail peptide with PDB 5yx2, containing tetrameric DNMT3A1-3L catalytic domain bound to DNA. The combined PDB model was then converted to a volume using UCSF Chimera 1.16. The volume was imported into Cryosparc v4.4.1 and converted to projections by

generating templates. Templates were compared with 2D classes of a dataset of H3Kc36me2 modified di-nucleosomes (DNA = Widom601 + Widom603 + 30 bp spacer) with DNMT3A1-DNMT3L as purified in the manuscript.

Other concerns:

"6xHis-MBP tagged fragments" - a positive control should be included for untagged fragment

We have included this control in figures S4D, S8B and see no affect due to the tag, beyond greater stabilisation overall of the protein fragments. We note we see similar binning patterns for the untagged DNMT3A1-DNMT3L complexes.

"PWWP domain alone bound poorly, likely due to the low affinity of the interaction with the histone mark ... and affinity for competitor DNA used in the assay buffer" - Does the PWWP domain bind nucleosome in the absence of competitor DNA?

We have performed this assay as suggested and see preferential binding of the PWWP in isolation to H3K36me2 modified nucleosomes alone (New Fig S8E-G), but as reported previously its binding is weak compared to larger fragments.

In streamlining the article we have removed this initial assay (old figure 1B) as we found it to be distracting to the overall message, but our results are in agreement with the PWWP driving recruitment to H3K36me2.

Page 7, line 3, The N-terminal region alone showed considerable interaction with nucleosomes irrespective of their modification state (Fig 1A), - wrong figure panel?

Thank you for spotting this, it has been rectified.

Dear Marcus,

Thank you for the transfer of your manuscript and point-by-point response to EMBO reports. I have now looked at all files and discussed all with my colleagues here, including our chief editor.

I am sorry to say that we are not convinced that the manuscript does provide a sufficient advance over the recently published papers for publication here. It would be very helpful to see the revised version of your ms for a more direct comparison. What I can offer is that you submit the fully revised manuscript as soon as possible and I will discuss the advance over the published papers with one of the referees. In case we cannot offer publication in EMBO reports, I am certain that we can find a solution with our sister journal Life Science Alliance.

While we encourage authors to post their work on preprint servers, and we also offer scooping protection from the day of posting, the scooping protection only applies to a reasonable timeframe, which is around 4 months after ms posting. Given that you posted your work already one year ago, and that Hartmut invited a point-by-point response from you already in March this year, I am afraid that a full scooping protection does not apply in this case anymore.

I am sorry that I cannot be more positive at this point, but I am happy to discuss your revised ms with one of the referees, most likely referee 1, for a final decision.

Kind regards,
Esther

To submit your manuscript, please follow this link.

Link Not Available

Response to reviewers comments Wapenaar et al., The N-terminal region of DNMT3A combines multiple chromatin reading motifs to guide recruitment.

Referee #1:

The article from Wapenaar et al. presents biochemical and structural analysis of the interaction between DNMT3A1 and the nucleosome. The authors use EMSA, pulldowns and SAH activity assays together with cryo-EM analysis to put forward a model in which they identify a UDR domain in the N-terminus of DNMT3A1 that binds the nucleosomal surface and favors ubiquitinated nucleosomes. The identification of a new interface between DNMT3A1 and nucleosomes is of high impact and interest. While this is addressed structurally and biochemically, I have some concerns that should be addressed before publication.

We thank the reviewer for their positive comments and appraisal.

The structure is a strong piece of evidence for this new interaction, and it is somewhat a pity that it is presented relatively late in the manuscript. Some clarifications could help:

We agree and based on comments by this and other reviewers have streamlined the manuscript, changing the order in which the results are presented. We now present the structure in the first figure and discuss this interaction with H2AK119ub before moving onto implications for H3K36me2 and activity.

- Fig 4D: Please show the fit of the PDB within the EM density for the DNMT3A1 peptide.

the full fit of the modelled sequence is now shown in Supplementary 3H (expansion of Supplementary 6H)

- The ubiquitin density isn't resolved at high resolution. It is therefore not possible to determine based on these results whether this is ubiquitin. What is the evidence that this density is ubiquitin, rather than e.g. the PWWP or anything else? This should be addressed by crosslinking mass spec or comparable approaches.

The reviewer makes a good point, we ascribed this density to ubiquitin and the ubiquitin contacting UDR. This was based on continuous density stemming from the tail of H2A at position 119 as expected and the UDR fragment of DNMT3A projecting towards this (old Figure 6A, now Figure 3A). We have amended the text to make the uncertainty of the annotation of this density clearer. As suggested, we have performed cross-linking mass spec for H2AK119ub nucleosome which shows crosslinks between DNMT3A1 N-terminal region and the proposed interaction site on the nucleosome surface and to the ubiquitin. No crosslinks were observed between ubiquitin and the PWWP. This is shown in new Supplemental Fig S1H (DNMT3A1-DNMT3L full complex) and S6A (DNMT3A1 N-terminal fragment).

Furthermore, biochemically we found the UDR region DNMT3A1¹⁶⁴⁻²¹⁹ (old figure 4E, new figure 1E) and the longer construct DNMT3A1¹⁻²⁷⁷ (adapted figure 4A) is sufficient for this interaction. We note that two complementary studies have become available since our initial submission (Gretarsson et al., *Science Advances* 2024; Chen et al., *Nature communications* 2024). These both determine the structure of just a short fragment of DNMT3A1 (with no other domains present) on H2AK119ub nucleosomes in the presence of crosslinking agent. The density assigned to ubiquitin in both these studies is similarly localised to the density we described as ubiquitin.

- Fig. S6. It's not always clear which maps these figures belong too. The euler angle plot of the other maps are missing. In addition, the particles still seem to have a preferential orientation. What do the authors think the effect of this could be on the final map? It seems like especially the DNMT3A1 containing part is overrepresented, could this have an effect on the extra density visible here?

We apologise that this was not clear. We have relabelled these plots and the figure legends to clarify in new Figure S3. We have also included the Euler plots for the other maps in Fig S2.

We were concerned about preferred orientation given the issue in the H3K36me2-DNMT3A1-3L data. However, this does not cause significant affects for the H2AK119ub-DNMT3A-3L data (Map 3), with complete overall angular coverage in the Euler plots. We do not see any observable anisotropy in the maps and 3DFSC plot shows good overlap of directional FSC and global FSC (Fig S3F).

In addition, we have now included more quantitative measures of sample distribution in supplemental Table S1. The measured sphericity for 3D-FSC is 0.966 (out of 1). The sample compensation factor (SCF) is 0.94 (out of 1) (greater than 0.81 considered to be no significant anisotropy) (Baldwin, P. R., & Lyumkis, D. (2020)).

The DNMT3A1 fragment is slightly lower resolution than the adjacent histones octamer (Supp Fig S2), which we believe explains the difference in representation seen at higher threshold (i.e overrepresentation). We do not believe this is due to masking as density is similar in looser masked Map 2.

- Could the authors add images of the initial models and provide information about the low-pass filtering of these classes?

This has been included in figure S2 (previously S6). Initial models were derived from the data using cryoSPARC *ab initio* approach. For the relion map, an initial cryoSPARC *ab initio* model of a subset of the data was used and low pass filtered to 50Å.

- For dataset 1, a 3D refinement in RELION using a B-factor of -50 is performed. However, with datasets below 10Å, usually an automated B-factor estimation is preferred, to prevent the amplification of noise (Rosenthal & Henderson 2003). Could the authors clarify this choice?

This map was to show our working methodology and confirm some additional density outside of the nucleosome was present in the data to show we were on the correct track.

Automated B factor assignment from linear fitting to Guinier fitting gave an estimated b factor of -203. Global sharpening B factor for this map is not the best approximation for this data given the large resolution range across the map (supp fig old s4 new S2) due to the structural flexibility observed. For this reason we used a lower global B factor to under sharpen the low-resolution features, leading to loss of interpretable detail rather than over sharpen, to avoid the interpretation and amplification of map noise.

- Map 1: why did the authors choose a contour level of 0.0638? This seems quite high. Even though an extra density becomes visible this quite noisy and difficult to interpret.

We agree this is a high threshold, we thought it appropriate to display the additional non-nucleosomal density present in the data.

We have now included Supplemental Figure S1F, showing this map at three representative threshold levels with no sharpening applied. The high structural heterogeneity in the data leads to delocalisation of the DNMT3A1-3L signal due to particle averaging. This is expected as the nucleosome tethering elements of the structure contain long flexible linkers, while the DNA methyltransferase we believe can engage in multiple orientations on the flexible linker DNA. The lower threshold clearly shows the nucleosome and linker DNA projecting away from the nucleosome core. Flexible DNMT3A and ubiquitin densities are only observable at the higher thresholds. We mainly focus on the nucleosome surface and ubiquitin interaction of the DNMT3A1 N-terminal region shown in Map 3 and do not draw strong conclusions from this map beyond the co-incidence of DNMT3A1-3L features.

- The paper starts with suggesting that the N-terminal region in DNMT3A1 (aa 1-227) is critical for nucleosome binding (more important than the PWWP). A control with just DNA should be included in Figure 1B (these controls are run in Figure 2 but the 1-227 construct is not included). While the structure disproves that this construct binds DNA, a DNA control is critical to trust all the EMSA that are used in this work. Construct 1-227 should also be included in the EMSA in Figure 4A, 5C and 5D, to confirm the authors statements.

This is a good point as we did see relatively high DNA binding affinity mapped to the N-terminal region (old figure 2), as reported previously (Suetake et al., Biochem J 2011). Indeed, based on this observation, all binding assays (EMSA and Pulldown) were performed in the presence of competitor DNA in a manner analogous to adding BSA as a background eliminating/crowding agent.

We have made this observation more prominent; showing DNA binding of the full length complex (new figure 1B), the N-terminal PWWP construct (Figure S4C), and 1-277 fragment as suggested to H2AK119ub, H3K36me2, acidic patch mutated and unmodified nucleosomes (figure 4A).

We also note that the PWWP is highly additive and essential for H3K36me2 mediated interactions explaining the *in vivo* recruitment of DNMT3A (old figure 4 and new figure 4A, new supp fig S8G).

Similarly, Figure 1D requires DNA as control, and the 1-227 construct should be included. This will clarify the performance of these pull-down experiments, which not always align with the EMSA experiments (e.g. binding to unmodified nucleosomes).

Old Figure 1D was performed with competitor DNA present so we would not expect any binding to be apparent to DNA alone. However, we have included DNA binding and the 1-277 construct (fig S4C) as suggested.

One major conclusion from our work, that we have made more prominent, is that the UDR region binds extensively to the nucleosome surface and increases overall affinity in both a K119ub and H3K36me2 dependent fashion. As such we would expect a degree of binding to unmodified nucleosomes, stimulated to more physiological relevant range by post translational marks.

The pulldown assays are complementary and consistent to the EMSA assays. However these are overall less informative. Pulldown experiments are performed at a single concentration point with washing amplifying the effect of koff, whereas EMSA's are across a broad concentration range. By using higher amounts of nucleosome/DNMT3A in pull-downs we do see binding to unmodified nucleosomes (Reviewer Figure 1) but due to the binary nature of the signal differences due to modification state are lower (effectively similar to saturating signal as in higher concentrations in EMSAs).

Reviewer figure 1: comparison of pull-down assay conditions using increasing amounts of DNMT3A and nucleosome (right).

- The role of the acidic patch is critical, Figure S2C should be moved to main to make this point very obvious from the beginning. Moreover, In the experiment in Figure 5C, binding to the acidic patch mutant nucleosome (without K119Ub) is missing and should be included.

We agree this is an essential result and have now moved this experiment (old Figure S2C) is now Fig 4E. The acidic patch is clearly involved in binding both within the context of an unmodified nucleosomes, H3K36me2 nucleosome and H2AK119ub nucleosome and is present in the revised manuscript (Fig 4A, 4D, 4E, 2C respectively).

- The authors quantify their binding experiments and activity assay by extrapolating binding curves, but no quantitative values are included (apparent K_d and K_{cat}/K_m). This would be helpful in comparing the different assays and the many figures. It would also help to estimate actual effects of some of the mutations/deletions. Could the authors include these and perhaps a table that summarizes all the binding and activity observations, so to aid into the interpretations?

We agree and have included this in the revised manuscript Figures and all K_d values together in one Table S2. As suggested Apparent K_d values have been added to the main figures rather than absolute numbers, in part due to the presence of competitor DNA in the assay.

We chose not to report K_{cat} , V_{max} and K_m values to avoid suggesting kinetic characteristics for DNMT3A1. for a few reasons:

- DNMT3A1 is a bi-substrate enzyme including two substrates: SAM and DNA. To determine V_{max} and K_m values more accurately, it would be required to change the concentrations of both substrates and determine the mechanism of action. As in our assays neither SAM, DNMT3A1 or the DNA sequence change and this requires a lot of material, we decided to use our assay comparatively, but not report V_{max} or K_m values
- Due to limited amounts of nucleosomes, we do not reach full V_{max} (requiring more material than we could feasibly make), making estimations of the V_{max} and K_m values uncertain.

3. The Mtase Glo kit by Promega uses two additional enzymatic steps. Although these are in excess (activity is linear far beyond the concentrations we present), we prefer to be careful with presenting values that can be interpreted as enzyme specific.
- As the authors state, the binding and activity data are not aligning. The authors should include activity assays with DNMT3A1-L mutants (e.g. deletion/mutations on the UDR or PWWP), to at least test if the UDR is important for activity at all.

This is an important conclusion and one we address through using a version of the DNMT3A protein that lacks the UDR region (old figure 7B, currently figure 6B) and through performing activity assays on acidic patch mutated nucleosomes. Both of these have no measurable influence on enzymatic activity, but do have an effect on recruitment. We therefore conclude that that the UDR is not involved in direct catalytic activity *in vitro*. We are still interested in understanding this disconnect and believe it may be due to enzyme processivity. We also include more extensive enzymology using diverse DNMT3A constructs, nucleosome & peptides (new Fig 6C, 6E, S11F) as well as nanopore sequencing data, which directly allows calling of DNA methylation, which follows a similar pattern to the SAM turnover assays (Fig 6F).

We therefore conclude that increased recruitment rather than an increased activity is responsible for hypermethylation observed in disease and ageing states on H2AK119ub marked chromatin. We have expanded the discussion of this point in the manuscript.

- The final model where the authors suggest different structural conformations depending on the histones PTMs is intriguing, but I don't see how this is supported by the data directly. For example, there is no evidence that the UDR is not engaging the acidic patch when the nucleosome are H3K36me2. It's quite the opposite actually, as shown already in Figure 1. Therefore, the model does not match the data and needs to be revisited.

We agree this is an important point and was the conclusion we were trying to suggest, the UDR region is important in both K119ub and H3K36me2 binding modes. We aimed to show this with an arrow going from the UDR to the acidic patch concurrently with arrows to the H3K36me2 but agree this was unclear. We have modified this figure to make this point clearer as well as including a new section in the manuscript as this was an important conclusion from the paper.

Some experiments have been performed once (e.g. stated in Figure legends 5E). Replicates should be included during revision.

This was a typographical error, all experiments were repeated prior to submission. Further replicates have now also been included.

- Regarding the crosslinking experiments: no intra-molecular crosslinks are shown in Figure 2A and S2B. This seems unlikely, so it would be helpful to share the complete results of these crosslinking mass spec experiments.

For the sake of clarity, we removed these from the representation, but agree it is good to show them for experts. We have re-added these in figure S8F.

All crosslinking data has been deposited and is available at the PRIDE database under code PXD046529 and PXD056554.

- Finally, the story could benefit from some streamlining, both at the text and figures level, to get the key messages out more clearly.

We thank the reviewer for this comment and agree. As suggested we have restructured the paper with more emphasis on the H2AK119ub-DNMT3A1 structure, interaction followed by the H3K36me2 and enzymology which has greatly improved readability

Referee #2:

The authors report the identification and analysis of novel histone binding capabilities of the N-terminal part of DNMT3A1. The paper reports a large piece of technically demanding data, but it lacks novelty, because key literature findings have not been cited and considered accordingly. Moreover, several findings are not in good agreement with literature data. Hence, I cannot recommend publication of this work in EMBO J. and do not see that a revision could mitigate these problems. In my assessment, this paper needs to be completely rewritten, considering (and describing) all relevant literature. Then, findings must be critically discussed in the light of the literature data. Based on the outcome of this, the paper may be publishable, but I do not see it in a high-end journal.

We regret that the reviewer took umbrage at our representation of our data and in retrospect we agree our choice of wording was poor at points. It was unclear what we were stating as novel and what was exciting and new about our manuscript. We however strongly disagree with the comments of the reviewer. We had fully cited, credited and considered the literature referenced by the reviewer throughout the manuscript. These were cited throughout, but we purposefully left the discussion of our results to the appropriate Discussion section of the manuscript.

We have rewritten and re-ordered the manuscript with extensive consideration to avoid a similar miscommunication with the audience to better reflect and highlight the novelty of our study.

Comments:

1) The lack of novelty is illustrated already in the abstract but reflected also in the detailed description of results:

"Using a combination of biochemical and structural approaches we find that DNMT3A interacts using multiple interfaces with chromatin; directly binding generic nucleosome features as well as site-specific post-translational histone modifications." - This claim is not novel, as the chromatin interaction mediated by the ADD and PWWP domains are well known since more than a decade.

We were not intending to suggest multivalent chromatin interaction of DNMT3A has not been previously reported. Of course the PWWP and ADD domain recognise specific genomic features (H3K36me2/3 & H3K4me0 respectively, outlined in the introduction section). We were mistaken for not explicitly naming the generic nucleosome feature in the abstract: we were referring to the nucleosome acidic patch and H3-H2A interface not previously reported to be engaged by DNMT3A1. This has now been amended.

"The N-terminal region, unique to the DNMT3A1 isoform, is essential for these interactions and stabilises H3K36me2-nucleosome recruitment. Intriguingly, in the same region critical for nucleosome binding we also map a ubiquitylation-dependent recruitment motif (UDR). The UDR binds specifically to ubiquitylated H2AK119, explaining the previously observed recruitment to Polycomb-occupied heterochromatin." - This part rediscovers published findings reported in Weinberg et al. 2021 Nat Genet 53(6):794-800 and Gu et al. 2022 Nat Genet 54(5):625-636, which are not appropriately mentioned.

This is not a fair representation of our manuscript. These excellent studies have been referred to in the introduction to introduce the importance of the N terminal region for recruitment to chromatin regions marked by H2AK119ub. They were also mentioned in the results section under "The N-

terminal region of DNMT3A1 binds to H2A Lys119ub within a nucleosome”, “Structure of DNMT3A1-DNMT3L bound to a H2AKc119ub-nucleosome reveals DNMT3A1 interactions on the nucleosome surface” and discussed extensively throughout, but most specifically in the discussion section (referred to 22 times throughout the original manuscript).

In our manuscript we have biochemically mapped the region that interacts with K119ub nucleosomes further than these studies and also show that this is both necessary and sufficient for recruitment. We have been working on this problem since the discovery by Edinburgh colleagues of relocalisation of DNMT3A to polycomb regions (Heyn et al 2019), independently finding direct DNMT3A1-H2AK119ub-nucleosome interaction in 2019. We were unaware of Weinberg et al. 2021 and Gu et al. 2022 until their publication and presented our data from the historical perspective in which we performed the experiments.

We have since re-written the manuscript to better highlight the novelty. Importantly, we show that in addition to Polycomb recognition, the same UDR region also engages with the nucleosome surface to aid H3K36me2 interaction as well and present a structure explaining molecular details of this interaction on nucleosomes. The observation on H3K36me2 is entirely novel as too was the structure and we extend the results proposed by Gu et al., and Weinberg et al., extensively. We have amended the abstract, introduction and discussion to make these points even more explicit.

Together, these parts comprise 2/3 of the entire abstract.

This problem is also already appearing in the title, which makes statements one can find in numerous reviews on DNMT3A written after 2015.

We have refined the abstract and title to better represent the key conclusion from our paper to:

The N-terminal region of DNMT3A engages the nucleosome surface to guide recruitment.

DNA methyltransferase 3A (DNMT3A) plays a critical role in establishing and maintaining DNA methylation patterns in vertebrates. Here we structurally and biochemically explore the interaction of DNMT3A1 with diverse modified nucleosomes indicative of different chromatin environments. A cryo-EM structure of the full-length DNMT3A1-DNMT3L complex with a H2AK119ub nucleosome reveals that the DNMT3A1 UDR motif interacts specifically with H2AK119ub as well as extensive contacts with the nucleosome surface. This interaction facilitates robust DNMT3A1 binding with nucleosomes and previously unexplained disease-associated mutations disrupt this interface. Furthermore, the UDR region nucleosome interaction synergises with other DNMT3A chromatin reading elements in the absence of histone ubiquitylation. Surprisingly, H2AK119ub does not stimulate DNA methylation activity, as observed for the previously described H3K36me2 mark, which may explain low levels of DNA methylation on H2AK119ub marked facultative heterochromatin. This study highlights the importance of multivalent binding of DNMT3A to histone modifications and the nucleosome and leads to an increased understanding of how DNMT3A1 recruitment occurs in the genome

2) In the first chapter of the results, the authors report that the DNMT3A1(1-427) region is essential for chromatin engagement and H3K36me2/3 binding. This claim is not in agreement with data reported in Weinberg et al. 2019, showing that the isolated PWWP domain specifically binds H3K36me2/3 containing nucleosomes and that DNMT3A2 expressed in cells delivers DNA methylation to chromatin regions containing H3K36me2.

We agree the PWWP is clearly important as reported by many studies (Weinburg 2019; Heyn 2019; Sendzikate 2019; Kibe 2021, Lue et al., 2023 and many others) and indeed our own discussed results. We showed HESJAS mutations that ablate histone methyl recognition greatly reduce binding (original supp fig S1E, now moved to main Fig 4B). We were not intending to suggest otherwise, simply that PWWP-H3K36me2 interactions are stabilised by additional interactions DNMT3A1 including the UDR region.

We do see preferential binding of just the PWWP to nucleosomes in the absence of competitor DNA (now included as Fig S8G). We have changed the way this is discussed in the manuscript to better reflect this.

3) In the second chapter of the results (starting on p. 7), the author study by XL analysis the interaction of DNMT3A1(1-427) with nucleosomes and observe an interaction with the acidic patch of nucleosomes. However, the information that DNMT3A/3B3 heterotetrameric complexes bind to the acidic patch is available in the literature since 2020, when a cryo-EM paper reporting this finding was published (Xu et al., Nature 586(7827):151-155). Again, this important part of literature is not mentioned. However, the authors mapped the acidic patch interaction to the N-terminal part, while Xu et al. show the ability of the C-terminal domain of DNMT3B3. This discrepancy can be explained because the authors used the DNMT3A1(1-427) fragment in the XL analysis. Hence, it remains open, which of these two interaction modes is more important under which condition.

Xu et al., 2020 was cited in the introduction and results section and the structure was discussed in the discussion. There is no discrepancy in our study: DNMT3A2-DNMT3B3 lacks the N-terminal region which is the main focus of our study. Throughout all observations on DNMT3A1(1-427) fragment are also validated for full length DNMT3A1-DNMT3L (lacking DNMT3B3 acidic patch interactor). The structure published by Xu et al. is obviously relevant, we originally discussed this in the discussion section when comparing our structure and directly addressing the comment above on how these two acidic patch binding modules may integrate and fascinating area of future study, we have now moved this to earlier in the manuscript. We note that there would be two acidic patch regions on either face of one nucleosome so it is conceivable that both could be engaged, one by DNMT3B3 one by DNMT3A1.

We have further purified a DNMT3A1-DNMT3B3 complex. Interestingly this complex shows a similar pattern in enzymatic assays as DNMT3A1-DNMT3L (new Figure 6C), except for a more marked affect of the acidic patch mutant, in line with the effect of acidic patch interacting DNMT3B3 mutations reported in Xu et al., 2020.

4) In the next results chapter (starting on p. 8), the authors studied methylation of nucleosomal DNA. Again, similar studies with better CpG site resolution have already been carried out, e.g. Bröhm et al. Commun Biol. 2022 5(1):192.

The Bröhm study is a very nice approach that we cited throughout. However, it was performed using murine DNMT3A2 at a single concentration point using a single DNA substrate and different histone modifications. Given the focus of our manuscript was on the role of multiple post-translational modifications (not reported in Bröhm) that interact with an N-terminal region that are lacking from DNMT3A2 the studies are very different. Our assays are performed across a range of substrate concentrations. We also utilised a greater array of nucleosome substrates including mutations and closer to native chemistry. Indeed, native chemistry appears to be important based on the

additional activity over the methyl-lysine analogues used previously (old figure 7C, now Figure S11E). This therefore provides a great deal of new insight beyond what has been previously shown.

Methyl-sequencing of our in vitro methylation reactions is important to show productive methylation, which we have now performed using nanopore based sequencing (new figure 6F). The result from this are in good agreement to the SAM turnover experiments.

It is unclear in Fig. 3C, why the DNA alone was not methylated at all, suggesting something went wrong with the experiment.

The observation of low activity on DNA alone is entirely expected and consistent with the literature. This is due to the auto-inhibitory nature of the ADD domain of DNMT3A1 catalytic activity, which is alleviated upon engagement of the H3 tail (Guo et al., 2015; Li et al, 2011; Zhang et al, 2010) and referenced in the original manuscript. We indeed see significantly more activity on nucleosomes over DNA alone, but this is reduced in nucleosomes lacking the N-terminal tail in line with the H3 tail activation, which was discussed in the manuscript in results section "H3K36me2 stimulates DNMT3A1-DNMT3L complex activity on nucleosomes". To make this more explicit we have now included our initial control experiments. DNA alone can of course be methylated, but this required far higher concentration of enzyme, way beyond the dynamic range of activity we see on chromatinized templates (now added in Fig S10D). We also used H3 peptides in trans to observe activation as has been previously reported (now added in Fig S10E and S11F).

They later conclude that DNMT3A1-DNMT3L can engage linker DNA, which again was not novel (see for example Brohm et al.).

In no way did we mean to suggest linker DNA engagement was novel we reported this when discussing our structure, in a similar vein to the overall structure of nucleosome is not novel we were simply explaining the context of our results.

5) The cryo-EM of the ubiquitylated nucleosome presented in Fig. 4 and 5 is novel, but to me the interaction is not very convincing (see for example Fig. 4C). More experiments need to be conducted testing the relevance of critical residues to validate this structural model.

It is unclear what further validation the reviewer would require to be convinced:

- The main map (Map 3) is at sufficient resolution to allow unambiguous modelling into the density and determine sequence register. Old Figure 4C (Map1) is a lower resolution map and we do not draw any conclusions based on this data beyond the presence of additional density concurrent with density on the nucleosome surface.
- We have made point mutations on DNMT3A1 in this region that ablate binding to both H3K36me2 and H2AK119ub nucleosomes.
- Conversely we have made point mutations on the nucleosome that ablate binding to DNMT3A1-427 (old figures 2C, 5C S2C; new figures 4D, 4A and 4E) and DNMT3A1-DNMT3L complex (Fig S8C). Crosslinking mass spectrometry corroborates the structure showing clear crosslinks between the region of interest in DNMT3A and H3K36me2 modified nucleosomes. We also show that DNMT3A1 shows less affinity for H2A.Z, which is in agreement to the observed interaction of the UDR region with the interface between the H2A and H2B dimer, where H2A.Z is divergent.
- We note our structure is in line with two others using a crosslinked smaller fragment of DNMT3A1 on H2AK119ub nucleosome core particle that have subsequently become

available after our submission (Gretarsson et al., *Science Advances* 2024; Chen et al., *Nature communications* 2024)

6) Finally, the authors report that the H2AK119ub interaction does not stimulate DNMT3A activity. However, based on their own observations hinting towards a close connection of H2AK119ub and H3K36me2 binding, a nucleosome carrying the double mark, would have been needed, which was not used. Instead, the H3K27me3/K36me3 double mark was tested, although it was shown before that H3K27me3 does not have an effect.

The experiment using a double mark is a good suggestion, especially given the co-incidence of these marks (Weinberg 2021; Conway et al, 2021; Fursova et al, 2019; Fursova et al, 2021). We have performed binding and enzymology assays using nucleosomes containing both H3K36me2 and H2AK119ub (Fig 5C/D, 6D,E). We see enzymatic activity and DNA methylation on these nucleosomes is similar to singly modified H3K36me2, suggesting that H2AK119ub is not in itself reducing catalytic activity.

Referee #3:

The manuscript outlines the potential functionalities of the DNMT3A1 N-terminal region (before the PWWP domain). This includes the binding of the nucleosome acidic patch and/or binding of ubiquitylated H2AK119 (as reported already by two recent studies). The N-terminal region, in conjunction with the PWWP domain's H3K36me2 binding, could guide DNMT3A1 to chromatin domains marked with intergenic H3K36me2 or heterochromatin H2AK119ub. In addition, the authors demonstrated that DNMT3A1 activity could be stimulated by H3K36me2 but not by H2AK119ub (despite strong binding). The study comprises two parts: in vitro binding/activity assays using modified reconstituted nucleosomes and a cryo-EM structural study of DNMT3A1-DNMT3L in complex with ubiquitylated nucleosomes.

We thank the reviewer for the positive appraisal and summary

A critical concern regarding experimental design is the choice of using DNMT3L in complex with DNMT3A1. Since DNMT3L primarily functions in the germline and is silenced upon differentiation, whereas DNMT3A1 is the predominant somatic isoform, the use of DNMT3A1-DNMT3L may be an artifact. The authors should provide a justification for choosing 3A1-3L over DNMT3A1-3B3, for instance. Additionally, one needs to consider the competitive binding of 3A1 and 3B3 to the same acidic patch, as demonstrated by a recent 3A2-3B3 structure.

This is an important concern and one we did not address fully in the initial submission and we have added extensive discussion in the revised manuscript (Introduction and Discussion). We chose DNMT3L as the activating substrate adaptor originally a technical necessity as we struggled to express and co-purify DNMT3A1 with DNMT3B3 in sufficient yield for structural studies. We agree using a DNMT3A1-DNMT3B3 would hold more direct relevance and we have since been successful with optimised constructs to produce sufficient high-quality DNMT3A1-DNMT3B3 methyltransferase domain complex (New Figure S11A).

Our DNMT3A1-DNMT3B3 is active and stimulated by nucleosomes. Similarly to reported by Xu 2020, we see that the acidic patch does affect catalytic activity; when we mutate this interface on the nucleosome it reduces catalytic activity. Intriguingly the same pattern of H3K36me2 stimulation but lack of H2AK119ub stimulation is retained for this complex. We present this new data in Figure 6C.

We do note that DNMT3L appears to be present with DNMT3A1 in some select tissues. Furthermore for our experiments we find using DNMT3L as an accessory factor to ensure activity without adding another nucleosome interacting region that may confuse our assays. DNMT3B3 and DNMT3A1 acidic patch interaction, while mutually exclusive on a single nucleosome face, would not be so in the context of two acidic patches per full nucleosome. As such comparing 3L and 3B3 in structural studies will be interesting in a future study.

The cryo-EM structure of 3A1-3L raises concern about its poor quality, lacking representation of any tetrameric features of 3A-3L. To address this, I recommend focusing on the structure of the N-terminal fragment (residues 1-427) in complex with the nucleosome (with either modifications of H3K36me2 or H2AK119ub), considering that all binding data were obtained with this fragment. Currently, there is a disconnect between the binding data and structural data in the manuscript. For instance, Figure 4A (and 4E) presents binding data with His-MBP-DNMT3A1 fragments, while the corresponding structures are depicted as DNMT3A1-DNMT3L. Focusing on the same fragment will enhance the manuscript's coherence and strengthen the overall argument.

Based on this comment and those of reviewer 1 we have shifted the focus of the revised manuscript more onto the DNMT3A1 N-terminal region, which we believe helps with the overall readability. Our map and model is high quality and we are confident the lower resolution observed for additional DNMT3A1-DNMT3L regions is biological rather than to do with technical limitations of the data, caused by structural heterogeneity inherent in the complex. We included the low-resolution structure of DNMT3A1-DNMT3L more to show our pipeline of investigation and confirm that the additional density over the acidic patch was concurrent with additional density on the linker DNA.

We have moved the lower-resolution structure of the full complex to supplementary figure S1E and include representations of this structure at multiple threshold levels (supp fig S1F). As shown, this did a disservice to the data, at lower thresholds the structure clearly shows features such as turn of DNA and separation between secondary structure elements in alpha helical octamer core indicative of the resolution we report. While the nucleosome core is overall at good resolution, the large mass of the nucleosome drives alignment during data processing. Due to the nature of averaging in single particle cryo-EM processing we often observe a range of resolutions indicative of flexibility across a map. Extensive structural heterogeneity of the untethered DNA linker combined with multiple possible DNMT conformations on substrate DNA leads to overall lower resolution represented in the local resolution estimation in old supp fig 5 new supp Fig 2 (bottom left).

We did attempt structural studies using smaller fragments of DNMT3A1, but these were unsuccessful we believe due to the necessary additional stabilising interactions from the ADD and methyltransferase domain present in the full-length protein in our study. We now discuss this observation in the discussion. Indeed we observed that the full length DNMT3A1-3L complex (Fig 1B and supp fig S8C) binds with higher affinity, but displays the same overall nucleosome binding patterns as NT-PWWP fragments. We note that a similar article that reports two complementary structures have recently been published (Gretarsson et al., *Science Advances* 2024 and Chen et al., *Nature communications* 2024). These use a short fragment of DNMT3A1 but required cross-linker to observe the density, not required in our structure.

As commented the tetramerization is in catalytic domains, not the region of interest in this manuscript. Nevertheless, it was important for us to use a well characterised and high-quality complex for our studies. DNMT3A1 and DNMT3L were equimolar and co-purified using a tandem affinity purification strategy with a large hydrodynamic radius indicative of tetrameric assembly (old supp fig S3A & B, current fig S1C & D). Extensive cross-links from cross-linking mass spectrometry are consistent with both DNMT3A Mtase-Mtase domain interaction and DNMT3L-Mtase interactions of a tetrameric assembly (new data Fig S1H)

We obtained 2D classes that were reminiscent of DNMT3A1-DNMT3L tetramer in isolation in the DNMT3A1-DNMT3L NCPH2AK119ub data, presumably from free DNMT complex not interacting with nucleosomes (reviewer figure 2). Multiple views could be obtained that match well with projections generated from tetramer crystal structures, but not single DNMT3A1 or DNMT3A1-DNMT3L dimers. These are also reminiscent of DNMT3A2-DNMT3B3 negative stain class (Zeng, Ren & Kaur et al., 2020 *Genes Dev*), which thoroughly characterised these complexes and inspired our work. Multiple bi-lobed orientations on the linker region that only match DNMT3-DNMT3L tetramers was also observed for the DNMT3A1-DNMT3L di-nucleosome-H3K36me2 data (old figure 3D and projection matched representations in reviewer figure 3). We are conscious to not overinterpret projection matching but this data and biochemistry suggests that we have purified a predominantly tetrameric of DNMT3A1-DNMT3L complex.

Reviewer Figure 2: Comparison of tetrameric DNMT3A1-DNMT3L crystal structure projections with 2D classes from our cryo-EM dataset. Crystal structure projections were made of two copies of PDB 4u7p, each containing DNMT3A1 catalytic domain and ADD domain and DNMT3L catalytic domain. To ensure correct tetramer orientation, PDB 4u7p was aligned with PDB 5yx2, containing tetrameric DNMT3A1-3L catalytic domain. The tetrameric model was then converted to a volume using UCSF Chimera 1.16. The volume was imported into Cryosparc v4.4.1 and converted to projections by generating templates. Templates were compared with 2D classes of protein-only particles found in the dataset of DNMT3A1-3L on H2AK119ub nucleosomes.

Reviewer Figure 3: Comparison of tetrameric DNMT3A1-DNMT3L crystal structure projections with 2D classes from cryo-EM dataset of H3Kc36me2 di-nucleosomes with DNMT3A1-DNMT3L. Crystal structure projections were made by combining PDB 4u7t, containing DNMT3A1 catalytic domain and ADD domain, DNMT3L catalytic domain and an H3 tail peptide with PDB 5yx2, containing tetrameric DNMT3A1-3L catalytic domain bound to DNA. The combined PDB model was then converted to a volume using UCSF Chimera 1.16. The volume was imported into Cryosparc v4.4.1 and converted to projections by

generating templates. Templates were compared with 2D classes of a dataset of H3Kc36me2 modified di-nucleosomes (DNA = Widom601 + Widom603 + 30 bp spacer) with DNMT3A1-DNMT3L as purified in the manuscript.

Other concerns:

"6xHis-MBP tagged fragments" - a positive control should be included for untagged fragment

We have included this control in figures S4D, S8B and see no affect due to the tag, beyond greater stabilisation overall of the protein fragments. We note we see similar binning patterns for the untagged DNMT3A1-DNMT3L complexes.

"PWWP domain alone bound poorly, likely due to the low affinity of the interaction with the histone mark ... and affinity for competitor DNA used in the assay buffer" - Does the PWWP domain bind nucleosome in the absence of competitor DNA?

We have performed this assay as suggested and see preferential binding of the PWWP in isolation to H3K36me2 modified nucleosomes alone (New Fig S8E-G), but as reported previously its binding is weak compared to larger fragments.

In streamlining the article we have removed this initial assay (old figure 1B) as we found it to be distracting to the overall message, but our results are in agreement with the PWWP driving recruitment to H3K36me2.

Page 7, line 3, The N-terminal region alone showed considerable interaction with nucleosomes irrespective of their modification state (Fig 1A), - wrong figure panel?

Thank you for spotting this, it has been rectified.

Dear Marcus,

Thank you for the submission of your revised manuscript. We have now received the enclosed reports from the referees that were asked to assess it, and I am happy to say that both support its publication. Referee 1 still has a few more suggestions that I would like you to incorporate before we can proceed with the official acceptance of your manuscript. I think that terms as H2AK119ub and H3K36me2 can stay in the ms text and do not need to be explained.

We can only offer to publish your ms if it can be included in our December issue, the deadline of which is on the 26th of October. Your ms will need to be exported to our publisher on the 25th of October, so please submit all final files latest on the 24th of October.

A few editorial requests will also need to be addressed:

- Please correct the conflict of interest subheading to "Disclosure Statement and Competing Interests"
- Please remove the author credits from the ms file. All credits need to be entered during online ms submission.
- Please move the References to after the Disclosure statement.
- The FUNDING INFO does not match. Please note that all funders and grant numbers acknowledged in the ms need to be entered during ms submission via the More Funders button even though they are not designated to a specific author; missing in eJP: - Institutional Strategic Support Fund (IS3-R1.37 22-23); BBS/E/B/000C0421; Darwin Trust of Edinburgh; Edinburgh Protein Production Facility (EPPF), (203149) to the Wellcome Centre for Cell Biology; Wellcome Trust (087658/Z/08/Z), proposal EM-BI24557 & EM-BI31827, funded by the Wellcome Trust; Wellcome Centre for Cell Biology (203149); Multi-User Equipment grant(s) (108504); Wellcome Discovery Research Platform for Hidden Cell Biology [226791]
- Fig 2A & B are called out before Fig 1D, please correct. Dataset 1 and 2 are called out but are missing.
- Movies are mentioned on p26, but are missing.
- Since this summer, all ms need to include a Reagents and Tools Table (listing key reagents, experimental models, software and relevant equipment and including their sources and relevant identifiers). A downloadable template (.docx) for the Reagents and Tools Table can be found in our author guidelines: <<https://www.embopress.org/page/journal/14693178/authorguide#manuscriptpreparation>>.
- Your ms has 11 Supplemental figures and their legends are in the ms file; 5-6 of them can be EV figures, the rest can go in an Appendix file; the correct nomenclature and ms callouts for EV figures should be: Figure EV1, Figure EV2, etc.; the correct nomenclature and ms callouts for Appendix should be Appendix Figure S1, Appendix Figure S2, etc. The Appendix file should have both the legends and the figures, and it also needs a table of content with page numbers.
- Supplemental Tables S1-S5 are in the ms, these need to be moved to the Appendix file and updated to Appendix Table S1, etc. (ms callouts included)
- Please note that n=2 in figures 2d; 6b. If n=2 no statistics should be calculated but please do show all individual data points along with their mean.
- Please note that the error bars are not defined in the legends of figures 6b-c, e-f.
- Please note that the specific URLs for EMD-18778 & EMD-18793, PDB 8QZM, PXD046529 & PXD056554 datasets need to be provided in the data availability statement.

I think the title could be a little more specific, e.g. recruitment of what? Do you mean:

The N-terminal region of DNMT3A engages the nucleosome surface to guide its recruitment.

I also made a few minor changes to the abstract. Could you please check carefully whether all info is still correct:

DNA methyltransferase 3A (DNMT3A) plays a critical role in establishing and maintaining DNA methylation patterns in vertebrates. Here we structurally and biochemically explore the interaction of DNMT3A1 with diverse modified nucleosomes indicative of different chromatin environments. A cryo-EM structure of the full-length DNMT3A1-DNMT3L complex with an H2AK119ub nucleosome reveals that the DNMT3A1 UDR motif interacts specifically with H2AK119ub and also makes extensive contacts with the nucleosome surface. This interaction facilitates robust DNMT3A1 binding to nucleosomes, and previously

unexplained disease-associated mutations disrupt this interface. Furthermore, the UDR-nucleosome interaction synergises with other DNMT3A chromatin reading elements in the absence of histone ubiquitylation. Surprisingly, H2AK119ub does not stimulate DNMT3A DNA methylation activity, as observed for the previously described H3K36me2 mark, which may explain low levels of DNA methylation on H2AK119ub marked facultative heterochromatin. This study highlights the importance of multivalent binding of DNMT3A to histone modifications and the nucleosome and increases our understanding of how DNMT3A1 chromatin recruitment occurs.

EMBO press papers are accompanied online by A) a short (1-2 sentences) summary of the findings and their significance, B) 2-3 bullet points highlighting key results and C) a synopsis image that is exactly 550 pixels wide and 200-600 pixels high (the height is variable). The synopsis image should provide a sketch of the major findings, like a graphical abstract. Please note that text needs to be readable at the final size. Please send us this information along with the final manuscript.

I look forward to seeing a final version of your manuscript at the very latest on the 24th of October.

Referee #1:

The authors have addressed most of my concerns. However, the clarity of the presentation could be significantly improved.

Below are a few examples from the abstract:

Special terms like UDR, H2AK119ub, and H3K36me2 should either be defined or avoided if possible.

Clarify what is meant by "the nucleosome surface."

The sentence "mutations disrupt this interface" is unclear. Mutations of what? Which interface is being disrupted (is it the nucleosome surface mentioned in the previous sentence)?

The word "Surprisingly" should be omitted—simply state the result and observation without implying surprise.

In the last sentence, clarify whether "histone modifications and the nucleosome" are being referred to as two different entities.

Page 3, second paragraph: "a S-adenosylmethionine (SAM) donor" is incorrect. SAM is a methyl group donor. The reference (Song et al., 2012) is inappropriate for this context. DNA methylation using SAM as a methyl donor was first described in the 1950s.

Page 3, third paragraph: What is meant by "inactivated CpG islands"? Unmethylated CpG island promoters require protection to maintain their hypomethylated status for active gene expression.

I will not list every problematic sentence. The main takeaway is that simplifying the text and reducing unnecessary details will improve the overall clarity.

Referee #2:

The authors have addressed the critical concerns. The manuscript presents a broad biochemical characterisation of the DNMT3A-nucleosome interaction. This complements and adds onto the recently published studies. Specifically, the paper present original data on the binding analysis with different ubiquitinated nucleosomes, the measurement of DNMT3 activity in vitro, and the use of several DNMT3A mutations. This study should be published in EMBO reports.

Dr Marcus D. Wilson
Michael Swann Building
University of Edinburgh
Max Born Crescent
Edinburgh EH9 3BF
United Kingdom

18th October 20234

Dear Esther,

Further to our correspondence, please find enclosed our modified manuscript, encapsulating comments by referee and requested Editorial items. See below for a detailed summary of changes.

Yours sincerely,

Marcus D Wilson

Sir Henry Dale Fellow

Tel: +441316507029

Email: marcus.wilson@ed.ac.uk

www.mdwilsonlab.com

Summary of changes and review experiments

Wapenaar et al., The N-terminal region of DNMT3A combines multiple chromatin reading motifs to guide recruitment.

Manuscript changes

- Title now changed to:

The N-terminal region of DNMT3A engages the nucleosome surface to aid chromatin recruitment.

- Abstract changed to:

DNA methyltransferase 3A (DNMT3A) plays a critical role in establishing and maintaining DNA methylation patterns in vertebrates. Here we structurally and biochemically explore the interaction of DNMT3A1 with diverse modified nucleosomes indicative of different chromatin environments. A cryo-EM structure of the full-length DNMT3A1-DNMT3L complex with a H2AK119ub nucleosome reveals that the DNMT3A1 **ubiquitin-dependent recruitment** (UDR) motif interacts specifically with H2AK119ub and also makes extensive contacts with the **core** nucleosome **histone** surface. This interaction facilitates robust DNMT3A1 binding to nucleosomes, and previously unexplained **DNMT3A** disease-associated mutations disrupt this interface. Furthermore, the UDR-nucleosome interaction synergises with other DNMT3A chromatin reading elements in the absence of histone ubiquitylation. H2AK119ub does not stimulate DNMT3A DNA methylation activity, as observed for the previously described H3K36me2 mark, which may explain low levels of DNA methylation on H2AK119ub marked facultative heterochromatin. This study highlights the importance of multivalent binding of DNMT3A to histone modifications and the nucleosome **surface** and increases our understanding of how DNMT3A1 chromatin recruitment occurs.

- Update references to figures and figure legends throughout to reflect new numbering.
- Minor changes to manuscript to reflect comments from referee 1, to simplifying the text and reducing unnecessary details:

Page 3 deleted

“A pivotal player in the dynamic management of vertebrate genomes is DNA methylation, an epigenetic mark on cytosine bases which generally represses transcription.”

For brevity

Page 3 deleted

“DNA methyltransferases (DNMTs) catalyse the transfer of a methyl group from a S-adenosylmethionine (SAM) donor to the C5 position of a cytosine base (Song *et al*, 2012).” As suggested by referee, unnecessary detail

Page 3 added reference *Taglini et al 2024*

Page 3 removed:

“Comprised of two copies each of the four core histones, the nucleosome wraps and compacts ~147 bp of DNA around its globular core (Luger *et al*, 1997).”

Assumed detail of reader, not necessary to define

Page 3 removed

“developmentally inactivated’ as suggested by referee

Page 3 removed

“DNMT3B3 has been shown to interact directly with the nucleosome acidic patch surface, using a feature absent from other DNMT3B isoforms, DNMT3L or DNMT3A (Xu *et al.*, 2020a).

The recruitment of DNMT3A to specific genomic loci is a highly regulated process, finely tuned to ensure the fidelity of DNA methylation pattern. This is mediated in part,”

Duplication mentioned in results and discussion section where more relevant

Page 4

“Unmodified H3 tail recognition by the ADD domain not only drives correct localisation but is important in promoting full catalytic activity (Guo *et al.*, 2015; Li *et al.*, 2011; Zhang *et al.*, 2010).”

Unnecessary detail, discussed more appropriately in results section where this is relevant.

Page 4

Reworded section on Polycomb enrichment to remove duplication

Page 5

Removed unnecessary experimental detail from start of results

Page 5

Removed unnecessary experimental detail on cryo-EM image processing

Page 6

Removed unnecessary experimental detail describing EMSA assay set up

Page 7

Top, rephrased for brevity

Page 9

Removed unnecessary experimental detail describing making ubiquitylated nucleosomes

Page 10

Rephrased discussion on ubiquitin binding to remove repetition

Page 13

Removed

“This elevated activity is presumably mediated through increased recruitment of the enzyme to substrate but may also be due to release of PWWP mediated inhibition and/or increased availability of substrate methylation sites as observed for methyltransferase domains alone (Brohm *et al.*, 2022).”

more relevant on page 14, removed to avoid duplication

Page 16

second paragraph, rephrased for brevity

Page 16 middle removed

“Due to technical reasons, we used DNMT3A1-DNMT3L complex in this study rather than the more likely accessory candidate DNMT3B3 (Duymich *et al.*, 2016; Xu *et al.*, 2020a; Zeng *et al.*, 2020). This allowed us to deconvolve the nucleosome surface binding effect of DNMT3A1, which would have been more complex in the presence of acidic patch binding DNMT3B3 (Xu *et al.*, 2020a).”

For clarity, unnecessary technical detail already stated in results. main point still salient without this addition

Page17

rephrased paragraph on other H2AK119ub readers to streamline and include recently published paper (Ciapponi *et al*, 2024)

Figure changes

Fig 2D full data shown for all replicates (as suggested)

Fig 6B full data shown for all replicates (as suggested)

Fig 6F full data shown for all replicates

Renamed supplements and moved to either EV or appendix.

Editorial requests:

- Please correct the conflict of interest subheading to “Disclosure Statement and Competing Interests”

Changed

- Please remove the author credits from the ms file. All credits need to be entered during online ms submission.

Changed and amended fully in submission

- Please move the References to after the Disclosure statement.

Changed

- The FUNDING INFO does not match. Please note that all funders and grant numbers acknowledged in the ms need to be entered during ms submission via the More Funders button even though they are not designated to a specific author; missing in eJP: - Institutional Strategic Support Fund (IS3-R1.37 22-23); BBS/E/B/000C0421; Darwin Trust of Edinburgh; Edinburgh Protein Production Facility (EPPF), (203149) to the Wellcome Centre for Cell Biology; Wellcome Trust (087658/Z/08/Z), proposal EM-BI24557 & EM-BI31827, funded by the Wellcome Trust; Wellcome Centre for Cell Biology (203149); Multi-User Equipment grant(s) (108504); Wellcome Discovery Research Platform for Hidden Cell Biology [226791]

Matched in the submission

- Fig 2A & B are called out before Fig 1D, please correct. Dataset 1 and 2 are called out but are missing.

Slight modification to text to adjust for Fig 1D/Fig2A. We have changed the figures so that Dataset 1 and 2 are now referred to appropriately in old supp fig S2 (new Appendix Fig 1) and old supp table s1 (new appendix table s1)

- Movies are mentioned on p26, but are missing.

These refer the raw cryo-EM images that are taken as a collection of individual frames, which the field refers to as movies. Any sample movement between frames is then accounted for and a single

stacked images is then used for subsequent steps. We think this is the appropriate way to discuss this data.

- Since this summer, all ms need to include a Reagents and Tools Table (listing key reagents, experimental models, software and relevant equipment and including their sources and relevant identifiers). A downloadable template (.docx) for the Reagents and Tools Table can be found in our author guidelines:

< <https://www.embopress.org/page/journal/14693178/authorguide#manuscriptpreparation>>;

Included at start of methods in main manuscript file

- Your ms has 11 Supplemental figures and their legends are in the ms file; 5-6 of them can be EV figures, the rest can go in an Appendix file; the correct nomenclature and ms callouts for EV figures should be: Figure EV1, Figure EV2, etc.; the correct nomenclature and ms callouts for Appendix should be Appendix Figure S1, Appendix Figure S2, etc. The Appendix file should have both the legends and the figures, and it also needs a table of content with page numbers.

Figures changed to conform with expanded view and appendix nomenclature

Old nomenclature	New nomenclature
Supplementary Figure S1	Expanded View Figure EV1
Supplementary Figure S2	Appendix Figure S1
Supplementary Figure S3	Appendix Figure S2
Supplementary Figure S4	Appendix Figure S3
Supplementary Figure S5	Appendix Figure S4
Supplementary Figure S6	Appendix Figure S5
Supplementary Figure S7	Expanded View Figure EV2
Supplementary Figure S8	Expanded View Figure EV3
Supplementary Figure S9	Appendix Figure S6
Supplementary Figure S10	Expanded View Figure EV4
Supplementary Figure S11	Expanded View Figure EV5

No changes to Figures otherwise, apart from inclusion of Dataset number in Appendix Figure S1 and modification of figure legends as requested.

- Supplemental Tables S1-S5 are in the ms, these need to be moved to the Appendix file and updated to Appendix Table S1, etc. (ms callouts included)

Included in Appendix file

- Please note that $n=2$ in figures 2d; 6b. If $n=2$ no statistics should be calculated but please do show all individual data points along with their mean.

altered

- Please note that the error bars are not defined in the legends of figures 6b-c, e-f.

These are standard error of mean, mentioned now in revised Figure legend

- Please note that the specific URLs for EMD-18778 & EMD-18793, PDB 8QZM, PXD046529 & PXD056554 datasets need to be provided in the data availability statement.

URLs provided, requested release form repositories

-Possible blot reuse between (email form data integrity)

Now mentioned in the Figure legend. For further details see email 17th October

-Raw Source data

Now provided as single zip file per main figure, including raw data and annotation. 1 zipped file for all expanded view and appendix figures.

- a short (1-2 sentences) summary of the findings and their significance, B) 2-3 bullet points highlighting key results and C) a synopsis image

Now included in resubmission

Dr. Marcus Wilson
University of Edinburgh
Michael swann building
edinburgh, Lothian eh9 3jr
United Kingdom

Dear Marcus,

I am very pleased to accept your manuscript for publication in the next available issue of EMBO reports. Thank you for your contribution to our journal.

Best,
Esther
